# Representing methane emissions from wet tropical forest soils using microbial functional groups constrained by soil diffusivity

Debjani Sihi[1,2], Xiaofeng Xu[3], Mónica Salazar Ortiz[4], Christine S. O'Connell[5,6], Whendee L. Silver[5], Carla López-Lloreda[7], Julia M. Brenner[1], Ryan K. Quinn[1,8], Jana R. Phillips[1], Brent D. Newman[9], and Melanie A. Mayes[1]*

[1]Climate Change Science Institute and Environmental Sciences Division, Oak Ridge National Laboratory, Oak Ridge, TN, 37831, USA
[2]Currently employed at Department of Environmental Sciences, Emory University, Atlanta, GA, 30322, USA
[3]Department of Biology, San Diego State University, San Diego, CA, 92182-4614, USA
[4]Institute of Plant Science and Microbiology, University of Hamburg, Hamburg, 20148, Germany
[5]Department of Environmental Science, Policy and Management, University of California, Berkeley, CA, 94720-3114, USA
[6]Currently employed at Department of Environmental Studies, Macalester College, St. Paul, MN, 55105-1899, USA
[7]Department of Natural Resources and the Environment, University of New Hampshire, Durham, NH, 03824, USA
[8]Department of Biology, Boston University, Boston, MA, 02215, USA
[9]Earth and Environmental Sciences Division, Los Alamos National Laboratory, Los Alamos, NM, 87545, USA

*Correspondence to*: Melanie A. Mayes (mayesma@ornl.gov)

**Abstract.** Tropical ecosystems contribute significantly to global emissions of methane ($CH_4$) and landscape topography influences the rate of $CH_4$ emissions from wet tropical forest soils. However, extreme events such as drought can alter normal topographic patterns of emissions. Here we explain the dynamics of $CH_4$ emissions during normal and drought conditions across a catena in the Luquillo Experimental Forest, Puerto Rico. Valley soils served as the major source of $CH_4$ emissions in a normal precipitation year (2016), but drought recovery in 2015 resulted in dramatic pulses in $CH_4$ emissions from all topographic positions. Geochemical parameters including dissolved organic carbon (C), acetate, and soil pH; and hydrological parameters like soil moisture and oxygen ($O_2$) concentrations, varied across the catena. During the drought, soil moisture decreased in the slope and ridge and $O_2$ concentrations increased in the valley. We simulated the dynamics of $CH_4$ emissions with the Microbial Model for Methane Dynamics-Dual Arrhenius and Michaelis Menten (M3D-DAMM) which couples a microbial functional group $CH_4$ model with a diffusivity module for solute and gas transport within soil microsites. Contrasting patterns of soil moisture, $O_2$, acetate, and associated changes in soil pH with topography regulated simulated $CH_4$ emissions, but emissions were also altered by rate-limited diffusion in soil microsites. Changes in simulated available substrate for $CH_4$ production (acetate, $CO_2$, and $H_2$) and oxidation ($O_2$ and $CH_4$) increased the predicted biomass of methanotrophs during the drought event and methanogens during drought recovery, which in turn affected net emissions of $CH_4$. A variance-based sensitivity analysis suggested that parameters related to aceticlastic methanogenesis and methanotrophy were most critical to simulate net $CH_4$ emissions. This study enhanced the predictive capability for $CH_4$ emissions associated with complex topography and drought in wet tropical forest soils.

**1 Introduction**
Wet tropical forest soils contribute significantly to global emissions of methane ($CH_4$; Pachauri et al., 2014).
Although net emissions of $CH_4$ from upland soils are infrequent in temperate climates, studies show that $CH_4$
emissions are common in wet tropical forests, even in upland soils (Cattânio et al., 2002; Keller and Matson, 1994;
Silver et al., 1999; Teh et al., 2005; Verchot et al., 2000). Landscape topography can strongly influence the
proportions of $CH_4$ production and oxidation in mountainous tropical regions, affecting net emissions (Silver et al.,
1999; O'Connell et al., 2018). Climate, and specifically patterns in rainfall, also affect emissions from tropical
forests. Climate change may increase the frequency and severity of extreme rainfall and drought events, altering the
spatial and temporal dynamics of $CH_4$ emissions through changes in redox dynamics and substrate availability
(Silver et al., 1999; Chadwick et al., 2016; Neelin et al., 2006). Thus, accurately estimating $CH_4$ emissions under a
variety of climatic and topographic conditions is important for predicting soil carbon-climate feedbacks in the humid
tropical biome.
Several studies have reported the effect of drought events on biogenic $CH_4$ emissions across different wet tropical
forest soils. For example, Aronson et al. (2019) demonstrated that the lower soil moisture conditions during 2015-16
El Niño event increased consumption of atmospheric $CH_4$ in a wet tropical forest Oxisol of Costa Rica. Similarly, a
large-scale, 5-year throughfall exclusion experiment in a moist tropical forest Oxisol in Brazil also reported
increased consumption of atmospheric $CH_4$ under the drought treatment, followed by a recovery of $CH_4$ emissions to
pre-treatment values after the experiment ceased (Davidson et al., 2004, 2008). Using rainout shelters, Wood and
Silver (2012) found spatial variability in $CH_4$ oxidation rates, with an increase of 480% uptake in valleys in an
Ultisol in Puerto Rico. More recently, in a similar Puerto Rico Ultisol, O'Connell et al. (2018) reported increasing
consumption of atmospheric $CH_4$ during a Caribbean drought event, followed by increased production of $CH_4$ after
the drought was over. The post-drought net $CH_4$ emission rates were higher than the pre-drought emissions, such
that the benefits to atmospheric radiation imparted by the lowered emissions during the drought were eliminated.
The sharp differences between pre- and post-drought emissions suggested that drought affected the balance of
methanogenesis and methanotrophy in the soils, but the study lacked analysis of the microbial community's
contributions to these two separate processes.
The concept of "microsites" inside soil aggregates or within soil micropores can help explain the coexistence of
oxidative and reductive processes in soils (Silver et al., 1999; Teh and Silver, 2006), which may have occurred in the
post-drought period in the O'Connell et al. (2018) study. Oxygen can remain inside micropores during saturated
conditions and thereby maintain aerobic microbial respiration; likewise, hypoxic conditions can persist in microsites
under extended droughts and thereby maintain anaerobiosis. Additionally, liquid substrates for methanogenesis such
as acetate can accumulate in microsites under dry conditions because their diffusion to hungry microbial
communities may be restricted. Conversely, gaseous substrates such as $CO_2$ and $H_2$ may accumulate in microsites
under saturated conditions because gaseous diffusion can be limited. The observed rapid flush of $CH_4$ in response to
a post-drought wetting event (O'Connell et al., 2018) suggests methanogenesis continued during the drought in the
Ultisol's microsites, despite low soil moisture and high $O_2$ supply (Andersen et al., 1998; Bosse and Frenzel, 1998;
Teh et al., 2005; von Fischer and Hedin, 2002). Finely-textured soils common to the humid tropics can facilitate the
co-existence of reduced solute and gas species with $O_2$ because the rate of solute and gaseous exchanges is
controlled by diffusion into and out of microaggregates (Hall and Silver, 2013; Liptzin et al., 2010; Silver et al.,
2013). In particular, hematite precipitation on clay minerals, found in both Oxisols and Ultisols, can enhance
formation of soil aggregates because of their high surface area and charge properties (Hall et al., 2016). Soil organic
matter can also enhance aggregation and at the same time consume $O_2$ (Six et al., 2004). However, few if any
measurements of microsites exist in real field soils.
To explain the diverse observations of $CH_4$ emissions during and after drought across a wet tropical forest catena,
we hypothesized that explicit representations of diffusion into and out of microsites for gas and solute transport
would be required. To account for the balance of methanotrophy and methanogenesis, separate microbial functional
groups for $CH_4$ production and oxidation would need to be defined. Therefore, a microbial functional group model
for $CH_4$ production and consumption (Xu et al., 2015) was merged with a soil diffusivity module (Davidson et al.,
2012; Sihi et al., 2018) to simulate the dynamics of net *in situ* $CH_4$ emissions from soil microsites (Sihi et al.,
2020a). This module considers three key mechanisms for $CH_4$ production and consumption: aceticlastic
methanogenesis (production from acetate) and hydrogenotrophic methanogenesis (production from $H_2$ and $CO_2$), and
aerobic methanotrophy (oxidation of $CH_4$ and reduction of $O_2$) (Fig. 1). Here we report a modeling experiment to
explain contrasting patterns of observed $CH_4$ emissions following a severe drought in 2015 and we provide new data
to describe $CH_4$ emissions under non-drought conditions in 2016. We explicitly account for changes in soil
moisture, $O_2$, acetate, and microbial functional group dynamics within soil microsites in the model.
**2 Materials and methods**
**2.1 Study site**
The study was conducted across a tropical forest catena near the El Verde Research Station in the Luquillo
Experimental Forest (LEF) in northeastern Puerto Rico in the United States (Latitude 18°19'16.83" N, Longitude
65°49'10.13" W). The site is part of a National Science Foundation Long-Term Ecological Research (LTER) and
Critical Zone Observatory (CZO) site and is also part of the U.S. Department of Energy's Next Generation
Ecosystem Experiment-Tropics. The mean annual temperature at the site is 23 °C and the long-term mean rainfall is
approximately 3500 mm $yr^{-1}$ with low seasonality (Scatena, 1989). Inter-annual variability of rainfall ranges
between 2600 mm $yr^{-1}$ to 5800 mm $yr^{-1}$, sometimes associated with extreme rainfall events (approximately 100 mm
$day^{-1}$) from Caribbean storm systems (Heartsill-Scalley et al., 2007). The LEF is classified as a wet tropical forest
according to the Holdridge life zone system, which considers rainfall, elevation, latitude, humidity, and
evapotranspiration (Harris et al., 2012).
The landscape at the field site is highly dissected with short catenas, characterized by a land surface distance of < 30
m from ridgetop to valley (O'Connell et al., 2018). This study partitioned sampling along a catena from ridgetop,
slope, and valley topographic positions (Fig. S1). The soils are clay-rich Ultisols, which were derived from basaltic
and andesitic volcanoclastic parent materials. Soils are acidic (average pH is 4.3 and 5.1 in ridge and valley
topographic positions, respectively, Fig. 2). The valley soils have approximately 30% clay and approximately 15%
sand, while the ridge soils have approximately 22% clay and approximately 30% sand (Brenner et al., 2019). The
soils contain high concentrations of iron (Fe) and aluminum (Al) (oxy)hydroxides where their relative
concentrations vary along the catena and differences in Fe speciation are associated with variable redox conditions
(Hall and Silver, 2013, 2015). A detailed soil survey in the immediate vicinity lists three soil types (Zarzal, Cristol,
and Prieto) with a minimal litter layer due to rapid decomposition (Parton et al. 2007, Cusack et al. 2009), surface
(A) horizons 5 cm thick, and B horizons of 130 to 150 cm thick (Soil Survey Staff, 1995). The surface soil bulk
density ranged from 0.5 to 0.7 g cm$^{-3}$, and by 25 cm depth was 0.8 to 1.1 g cm$^{-3}$ (Cabugao et al., in press), similar to
previous observations (Johnson et al., 2014; Silver et al., 1999). The forest composition is relatively diverse with the
mature Tabonuco (*Dacryodes excelsa* Vahl) and Sierra palm (*Prestoea montana*) trees being most dominant
(Scatena and Lugo, 1995; Wadsworth et al., 1951).
**2.2 Soil and porewater sampling**
Previous $CH_4$ measurements in the LEF at the soil surface, 10 cm depth, and 35 cm depth, found the highest $CH_4$
concentrations at 10 cm depth (Silver et al. 1999), while 30 cm depth was the location of maximum soil organic
carbon (SOC) concentrations (Johnson et al., 2014). To initialize the model, soil and soil water samples were
collected from depths ranging from 0 to 30 cm in accordance with these previous studies. Soil samples were
collected in triplicate from a depth of 0-10 cm and on a quarterly timeframe from the ridgetop, slope, and valley
positions for over two years. The soil pH was determined using a 1:2 ratio of soil:solution using a glass electrode
with 0.005 M $CaCl_2$ as the equilibrated soil solution (Thomas, 1996; Sihi et al. 2020b). Porewater samples were
collected approximately weekly for over two years using macro-rhizon soil water samplers (length = 5 cm)
(Rhizosphere Research Products B.V.; Wageningen, The Netherlands) installed at both 5-10 cm and 25-30 cm depth
in triplicate in the ridge, slope, and valley topographic positions (Sihi et al., 2020c). The soil water samples were
analyzed for organic acid concentrations (acetate) using High Performance Liquid Chromatography (Dionex ICS-
5000+ Thermo-Fisher Waltham, MA, USA) with the Dionex IonPac AS11-HC column using a potassium hydroxide
eluent and gradient elution. The samples were analyzed for total dissolved organic carbon (DOC) using a Shimadzu
total organic C analyzer (Shimadzu TOC-L CSH/CSN Analyzer Baltimore, MD, USA). The soil and porewater
measurements were conducted in 2017-2018 (the number of samples $n$ ranged between 20 to 35, Fig. 2) to initialize
different model parameters for the catena, because measurements were not available for 2015-2016. To that end, the
chemical data were used as the reference characteristics of the bulk soil, and the temporal evolution of DOC, acetate,
and soil pH at the microsites were calculated using probability distributions of soil moisture and $O_2$ across soil
microsites over the two-year measurement window. Soil bulk density and particle density values were taken from
O'Connell et al. (2018).

## 2.3 In situ methane flux and soil driver measurements

Campbell Scientific CS 655 soil moisture and temperature sensors and Apogee SO-110 $O_2$ sensors were co-located with soil gas flux chambers at 15 cm soil depth along the catena, each with five replications along five transects (Fig. S1) (O'Connell et al., 2018). Following Liptzin et al. (2011), soil $O_2$ sensors were installed in gas-permeable soil equilibration chambers (295 $cm^3$). Data from these sensors were collected hourly using Campbell Scientific CR10000 data loggers and AM16/32B multiplexers (Campbell Scientific, Logan, UT, USA), which were processed using site-based calibration equations.

Soil flux chambers were placed on the top of the soil surface. Soil $CH_4$ emissions along the catena were measured during 2015 (February 26 to December 23, O'Connell et al., 2018; Silver, 2018) and 2016 (April 5 to July 18) (Sihi et al., 2020d) using a Cavity Ring-Down Spectroscopy gas analyzer (Picarro G2508, Santa Clara, CA, USA) connected to 12 automated eosAC closed dynamic soil chambers (Pumpanen et al., 2004) using a multiplexer (Eosense Inc., Dartmouth, Nova Scotia, Canada). Data for soil $CH_4$ emissions were processed using eosAnalyze-AC (v3.5.0) software followed by a series of quality control protocols (O'Connell et al., 2018). We used daily average values of drivers (soil temperature, soil moisture, and $O_2$ concentrations) and $CH_4$ emissions in the modeling exercise. See O'Connell et al (2018) for more information on the soil sensor, chamber arrays, and the data analysis pipeline.

The data from the 2015 Caribbean drought was partitioned into four distinct periods (O'Connell et al., 2018): (1) pre-drought from day of year (DOY) 57 to 115 (dark gray on Fig. 3), (2) the drought from DOY 116 to 236 (medium gray on Fig. 3), (3) drought recovery from DOY 237 to 328 (light gray on Fig. 3), and (4) post-drought from DOY 329 to 354 (white on Fig. 3). Total precipitation during the drought period was 700 mm in 2015 and 1088 mm during the same time frame in 2016 (Meteorological data from El Verde Field Station: NADP Tower, available at https://luq.lter.network/data/luqmetadata127).

## 2.4 Modelling approach

### 2.4.1 Microbial functional group model for methane production and oxidation

An existing microbial functional group-based model for $CH_4$ production and consumption (Xu et al., 2015) was adopted for this research (Sihi, 2020). As shown in Fig. 1, acetate and $H_2/CO_2$ represent substrate [Substrate$_{func_i}$] (nM $cm^{-3}$) for aceticlastic and hydrogenotrophic methanogenesis reactions, respectively. On the other hand, $CH_4$ and $O_2$ concentrations represent substrate for the methanotrophy reaction. Acetate and $CO_2$ are inputs based on measurements of soil water and pH described in Section 2.2. In the model, acetate is formed by fermentation and by homoacetogenesis (but not by syntrophic acetate oxidation) as defined in Xu et al. (2015) in their Appendix in Eq. A15 and A16 (Fig. 1b). Methylotrophic methanogenesis (Narrowe et al. 2019) is neglected in the model. The overall reaction rates are represented as:

$$\text{Reaction}_{\text{rate}_i} = \text{Biomass}_{\text{func}_i} \times \frac{\text{GrowR}_{\text{func}_i}}{\text{Efficiency}_{\text{func}_i}} \times \frac{[\text{Substrate}_{\text{func}_{1...n}}]}{[\text{Substrate}_{\text{func}_{1...n}}] + \text{KM}_{\text{func}_{1...n}}} \times f(T) \times f(pH) \tag{1}$$

where $Reaction_{rate_i}$ (in nM $cm^{-3}$ $hr^{-1}$) is rate of $CH_4$ production and/or consumption under variable substrate
concentrations. $Biomass_{func_i}$ (nM $cm^{-3}$) represents microbial functional groups: aceticlastic methanogens,
hydrogenotrophic methanogens, and aerobic methanotrophs, respectively. Growth rates and substrate use
efficiencies of microbial functional groups are represented as $GrowR_{func_i}(hr^{-1})$ and $Efficiency_{func_i}$ (unitless),
respectively (Table 1). The substrate limitation on $CH_4$ production is imposed by assuming a Michaelis-Menten
relationship with the half-saturation constants for $CH_4$ production and oxidation being $KM_{func_{1...n}}$ (nM $cm^{-3}$).
Although minor contributions of iron dependent anaerobic $CH_4$ oxidation to net $CH_4$ emissions can be expected in
our study site (Ettwig et al., 2016), we did not represent this process here as anaerobic oxidation of $CH_4$ is still not
fully understood and it is generally low in most ecosystems.
The extent of change in $Biomass_{func_i}$ ($dBiomass_{func_i}$) is controlled by the balance between $Growth_{func_i}$ and
$Death_{func_i}$ following:
$$\frac{dBiomass_{func_i}}{dt_{func_i}} = Growth_{func_i} - Death_{func_i} \qquad (2)$$
$$Growth_{func} = Efficiency_{func_i} \times Reaction_{rate_i} \qquad (3)$$
where $Growth_{func_i}$ is calculated as a multiplicative function of $Efficiency_{func_i}$ and the $Reaction_{rate_i}$,
$$Death_{func_i} = DeadR_{func_i} \times Biomass_{func_i} \qquad (4)$$
and $Death_{func_i}$ is a function of $DeadR_{func_i}$ (death rate, Table 1) and $Biomass_{func_i}$ (microbial biomass).
All rate equations were modified by the scalers for temperature, f(T) and pH, f(pH) functions, described below. We
represented the temperature effect, f(T), using a classic $Q_{10}$ function:
$$f(T) = Q_{10_i}^{\frac{Temperature_{soil} - Temperature_{reference}}{10}} \qquad (5)$$
We represented the pH effect, f(pH), based on Cao et al. (1995):
$$f(pH) = \frac{(pH - pH_{minimum}) * (pH - pH_{maximum})}{(pH - pH_{minimum}) * (pH - pH_{maximum}) - (pH - pH_{optimum})^2} \qquad (6)$$
where we set the minimum, optimum, and maximum soil pH values to 4, 7, and 10, respectively. Following Xu et al.
(2015), we considered the contribution of acetate to pH as follows:
$$pH = -1 * \log(10^{pH_{initial}} + 4.2E - 9 * Acetate) \qquad (7)$$
Although other mechanisms to alter soil pH are present at the site, e.g., Fe reduction and oxidation (Teh et al., 2005;
Hall and Silver, 2013), these are not considered in the model at this time. Calibrated values of $GrowR_{func_i}$,
$DeadR_{func_i}$, $Efficiency_{func_i}$, $KM_{func_i}$, and $Q_{10_i}$ are presented in Table 1.

**2.4.2 Diffusion module for gaseous and solute transport in soil profile and across soil-air boundary**

In order to account for the diffusion of gases across the soil-air boundary and solutes (e.g. acetate) through soil water
films (Fig. 1), we added the diffusion module of the Dual Arrhenius and Michaelis Menten (DAMM) model
(Davidson et al., 2012; Sihi, 2020; Sihi et al., 2018, 2020a) to the existing microbial functional group model, which
we refer to as M3D-DAMM. We calculated initial concentration of gases like $O_2$, $H_2$, $CO_2$, and $CH_4$, [$Gas_{conc}$], (unit:
$V V^{-1}$), as a function of a unitless diffusion coefficient of gas in air ($D_{gas}$), volume fraction of gas in air ($V V^{-1}$), and
gas diffusivity ($a^{4/3}$) as follows:
$[Gas_{conc}] = D_{gas} \times$ atmospheric concentration $\times a^{4/3}$         (8)
where $a^{4/3}$ represents the tortuosity of diffusion pathway for gases as a function of soil water (SoilM) and
temperature (SoilT):
$a^{4/3} = \left(Porosity - \frac{SoilM}{100}\right)^{4/3} \times \left(\frac{SoilT+273.15}{293.15}\right)^{1.75}$       (9)
where the air-filled porosity (a) was calculated by subtracting the volume fraction of soil moisture ($V V^{-1}$) from total
porosity. Porosity was calculated as:
$(1 - \frac{Bulk\ density}{Particle\ density})$         (10)
The exponent of 4/3 accounts for diffusivity of gases through porous media (Davidson and Trumbore, 1995). The
exponent of 1.75 represents the temperature response of gaseous diffusion (Massman, 1998; Davidson et al., 2006).
Following Davidson et al. (2012), the value used for gaseous diffusivity coefficient ($D_{gas}$) was calculated based on
an assumed boundary condition such that the concentration of gaseous substrates in the soil pore space would be
equivalent to the volume fraction of gases in air under completely dry conditions.
We assumed another boundary condition to determine the value of the aqueous diffusion coefficient, $D_{liq}$, such that
soluble substrates like acetate would be available at the enzymatic reaction site under conditions with saturating soil
water content (Davidson et al., 2012):
$D_{liq} = \frac{1}{Porosity^3}$         (11)
We represented soluble substrates (acetate) diffused through a soil water film as $Aqueous - substrate$ ($\mu M\ L^{-1}$),
which we calculated as follows:
$Aqueous - substrate_{av} = Aqueous - substrate \times D_{liq} \times (\frac{SoilM}{100})^3$     (12)
where the $(\frac{SoilM}{100})^3$ term represents the diffusion rate of aqueous substrates to the enzymatic active site (Papendick
and Campbell, 1981). Concentrations of acetate in the aqueous phase ($\mu M\ L^{-1}$) were obtained from the
measurements across the catena averaged by depths (10 and 30 cm) of rhizon samplers.
We calculated $CH_4$ emissions, $CH_{4emission}$ (unit: $\mu mole\ m^{-2}\ hr^{-1}$), as a function of concentration ($[CH_{4conc}]$),
production ($CH_{4prod}$), and oxidation ($CH_{4ox}$) of $CH_4$, multiplied by the equivalent "depth" (set to 15 cm) (for $cm^{-3}$
volume to $cm^{-2}$ area conversion) and $10^4$ (for $m^2$ to $cm^2$ conversion) as follows:
$CH_{4emission} = [CH_{4conc}] + (CH_{4prod} - CH_{4ox}) \times 10^4 \times depth$      (13)
**2.4.3 Soil microsites**
The importance of diverse microsite conditions was inferred based on many previous observations in the field and
the lab of co-occurrences of oxic soil concentrations and reduced redox-active species (Silver et al., 1999; Teh et al.,
2005; Megonigal and Geunther 2008; Hall et al. 2013, 2016; Sihi et al., 2020a). The high clay content, abundant Fe
oxides, and visible redox mottling, particularly in the valley and slope soils facilitates a diversity of soil micro-
environments where $O_2$ and $CH_4$ can seemingly co-occur, albeit in different microsite locations (Silver et al., 1999;
Teh and Silver, 2006). Microsite diversity was also invoked to help explain the rapid $CH_4$ emissions following
drought at the field site (O'Connell et al., 2018). Techniques for accurately measuring in-situ microsite activities
remain very limited to date, here or elsewhere. Therefore, we simulated production, consumption, and diffusion
processes within soil microsites using a log-normal probability distribution function of soil moisture and available C
based on these previously observed relationships (Fig. 1). The average values of individual processes across
simulated microsites (represented by "i") represent the reaction in the bulk soil, which we constrained using the net
measured $CH_4$ emissions.
$$\text{Bulk soil}_{average} = \frac{\sum \text{Frequency}_i \times [\text{microsite}]_i}{\text{Total microsites}} \tag{14}$$
We directly adopted the probability distribution function of soil moisture and C from Sihi et al. (2020a), which
constrained values of $\text{Frequency}_i$ of soil microsites. We a priori assigned the size of the microsites to be at least an
order magnitude smaller than the diameter used for bulk measurements of $CH_4$ fluxes. Thus, the mean diameter of
microsites was assumed to be at the mm-scale (the size-class of small stable aggregates in these soils), as the
diameter of soil chambers was 15.24 cm. Thus, the resultant number of total microsites below each soil flux
chamber was 10,000.
**2.4.4 Sensitivity Analysis**
We evaluated the sensitivity of model parameters with a global variance-based sensitivity analysis using the *R-*
*multisensi* package. This method uses a global sensitivity index ($0 < GSI < 1$) to determine the sensitivity of $CH_4$
emissions to model parameter values (Bidot et al., 2018). We conducted a multivariate technique to estimate GSI
values in sequential steps. First, we implemented a factorial design on the uncertain model parameters, which is
followed by a principal component analysis on model outputs. Then, we extracted GSI values by an ANOVA-based
sensitivity analysis on the first principal component. To that end, parameters with high GSI values may explain high
temporal variations of the observed $CH_4$ emissions and those with low GSI values are insignificant to reproduce the
temporal dynamics of $CH_4$ emissions.
**2.4.5 Statistical Analysis**
We used R (version 3.5.1) for statistical analyses, modeling, and visualization purposes (R Core Team, 2018).
Statistical analyses and figures were produced using *R-ggstatsplot* (Patil, 2018) and *R-ggplot2* (Wickham, 2016)
packages. Differences in soil and porewater chemistry across the catena were compared using robust t-test.
Correlograms for soil temperature, soil moisture, $O_2$, and soil $CH_4$ emissions were created using adjusted Holm
correlation coefficients. All statistical analyses were conducted at the 5% significance level. We implemented the
M3D-DAMM model using *R-FME* package (Soetaert, 2016).

## 3 Results

### 3.1 Observational dynamics of soil biogeochemistry

Soil and porewater chemistry varied along the catena (Fig. 2). Dissolved organic carbon (DOC) values followed the trend of ridge > slope > valley ($p \leq 0.001$). Soil DOC concentrations (mean ± SE) were $0.55 \pm 0.10$, $0.30 \pm 0.03$, and $0.18 \pm 0.03$ mg $g^{-1}$ in ridge, slope, and valley soils, respectively. Organic acid (acetate) concentrations were significantly higher in the ridge ($6.57 \pm 1.48$ µM $L^{-1}$) and slope ($6.42 \pm 2.19$ µM $L^{-1}$) than in the valley ($1.80 \pm 0.20$ µM $L^{-1}$) ($p = 0.003$). Soil pH followed the trend of valley > slope > ridge ($p < 0.001$). Average soil pH ranged from $4.25 \pm 0.11$ in the ridge, to $4.49 \pm 0.08$ in the slope, and to $5.05 \pm 0.09$ in the valley.

Soil moisture and soil $O_2$ concentrations were distinctly different in the drought year (2015) compared to 2016. The drought in 2015 decreased soil moisture in the slope and ridge soils and increased $O_2$ concentrations in the valley soils (Fig. 3) (also see O'Connell et al., 2018). Generally, average soil moisture was higher in the valley ($0.47 \pm 0.05$ in 2015 and $0.51 \pm 0.01$ v $v^{-1}$ in 2016) as compared to the ridge ($0.31 \pm 0.12$ in 2015 and $0.39 \pm 0.03$ v $v^{-1}$ in 2016) and slope ($0.30 \pm 0.16$ in 2015 and $0.41 \pm 0.04$ v $v^{-1}$ in 2016). Average $O_2$ concentrations were generally lower in the valley ($11.54 \pm 5.94$ in 2015 and $6.30 \pm 2.96$ % in 2016) as compared to the ridge ($18.37 \pm 0.72$ in 2015 and $17.52 \pm 0.42$ % in 2016) and slope ($18.09 \pm 1.22$ in 2015 and $16.89 \pm 0.58$ % in 2016). After the drought ended, the recovery of soil moisture in the ridge and slope soils proceeded more quickly than the recovery of $O_2$ concentrations in the valley soils (Fig. 3). Soil temperature ranges were averaged across the topographic gradient and were similar in both years (average was $21.58 \pm 1.88$ in 2015 and $22.97 \pm 1.04$ °C in 2016).

In 2016, net $CH_4$ emissions were generally positive in the valley and were marginally negative in the ridge and slope (Fig. 4). The dynamics of $CH_4$ were very different following the 2015 drought, resulting in net positive $CH_4$ emissions in the post-drought period for all topographic positions (Fig. 3) (as described in more detail in O'Connell et al. 2018). The magnitude of $CH_4$ emissions was greater in the valley, followed by the slope and then the ridge. The strength of the relationships between net $CH_4$ emissions and soil temperature, moisture, and $O_2$ concentrations were contingent on both topographic position and year (2015 vs 2016) (Fig. 5). For example, the relation between $CH_4$ emissions and soil moisture was stronger in 2016 (normal year) than in 2015 (drought year). The correlation between $CH_4$ emissions and $O_2$ concentrations was stronger and more negative in 2015 than 2016. Correlations between soil moisture and $O_2$ concentrations were negative and stronger in 2016 than in 2015. Correlation coefficients between soil $O_2$ concentrations and $CH_4$ emissions were negative and strongest for valley soils and lowest for ridge soils in 2015, but were uncorrelated in 2016 for ridge and slope soils (Fig. S2).

### 3.2 Model simulations of methanogenesis and methanotrophy

In general, there was little bias in the relationships between the observed and simulated $CH_4$ emissions (Fig. 6). The model explained 72% and 67% of the variation in soil $CH_4$ emissions for 2015 and 2016, respectively, although the model performance varied across the catena (Figs. 6, S3, S4). Overall, simulated $CH_4$ emissions captured the trend of valley > slope ≥ ridge for 2016. The model also captured the dramatically different dynamics of field $CH_4$

emissions as a function of topography during and after the 2015 drought. Net positive $CH_4$ emissions were simulated in the drought recovery and post-drought periods in the ridge and slope in 2015, while net negative emissions were simulated in the other times for these landscape positions. Additionally, simulated net $CH_4$ emissions were decreased during the drought and drought recovery in the valley soils, as well as the strong net $CH_4$ emissions in the valley soils in the post-drought period.

The ridge and slope positions were more similar to each other than to the valley soils. Simulated decreased production of acetate and hydrogen during the 2015 drought in the ridge and slope positions resulted in decreased biomass of aceticlastic methanogens and hydrogenotrophic methanogens (Figs. S5, S6). Gross $CH_4$ production therefore decreased during these time periods (Fig. S7). Simultaneously, as soil moisture decreased, simulated methanotrophic biomass increased during the drought (Fig. S5). The simulated biomass of both aceticlastic methanogens and hydrogenotrophic methanogens increased dramatically in the ridge and slope soils during drought recovery (aceticlastic methanogens: 3.3 and 5.3 times higher than drought period for ridge and slope, respectively; hydrogenotrophic methanogens: 6.1 and 12 times higher than drought period for ridge and slope, respectively) and post-drought (aceticlastic methanogens: 5.2 and 8.8 times higher than drought period for ridge and slope, respectively; hydrogenotrophic methanogens: 12 and 24 times higher than drought period for ridge and slope, respectively) period. Concomitantly, production of acetate and $H_2$ was much higher in the ridge and slope soils during the drought recovery (acetate: 1.8 and 2.4 times higher than the drought period for ridge and slope soils, respectively; $H_2$: 3.5 and 6.0 times higher than the drought period for ridge and slope soils, respectively) and the post-drought (acetate: 2.3 and 3.2 times higher than the drought period for ridge and slope, respectively; $H_2$: 5.6 and 10  times higher than the drought period for ridge and slope, respectively) period. Together, gross $CH_4$ production in the ridge and slope soils was significantly higher during the drought recovery (1.9 and 2.5 times higher than the drought period for ridge and slope, respectively) and post-drought periods (3.4 and 4.6 times higher than the drought period for ridge and slope, respectively) compared to the drought (Fig. S7). Simulated production of acetate was increased that also lowered soil pH values during drought recovery (Fig. S6), with a more pronounced effect in the ridge and slope soils. Additionally, simulated methanotrophic biomass and $CH_4$ oxidation decreased during the post-drought period (Figs. S5, S7), which is the same time period during which net $CH_4$ production increased strongly.

For the valley soils, simulated values of aceticlastic methanogens and concomitant acetate production increased during the 2015 drought (Figs. S5, S6). During the drought recovery and post-drought period, both aceticlastic methanogens and acetate production decreased in the valley, while hydrogenotrophic methanogens and $H_2$ production were stable. Gross $CH_4$ production, however, remained relatively flat during the drought event in the valley, and only increased during the post-drought period (Fig. S7). Simulated $CH_4$ oxidation and methanotrophic biomass, on the other hand, increased dramatically during the drought and drought recovery period (Figs. S5, S7), and then decreased strongly during the post-drought period. However, simulated methanotrophic biomass was smaller in the valley soils compared to the ridge and slope soils. Methane oxidation by methanotrophs exerted strong controls on simulated net $CH_4$ emissions, not only in the valley but in all the topographic positions.

**3.3 The influence of microsites on net methane emissions**

Concomitant with decreased soil moisture, the simulated diffusion of gases ($O_2$, $H_2$) was enhanced during the drought event in 2015, while diffusion of the solute (acetate) was dramatically decreased, particularly for the ridge and slope soils (Fig. S8). However, reduction in soil moisture and increase in $O_2$ can inhibit fermentative hydrogen production (Cabrol et al., 2017). Consequently, simulated gross $CH_4$ production through hydrogenotrophic and aceticlastic pathways both decreased during the drought event for the ridge and slope positions (Figs. S7, S9). As soil moisture increased during the drought recovery and post-drought periods, the diffusion of gases decreased, and diffusion of acetate increased in the ridge and slope soils (Fig. S8). Consequently, simulated values of gross $CH_4$ production increased and gross $CH_4$ oxidation decreased during drought recovery and the post-drought period (Fig. S7). These factors likely contribute to the large pulses of net $CH_4$ emissions during the post-drought period for ridge and slope positions (Fig. 3).

Overall, the valley soils were relatively insensitive to changes in the diffusion rate of either gases or solutes (Fig. S8), most likely because soil moisture remained relatively stable, regardless of drought conditions (Fig. 3). The lower sand and higher clay contents in the valley soils (Brenner et al., 2019), as well as the lower topographic position, likely caused the valley soils to remain wetter than the slope and ridge soils. Therefore, simulated values of gross $CH_4$ production were fairly stable in the valley soils (Fig. S7) during the drought and drought recovery period. Simulated production, oxidation, and net flux of $CH_4$ was further modified by reactions occurring within soil microsites. For example, during the drought (~DOY 200 in 2015), gross $CH_4$ production was more frequent in soil microsites in the valley compared to the slope and ridge (Fig. 7). Simulated values of $CH_4$ oxidation were much greater in microsites in the slope and ridge positions, so the net $CH_4$ emissions were positive in the valley soils and negative in the ridge and slope positions. During the 2015 post-drought period (DOY 345), the frequency of $CH_4$ production was much greater in all topographic positions compared to the drought period (DOY 200), and it was also more enhanced in the valley soils compared to the slope and ridge. Thus, net positive $CH_4$ emissions were observed in all topographic positions in the post-drought period (Fig. 3). Methane oxidation at DOY 345 was much greater in the ridge and slope compared to the valley, similar to predictions at DOY 200. Therefore, the prominent $CH_4$ emissions from all three topographic positions were primarily due to increased production ($CH_4$ production on DOY 345 was 150, 248, and 80 % higher than DOY 200 in ridge, slope, and valley, respectively) rather than decreased oxidation ($CH_4$ oxidation was 32, 31, and 43 % lower on DOY 345 than DOY 200 in ridge, slope, and valley, respectively), which agrees with previous studies in our site (Teh et al., 2005, 2008; von Fischer and Hedin, 2002).

Diffusion into microsites strongly affected the concentrations of gases and solutes experienced by microbes, and differences as a function of topographic position were again predicted. Acetate production and diffusion were enhanced in valley soils during the drought, when compared to the slope and ridge soils (Fig. S10). The $H_2$ production was also enhanced in the valley soils during the drought, but the wetter valley soils experienced lower rates of $H_2$ diffusion compared to the ridge and slope soils. Increases in $O_2$ diffusion were also apparent in the ridge and slope soils during the drought, and those increases were greater than in the valley soils. During the post-drought

period, however, the frequency of $H_2$ and $O_2$ diffusion was much greater for the ridge soils compared to the valley
soils (Fig. S10).
Of all parameters, the most sensitive ones were those that controlled $CH_4$ production through the aceticlastic
pathway, followed by the parameters related to $CH_4$ oxidation (Fig. 8). The GSI values for parameters related to
aceticlastic methanogenesis and methanotrophy ranged between 0.25 - 0.75, whereas the corresponding GSI values
for hydrogenotrophic methanogenesis were always $< 0.1$.
**4 Discussion**
**4.1 Mechanisms governing net methane emissions**
Although the initial concentrations of available C for fermentation (i.e. DOC) and substrate for aceticlastic
methanogenesis (i.e. acetate) in the bulk soil followed the trend of ridge > slope > valley (Fig. 2), the pattern of net
$CH_4$ emissions across the catena was opposite (valley > slope $\geq$ ridge), especially in 2016 (Fig. 4). The seemingly
counterintuitive relations of substrate concentrations in the bulk soil versus net $CH_4$ emissions can be explained by
modeling the differing redox conditions across soil microsites. Diffusion promoted the availability of the acetate
substrate through more connected soil water films in the wetter valley soils and caused higher gross $CH_4$ production
in 2016, as compared to the relatively drier slope and ridge soils (Figs. S7, S8). In contrast, diffusion of gaseous
methanotrophic substrates ($CH_4$ and $O_2$) was promoted in the air-filled pore spaces in the drier ridge and slope soils
(Fig. S8), resulting in reduced net $CH_4$ emissions for these two topographic positions in 2016 (Fig. 4). Further,
reduced diffusion of $O_2$ in the wetter valley soils decreased gross methanotrophy compared to the slope and ridge
soils (Figs. S7, S8). Consequently, in 2016, net $CH_4$ emissions dominated the valley soils but were minimal in the
ridge and slope soils.
On the other hand, the drought event in 2015 decreased the simulated $CH_4$ emission in the slope and ridge soils by
decreasing $H_2$ production, and both production (Fig. S6) and diffusion of acetate (Fig. S8). The drought increased
the $CH_4$ sink strength of both ridge and slope soils as the observed net $CH_4$ emissions became more negative during
the drought compared to the pre-drought period (Fig. 3). Contributing factors predicted by the model include
enhanced $O_2$ diffusion into the drier ridge and valley soils (Fig. S8), as well as enhanced methanotrophic biomass
(Fig. S5). In the valley, the primary impact of the drought appeared to be due to increased methanotrophy (Fig. S7),
since acetate, $H_2$, and gross $CH_4$ production were predicted to continue unabated (Fig. S6, S7). This suggests that
drought enhanced consumption of atmospheric $CH_4$ in our site, which is consistent with findings from natural
droughts and throughfall exclusion experiments in other wet tropical forest soils (Aronson et al., 2019; Davidson et
al., 2004, 2008; Wood and Silver, 2012).
However, simulation of observed $CH_4$ emission during drought recovery in 2015 required explicit representations of
the complex interaction of the diffusive supply of solute and gases, dynamics of the microbial functional groups, and
the associated acetate-pH feedback loop across the distribution of soil microsites (Fig. 3). The drought recovery
increased soil moisture which likely prompted anaerobiosis across all topographic locations by significantly
reducing gas diffusivity in a fraction of the simulated microsites (11, 17, and 21 % in ridge, slope, and valley,
respectively) (McNicol and Silver, 2014; Sihi et al., 2020a; Teh et al., 2005). The return to dominantly reducing
conditions also was predicted to stimulate fermentation and the production of acetate through homoacetogenesis
(Fig. S6). Enhanced production and diffusion of acetate during recovery (Fig. S8) triggered growth in the predicted
biomass of aceticlastic methanogens (Fig. S5), which in turn, increased rates of aceticlastic methanogenesis (Fig.
S9).
Simulated rates of hydrogenotrophic methanogenesis also increased in anaerobic microsites (Figs. S9, S10),
mediated by increased production of $H_2$ and subsequent stimulation of the biomass of hydrogenotrophic
methanogens during the drought recovery in 2015 (Fig. S5). Overall, the absolute values of simulated gross $CH_4$
production through hydrogenotrophic and aceticlastic pathways (Fig. S9) outweighed the simulated gross $CH_4$
oxidation rates (Fig. S7), resulting in net soil $CH_4$ emissions across the catena during the post-drought period (Fig.

424 3).

Acetate-driven $CH_4$ increases, decreases in methanotrophy due to decreasing $O_2$, and increasing hydrogenotrophic
methanogenesis all contributed to the post-drought pulses of $CH_4$ (Fig. 8). Both kinds of methanogens increase
during drought recovery and post-drought, but aceticlastic methanogens were two orders of magnitude more
abundant than hydrogenotrophic methanogens. Additionally, acetate may accumulate in microsites during drought,
and then become more available with drought recovery due to enhanced solute diffusion (Fig. S8). The model
simulations suggest that hydrogen diffusion was lessened under the drought recovery which is consistent with
decreasing rates of gas diffusion through saturated soils (Fig. S8). Further, $H_2$ has a faster turnover rate compared to
acetate (Xu et al. 2015) and therefore accumulation in soils, especially shallow soils which are the subject of this
study, is minimized. So, acetate versus hydrogen substrate availability in microsites better explains the observations
of higher $CH_4$ production under the post-drought conditions.
Additionally, acetate is a source of proton and should reduce soil pH (Amaral et al., 1998; Conrad and Klose, 1999;
Jones et al., 2003). Previous studies (Xu et al., 2015; Xu et al., 2010) demonstrated that acetate-driven soil pH
reduction can reduce net $CH_4$ production by as much as 30%, especially in systems with low initial soil pH like our
study site. Given that optimal pH for biological activities peaks near neutral pH, the relatively higher soil pH in the
valley versus ridge and slope soil further enhanced the topographic patterns of $CH_4$ emissions (Conrad, 1996; also
see Figs. 2, 3, and 4). Note that the initial soil pH across the landscape was already in the acidic range (Fig. 2),
consequently, the simulated acetate production and concomitant decrease in soil pH during the 2015 drought
recovery further suppressed gross $CH_4$ production in ridge soils in comparison to the valley soils (Figs. S6 and S7).
Iron reducing bacteria can also suppress $CH_4$ production either by competing with aceticlastic methanogens for
acetate substrate or controlling the flow of acetate to both hydrogenotrophic and aceticlastic methanogens by
dissimilatory iron reduction (Teh et al., 2008). Additionally, Fe reduction can increase soil pH either by proton
consumption and colloid dispersion, while Fe oxidation can lead to more acidic conditions (Hall and Silver, 2013;
Thompson et al., 2006). None of the Fe-associated mechanisms are currently represented in the M3D-DAMM
model.
Hence, high temporal resolution field-scale measurements of $CH_4$ emissions and soil and porewater chemistry
facilitated evaluation of the combined effects of soil redox conditions (moisture and $O_2$ concentrations) and
associated pH feedbacks on underlying processes occurring across soil microsites, while accounting for variation
along the catena as a result of changing climatic drivers over time. The M3D-DAMM model captured the Birch-type
effect by quantifying the pulses in soil $CH_4$ emissions as a function of increases in soil moisture following a strong
drought (Birch, 1958). Specifically, the model coupled with microsite diffusivity explained $CH_4$ emissions common
to wet valley soils and rare in comparatively drier ridge and slope soils and predicted the net release of $CH_4$
emissions from all topographic positions following a strong drought.

**4.2 Sensitivity analysis**

The variance-based sensitivity analysis confirmed the importance of microbial functional groups and their complex
interactions with the surrounding biophysical and chemical environments in controlling $CH_4$ production and
oxidation. For example, the growth and death of aceticlastic methanogens and the relative efficiency of aceticlastic
methanogenesis were the most sensitive parameters (Fig. 8), which is consistent with another modeling effort on
$CH_4$ fluxes across the Arctic landscape (Wang et al., 2019). Although from completely different ecosystem types,
Wang et al. (2019) and the present study confirmed the importance of simulating soil topographies and microbial
mechanisms when evaluating the heterogeneities in $CH_4$ fluxes. Representations of both direct (methanogenic
substrate) and indirect (soil pH feedback) effects of acetate may have contributed to higher GSI values for
parameters representing aceticlastic methanogenesis, which is similar to a previous study (Xu et al., 2015). The
sensitivity of $CH_4$ emissions to the parameters representing methanotrophy were secondary to those representing
aceticlastic methanogenesis, which is consistent with the increase in methanotrophic biomass during the drought.
Our predicted changes in microbial biomass might be unacceptably large for the entire soil microbial community,
which may only double or perhaps quadruple in response to changes in conditions, but individuals can grow
exponentially (Goberna et al. 2010; Pavlov and Ehrenberg 2013; Roussel et al. 2015; Buan 2018).

**4.3 Other processes**

We did not completely reproduce the net emissions of soil $CH_4$ during the 2015 post-drought period across the
catena with the M3D-DAMM model. To capture the full potential of net emissions of $CH_4$ (white shading in Fig. 3)
from sesquioxide-rich soils, future modeling efforts may need to explicitly include the dynamics of redox-sensitive
elements such as Fe and associated pH feedback under contrasting redox conditions (Barcellos et al., 2018;
Bhattacharyya et al., 2018; Hall and Silver, 2013, 2015, 2016; O'Connell et al., 2018; Parfitt et al., 1975; and Silver
et al., 1999). Wetting events can lower soil redox potential and reduce electron acceptors like Fe(III) to Fe(II). This
concomitant reduction of Fe may increase soil pH, especially in anaerobic microsites, which could further increase
net emissions of soil $CH_4$ (Tang et al., 2016; Zheng et al., 2019). Accounting for these effects may allow model
simulations to better match the highest observed net $CH_4$ emissions in the post-drought period (Fig. 3).
Additionally, the reduction of Fe(III) to Fe(II) has supported anaerobic $CH_4$ oxidation in other ecosystems (Ettwig et
al., 2016). Within this context, a measurable amount of anaerobic oxidation of $CH_4$ has previously been reported at
our study site (Blazewicz et al., 2012). Additionally, Fe-reducing microorganisms can utilize acetate as a substrate
and thereby compete with methanogens and reduce net methane emissions (Teh et al., 2008). Given the gradient of
Fe in our study site, it is likely that biogeochemical cycling of Fe and $CH_4$ are coupled (O'Connell et al., 2018)
which should be accounted for in future modeling efforts. For example, a modeling study supported the importance
of Fe in simulating $CH_4$ cycling in an Arctic soil (Tang et al., 2016). To that end, building a comprehensive
framework that also includes Fe biogeochemistry will afford greater confidence in projected $CH_4$ emissions from
wet tropical forests under future climatic conditions (Bonan et al., 2008; Pachauri et al., 2014; Xu et al., 2016).
**5 Conclusions**
High-frequency $CH_4$ emission measurements coupled with real-time soil chemical measurements identified spatial
and temporal variations affecting $CH_4$ production and oxidation in wet tropical forest soils of Puerto Rico. Overall,
contrasting patterns of soil moisture between ridge and valley soils played an instrumental role in governing net $CH_4$
emissions. For example, consistently greater soil moisture likely favored methanogenesis by lowering the
availability of $O_2$ in valley soils compared to ridgetop soils, especially in microsites with high soil moisture and soil
C content. However, soil porewater chemistry, particularly the concentrations of acetate and associated soil pH
influenced the pattern of net emissions of $CH_4$ across the catena (valley > slope > ridge) during wetting after the
2015 drought. Thus, our results provide compelling evidence of the importance of both hot spots and hot moments
in generating and mediating $CH_4$ emissions in wet tropical forest soils. A microbial functional group-based model
coupled with a diffusivity module and consideration of soil microsites adequately reproduced both the spatial and
temporal dynamics of soil $CH_4$ emissions, although mechanisms involving Fe biogeochemistry were neglected.
This study suggests that representing the microbial mechanisms and the interactions of microbial functional groups
with the soil biophysical and chemical environment across soil microsites is critical for modeling $CH_4$ production
and consumption. To that end, explicit consideration of these underlying mechanisms improved predictions of $CH_4$
dynamics in response to regional climatic events and provided insight into differential dynamics of solute and gas
diffusion, different microbial functions, and gross $CH_4$ production and oxidation as a function of topography. Hence,
we contribute to the ongoing development and improvements of Earth system and process models to better simulate
microbial roles in $CH_4$ cycling at regional and global scales. However, observational data concerning the activities
of different soil microbial functional groups is still needed to confirm the mechanisms proposed here. Future studies
should integrate geochemical and microbiological information relevant for oscillatory redox conditions in wet
tropical forests, especially those related to the redox-sensitive elements to build a comprehensive framework for
modeling tropical soil $CH_4$ emissions.
**Code and data availability**
Meteorological data (http://criticalzone.org/luquillo/data/dataset/4723/) are available from the Luquillo CZO
repository. 2015 greenhouse gas fluxes (DOI: 10.6073/pasta/316b68dd254e353e1acfb16d92bac2dc) are available
from the Luquillo LTER repository. The 2016 greenhouse gas fluxes (DOI: 10.15485/1632882), soil chemistry
(DOI: 10.15485/1618870), and rhizon lysimeter data (DOI: 10.15485/1618869) are available from ESS-DIVE
repository. R scripts used for this modeling exercise are archived at the following Zenodo repository (DOI:
10.5281/zenodo.3890562).

## Author contributions

DS performed the data curation of 2016 flux data and the soil and lysimeter data, collated diffusion and microsite processes into the model presented herein, interpreted and validated the model application, developed the visualization, and wrote the original draft. XX provided the model code used in the investigation and assisted with its modification and application. MSO collected the 2016 field flux data. CSO and WLS provided the 2015 flux data and the 2015-2016 field soil measurements for temperature, oxygen, and moisture. CSO developed workflow for field flux data management, cleaning and analysis. WLS acquired the funding, administered the project, and supervised the research team involved with collection of the 2015 data. CLL collected the rhizon water samples and soil samples from the field site, with assistance from MAM. JMB analyzed the rhizon water samples in the lab. JRP, RKQ, and JMB completed the laboratory soil analyses. BDN supplied, installed, and maintained the rhizon water samplers. MAM acquired the funding and administered the project that collected the 2016 data, conceptualized the paper and proposed the methods, supervised the research team, and contributed to the writing, interpretation, and visualization of subsequent drafts. All authors contributed to the manuscript through reviewing and editing subsequent drafts.

## Competing interests

The authors declare that they have no conflicts of interest.

## Acknowledgements

We appreciate the site access and support facilitated by Dr. Grizelle González of the U.S. Department of Agriculture (USDA) Forest Service International Institute of Tropical Forestry and by Dr. Jess Zimmerman of the University of Puerto Rico at Rio Piedras (UPR). We thank Dr. William McDowell of the University of New Hampshire (UNH) for logistical support. We thank Mr. Brian Yudkin, Ms. Jordan Stark, and Ms. Gisela Gonzalez for assistance with field data and sample collection. This work was supported through an Early Career Award to MAM through the U.S. Department of Energy (DOE) Biological and Environmental Research Program; and by grants from the US DOE (TES-DE-FOA-0000749) and the National Science Foundation (NSF) (DEB-1457805) to WLS, as well as the NSF Luquillo Critical Zone Observatory (EAR-0722476) to UNH and the NSF Luquillo Long Term Ecological Research Program (DEB-0620910) to UPR. WLS received additional support from the USDA National Institute of Food and Agriculture, McIntire Stennis project CA-B-ECO-7673-MS. This research used resources of the Compute and Data Environment for Science (CADES) at the Oak Ridge National Laboratory, which is managed by UT-Battelle, LLC, under contract DE-AC05-00OR22725 with the U.S. Department of Energy.

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

**Table 1: Fitted values of M3D-DAMM model parameters.**

| Parameters | Fitted values | Description | Unit | Source |
|---|---|---|---|---|
| $\text{GrowR}_{\text{H2Methanogens}}$ | 0.31 | | 1/day | Servais et al., 1985 |
| $\text{GrowR}_{\text{AceMethanogens}}$ | 1.59 | Growth rates | 1/day | Servais et al., 1985 |
| $\text{GrowR}_{\text{Methanotrophs}}$ | 0.12 | | 1/day | Servais et al., 1985 |
| $\text{DeadR}_{\text{H2Methanogens}}$ | 0.03 | | 1/day | Servais et al., 1985 |
| $\text{DeadR}_{\text{AceMethanogens}}$ | 0.54 | Death rates | 1/day | Servais et al., 1985 |
| $\text{DeadR}_{\text{Methanotrophs}}$ | 0.008 | | 1/day | Servais et al., 1985 |
| $\text{Efficiency}_{\text{H2Methanogens}}$ | 0.2 | | unitless | Grant, 1998 |
| $\text{Efficiency}_{\text{AceMethanogens}}$ | 0.04 | Substrate use efficiencies | unitless | Kettunen et al., 2003 |
| $\text{Efficiency}_{\text{Methanotrophs}}$ | 0.4 | | unitless | Kettunen et al., 2003 |
| $\text{KM}_{\text{Ace}}$ | 16 | | $mmol/m^3$ | Grant, 1998; McGill et al., 1981 |
| $\text{KM}_{\text{H2ProdAce}}$ | 11 | | $\mu mol/m^3$ | Conrad, 1989 |
| $\text{KM}_{\text{H2ProdCH4}}$ | $2.14 \ast 10^{-5}$ | | $mmol/m^3$ | Fennel and Gossett, 1998 |
| $\text{KM}_{\text{CO2ProdCH4}}$ | $9.08 \ast 10^{-9}$ | Half-saturation | $mmol/m^3$ | Stoichiometry theory |
| $\text{KM}_{\text{CH4ProdAce}}$ | 13 | constants | $mmol/m^3$ | Kettunen et al., 2003 |
| $\text{KM}_{\text{CH4ProdO2}}$ | 0.03 | | $mmol/m^3$ | Kettunen et al., 2003 |
| $\text{KM}_{\text{CH4OxidCH4}}$ | 0.06 | | mmol/l | Kettunen et al., 2003 |
| $\text{KM}_{\text{CH4OxidO2}}$ | 0.74 | | mmol/l | Kettunen et al., 2003 |
| $\text{ACmax}_{\text{AceProd}}$ | 0.52 | | $mmol/m^3/h$ | Smith and Mah, 1966 |
| $\text{Acemax}_{\text{H2Prod}}$ | 1.31 | Maximum reaction rates | mmol acetate/g/h | Conrad, 1989 |
| rCH4Prod | 0.84 | | mol $CH_4$/mol acetate | Kettunen et al., 2003 |
| rCH4Oxid | 3.06 | Rate constants | mol $O_2$/mol $CH_4$ | Kettunen et al., 2003 |
| $\text{Q}_{\text{10ACMin}}$ | 1.16 | | unitless | Segers, 1998 |
| $\text{Q}_{\text{10AceProd}}$ | 1.21 | Temperature sensitivities | unitless | Atlas and Bartha, 1987; Kettunen, 2003; van Hulzen et al., 1999 |
| $\text{Q}_{\text{10H2CH4Prod}}$ | 1.27 | | unitless | Segers, 1998 |
| $\text{Q}_{\text{10CH4Prod}}$ | 1.13 | | unitless | Kettunen et al., 2003 |

| $Q_{10CH4Oxid}$ | 1.18 | unitless | Kettunen et al., 2003 |

Initial values of model parameters were collected from literature ("Source"). Also see Xu et al. (2015) for detailed information on model parameters.

Figure Captions:

**Figure 1: Conceptual figure of the modelling approach. Top panel (a) Top panel (a) shows the model representation of soil microsite distribution (modified from Sihi et al., 2020a, also see Eq. 14). The cylinder refers to the volume beneath the soil chambers. The intensity of different cylinder colors figure refers to rate of a process or the intensity of a concentration inside microsites in each theoretical cylinder, e.g., a dark color means a higher rate/intensity, and a light color means a lower rate/intensity for a given process. The 2D graph on the right refers to the probability density function of the rate of the process or intensity of the concentration in the bulk soil. A wide distribution skewed to the right (dark line) implies higher bulk rates of the process or higher concentrations, and a narrow distribution skewed to the left (light line) implies lower bulk rates of the process or lower concentrations, of any of the following: solute concentration [$S_i$], gas concentration [$G_i$], soil moisture ($SoilM_i$), gas and solute diffusion ($Diff_i$), methane production ($Prod_i$), and methane oxidation ($Ox_i$). Bottom panel (b) is the schematic of the microbial functional group-based model for simulating soil methane ($CH_4$) dynamics in field soils (modified from Xu et al., 2015). The schematic represents the decomposition of soil organic matter (SOM) and plant litter into carbon dioxide ($CO_2$) and dissolved organic matter (DOC); the production of acetate and hydronium ion ($H^+$) from decomposition and fermentation of DOC which also decreases pH, the production of acetate and hydronium ion ($H^+$) from homoacetogenesis which decreases pH; and the production of dihydrogen ion ($H_2$) and $CO_2$ from decomposition of DOC. The intermediary products then have three possible non-mutually exclusive pathways (1) aceticlastic methanogenesis, which is the production of methane from aqueous acetate found in soil solutions, (2) hydrogenotrophic methanogenesis, which is the production of methane from hydrogen, and (3) methanotrophy, which is the oxidation of methane into carbon dioxide.**

**Figure 2: Soil and porewater chemistry (dissolved organic carbon [DOC] (a), acetate (b), and pH (c)) along the ridge-slope-valley topographic gradient.**

**Figure 3: Temporal dynamics of observed meteorological drivers (soil temperature (a-c), soil moisture (d-f), soil oxygen (g-i)) and net methane emissions (j-l) for 2015 (Data are taken from O'Connell et al., 2018). For methane emissions, symbols represent observed data and lines represent model simulations. Dark gray, medium gray, light gray, and white shading represent pre-drought, drought, drought recovery, and post drought events (O'Connell et al., 2018).**

**Figure 4: Temporal dynamics of observed meteorological drivers (soil temperature (a-c), soil moisture (d-f), soil oxygen (g-i)) and net methane emissions (j-l) for 2016. For methane emissions, symbols represent observed data and lines represent model simulations.**

**Figure 5: Relation between soil meteorology and methane emissions for 2015 (a) and 2016 (b). SoilM, SoilT, $O_2$, $CH_4$ represent soil moisture, soil temperature, oxygen, and methane, respectively. Numbers represent adjusted Holm correlation coefficients, and numbers with "X" indicate a non-significant correlation at $p < 0.05$.**

**Figure 6: Observed versus simulated methane ($CH_4$) emissions and model residuals for 2015 (a, b) and 2016 (c, d).**

**Figure 7: Rates of gross methane ($CH_4$) production (a, b), oxidation (c, d), and net flux (e, f) across simulated soil microsites. Day of year 200 and 345 represent drought and post-drought recovery, respectively (see medium gray and white shading in Fig. 3).**

**Figure 8: Global sensitivity indices of M3D-DAMM model parameters (defined in Table 1). Gray, yellow, and blue colors represent parameters for aceticlastic methanogenesis, hydrogenotrophic methanogenesis, and methanotrophy, respectively.**

816

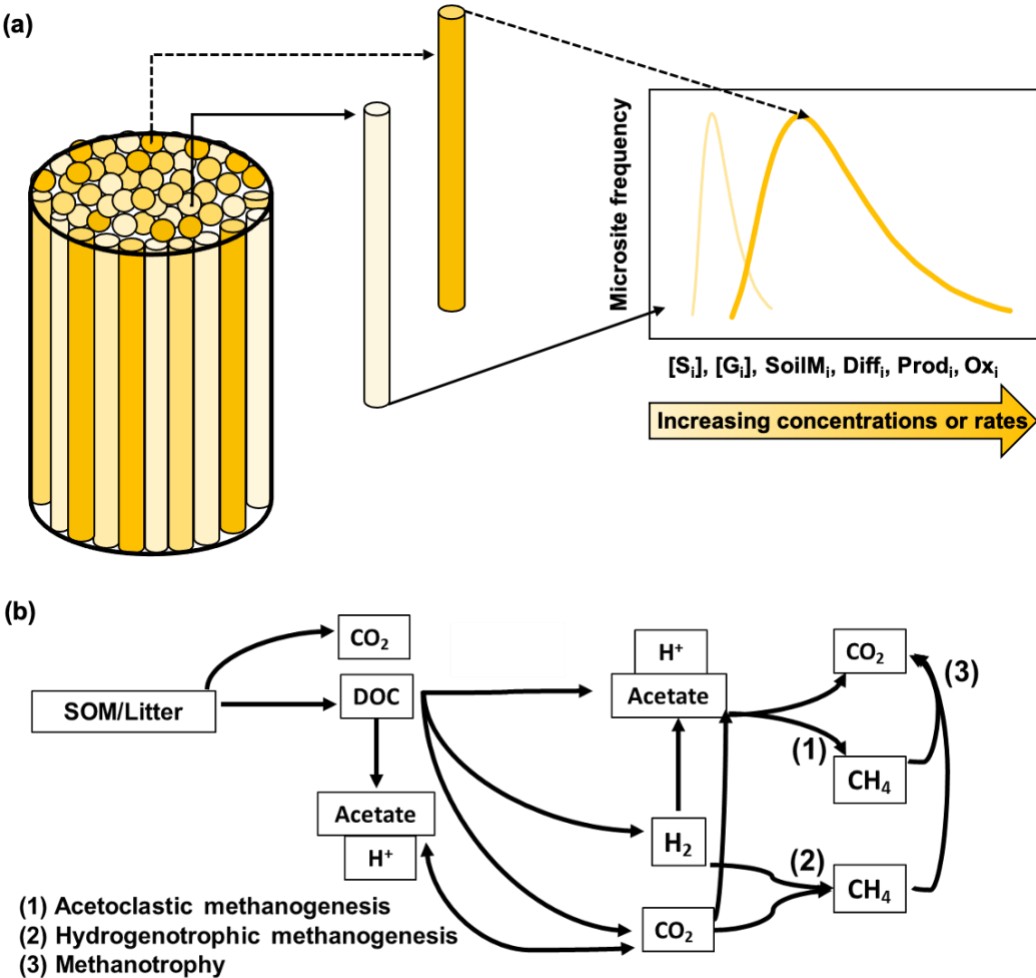

817

**Figure 1: Conceptual figure of the modelling approach. Top panel (a) Top panel (a) shows the model representation of soil microsite distribution (modified from Sihi et al., 2020a, also see Eq. 14). The cylinder refers to the volume beneath the soil chambers. The intensity of different cylinder colors figure refers to rate of a process or the intensity of a concentration inside microsites in each theoretical cylinder, e.g., a dark color means a higher rate/intensity, and a light color means a lower rate/intensity for a given process. The 2D graph on the right refers to the probability density function of the rate of the process or intensity of the concentration in the bulk soil. A wide distribution skewed to the right (dark line) implies higher bulk rates of the process or higher concentrations, and a narrow distribution skewed to the left (light line) implies lower bulk rates of the process or lower concentrations, of any of the following: solute concentration [$S_i$], gas concentration [$G_i$], soil moisture (SoilM$_i$), gas and solute diffusion (Diff$_i$), methane production (Prod$_i$), and methane oxidation (Ox$_i$). Bottom panel (b) is the schematic of the microbial functional group-based model for simulating soil methane (CH$_4$) dynamics in field soils (modified from Xu et al., 2015). The schematic represents the decomposition of soil organic matter (SOM) and plant litter into carbon dioxide (CO$_2$) and dissolved organic matter (DOC); the production of acetate and hydronium ion (H$^+$) from decomposition and fermentation of DOC which also decreases pH, the production of acetate and hydronium ion (H$^+$) from homoacetogenesis which decreases pH; and the production of dihydrogen ion (H$_2$) and CO$_2$ from decomposition of DOC. The intermediary products then have three possible non-mutually exclusive pathways (1) aceticlastic methanogenesis, which is the production of methane from aqueous acetate found in soil solutions, (2) hydrogenotrophic methanogenesis, which is the production of methane from hydrogen, and (3) methanotrophy, which is the oxidation of methane into carbon dioxide.**


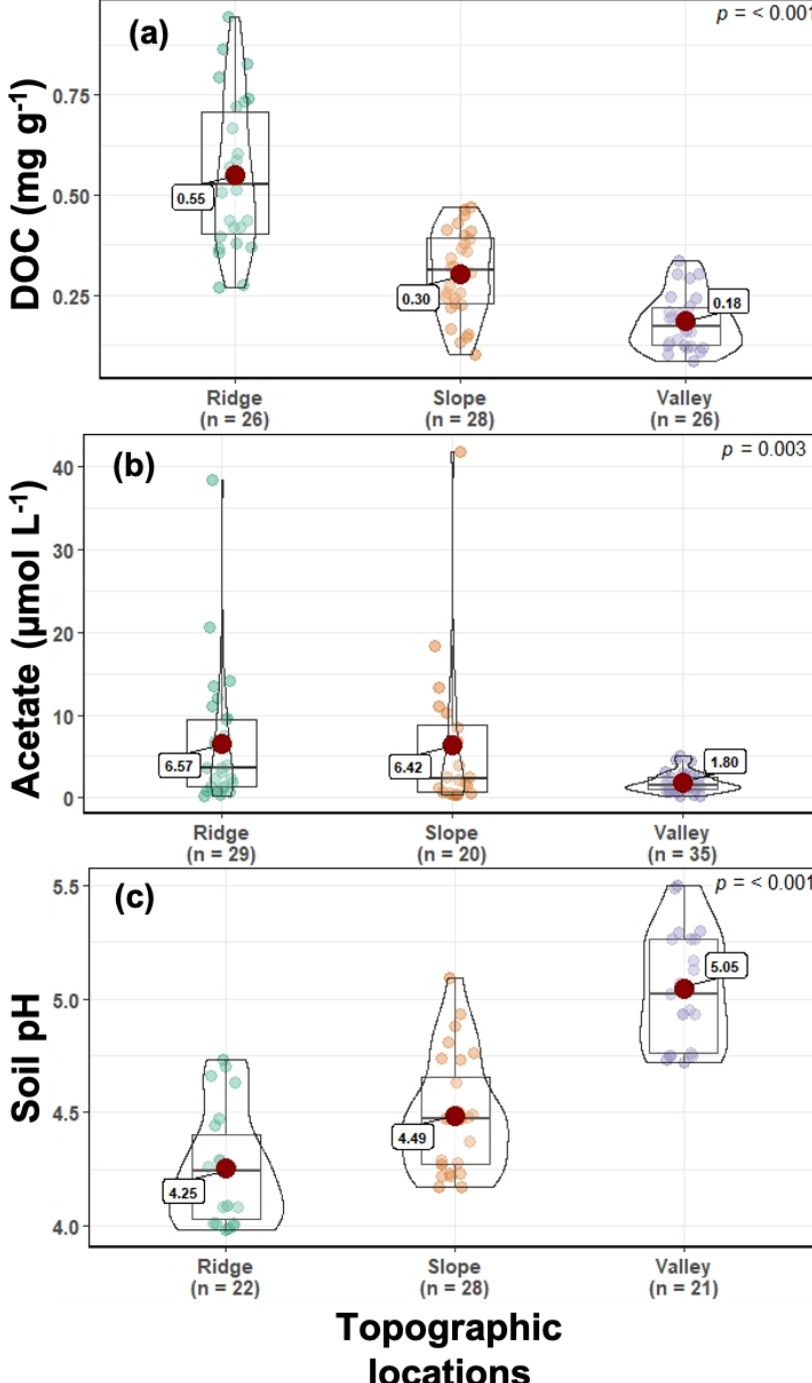

**Figure 2: Soil and porewater chemistry (dissolved organic carbon [DOC] (a), acetate (b), and pH (c)) along the ridge-slope-**
**valley topographic gradient.**

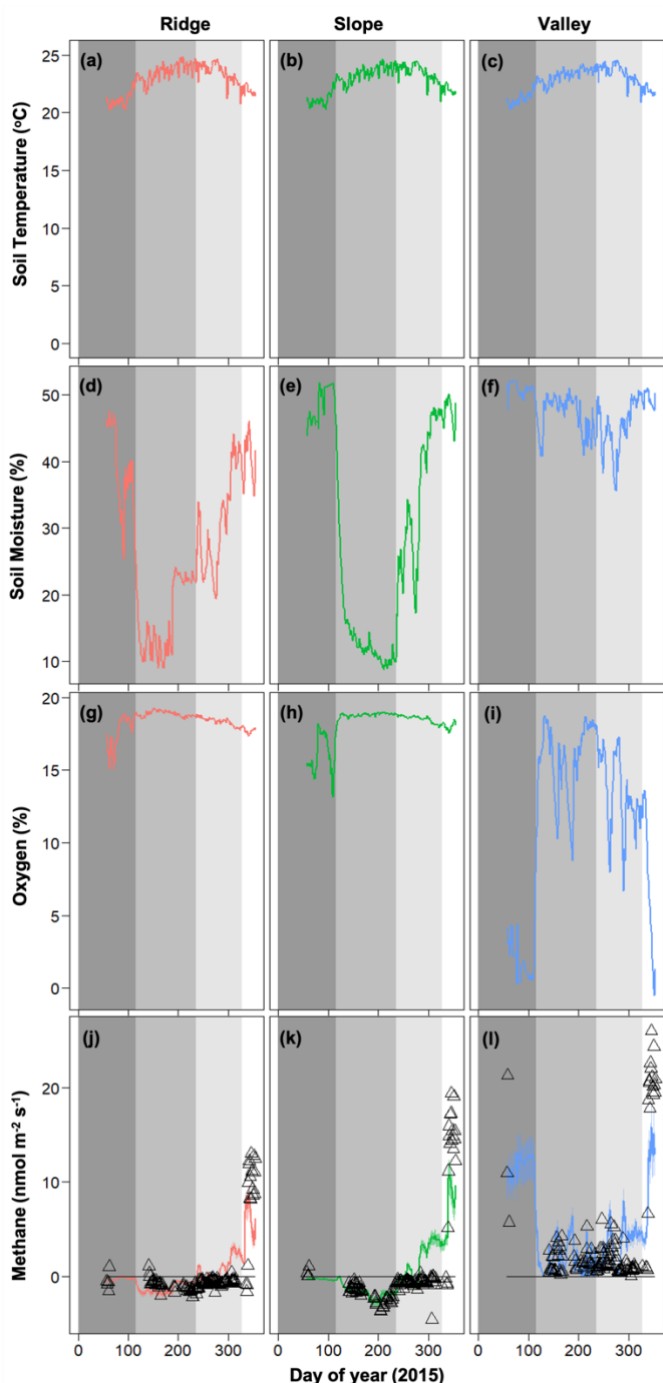

**Figure 3: Temporal dynamics of observed meteorological drivers (soil temperature (a-c), soil moisture (d-f), soil oxygen**
**(g-i)) and net methane emissions (j-l) for 2015 (Data are taken from O'Connell et al., 2018). For methane emissions,**
**symbols represent observed data and lines represent model simulations. Dark gray, medium gray, light gray, and white**
**shading represent pre-drought, drought, drought recovery, and post drought events (O'Connell et al., 2018).**

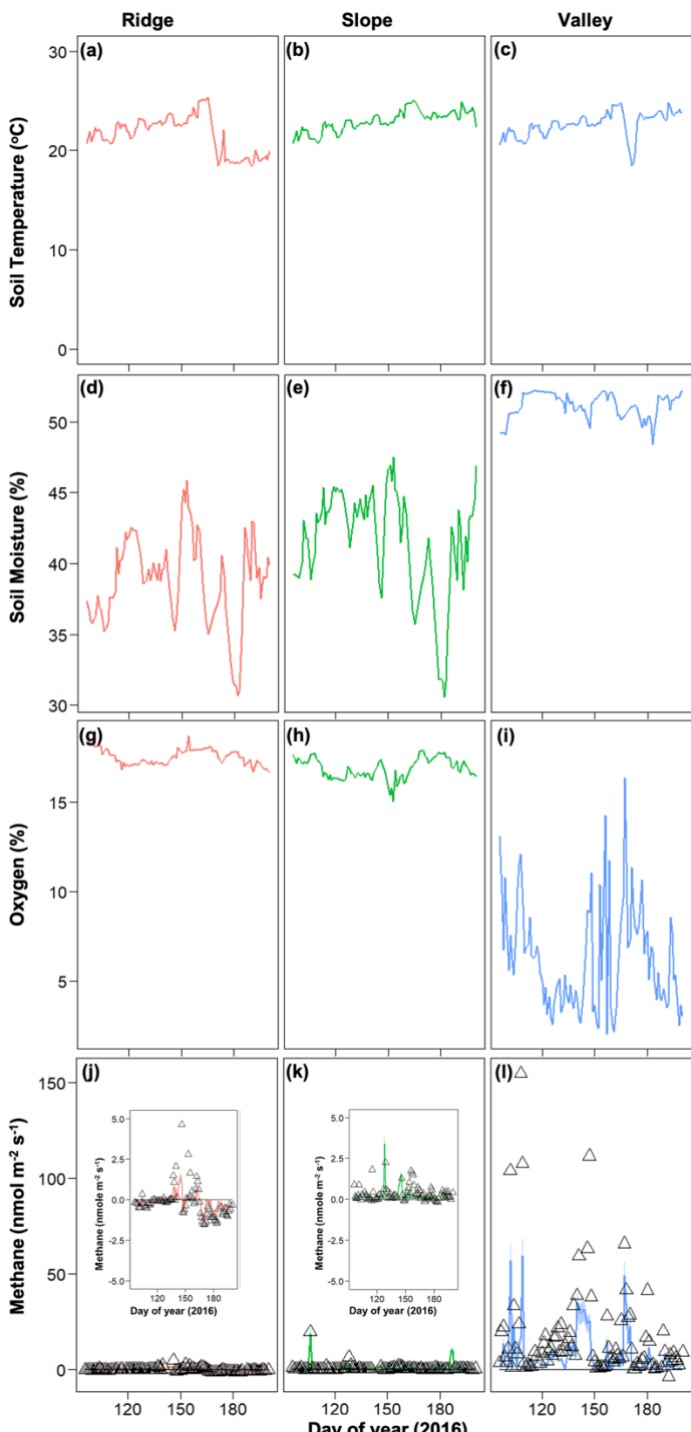

**Figure 4: Temporal dynamics of observed meteorological drivers (soil temperature (a-c), soil moisture (d-f), soil oxygen (g-**
**i)) and net methane emissions (j-l) for 2016. For methane emissions, symbols represent observed data and lines represent**
**model simulations.**

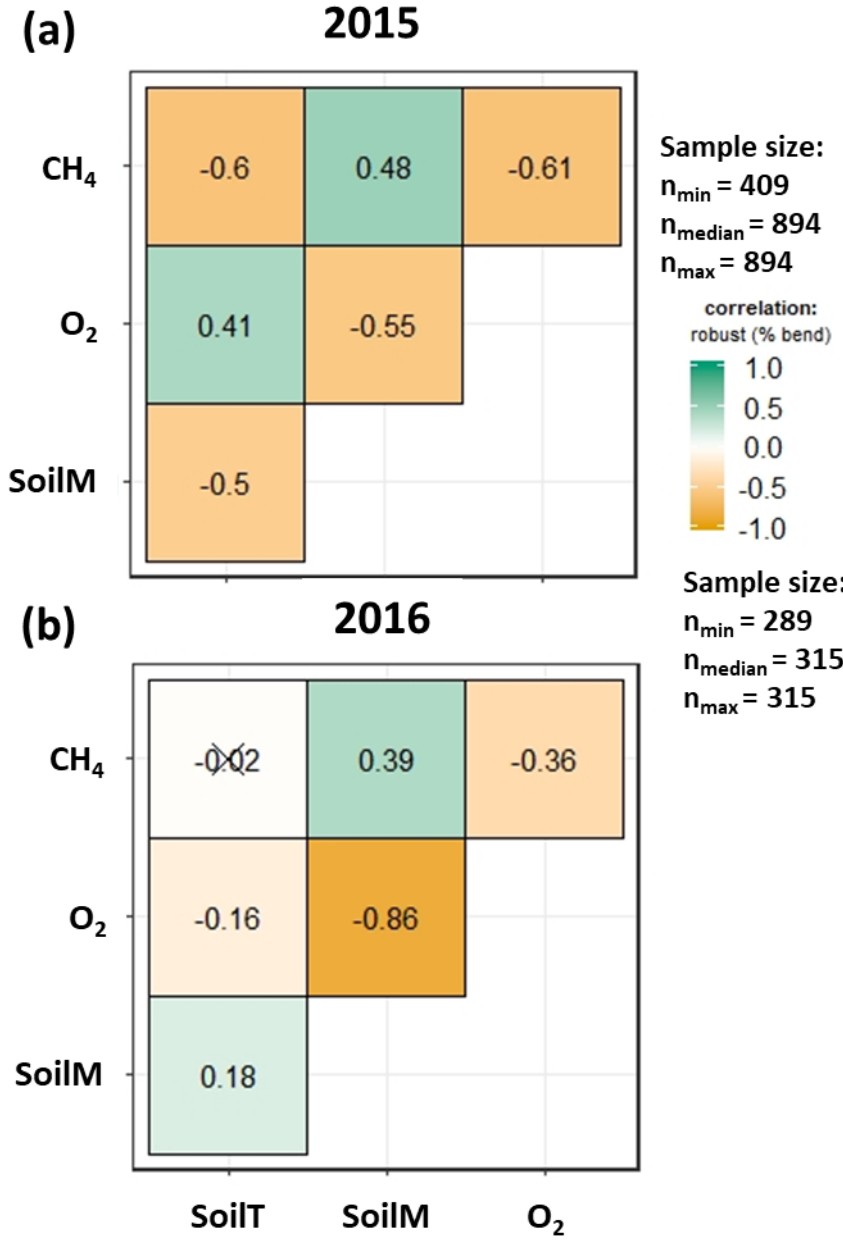

**Figure 5: Relation between soil meteorology and methane emissions for 2015 (a) and 2016 (b). SoilM, SoilT, $O_2$, $CH_4$**
**represent soil moisture, soil temperature, oxygen, and methane, respectively. Numbers represent adjusted Holm**
**correlation coefficients, and numbers with "X" indicate a non-significant correlation at $p < 0.05$.**

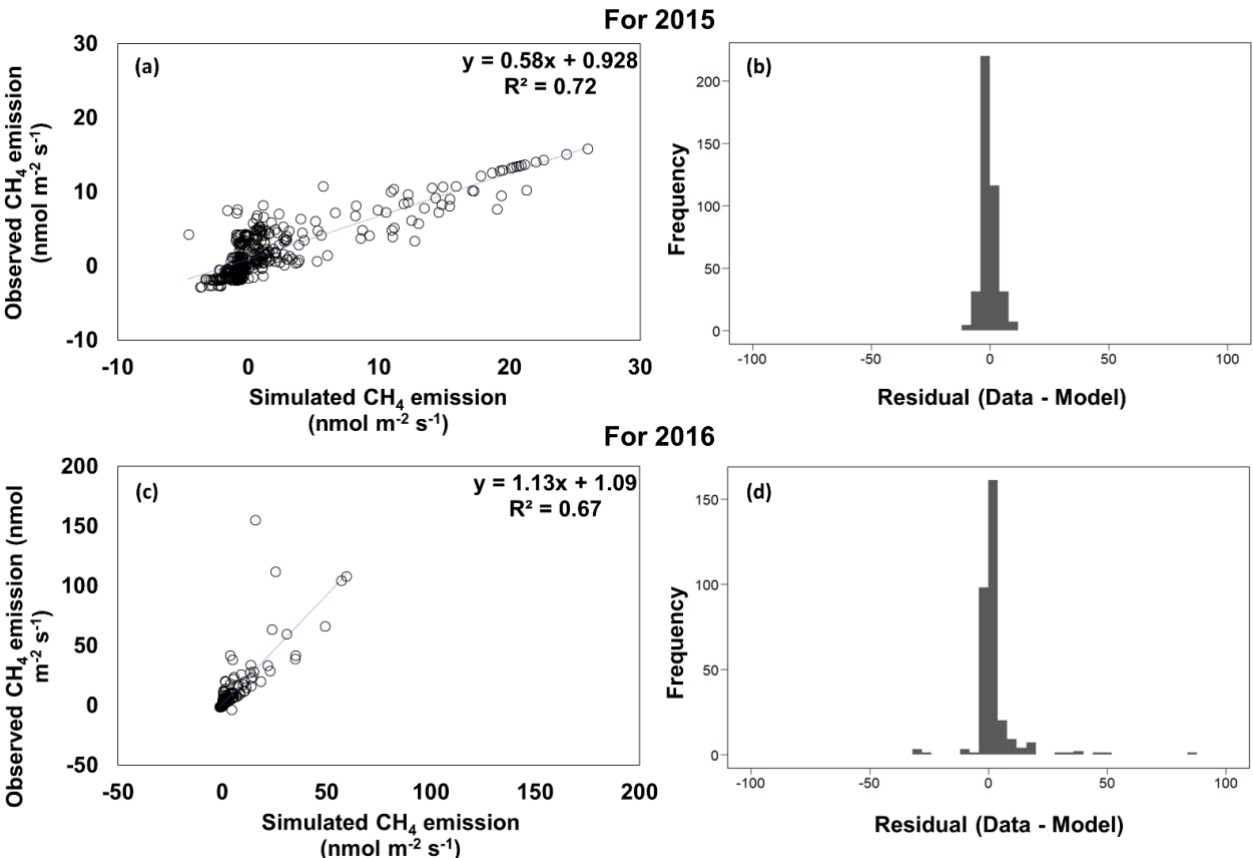

853
854
855

**Figure 6: Observed versus simulated methane (CH4) emissions and model residuals for 2015 (a, b) and 2016 (c, d).**

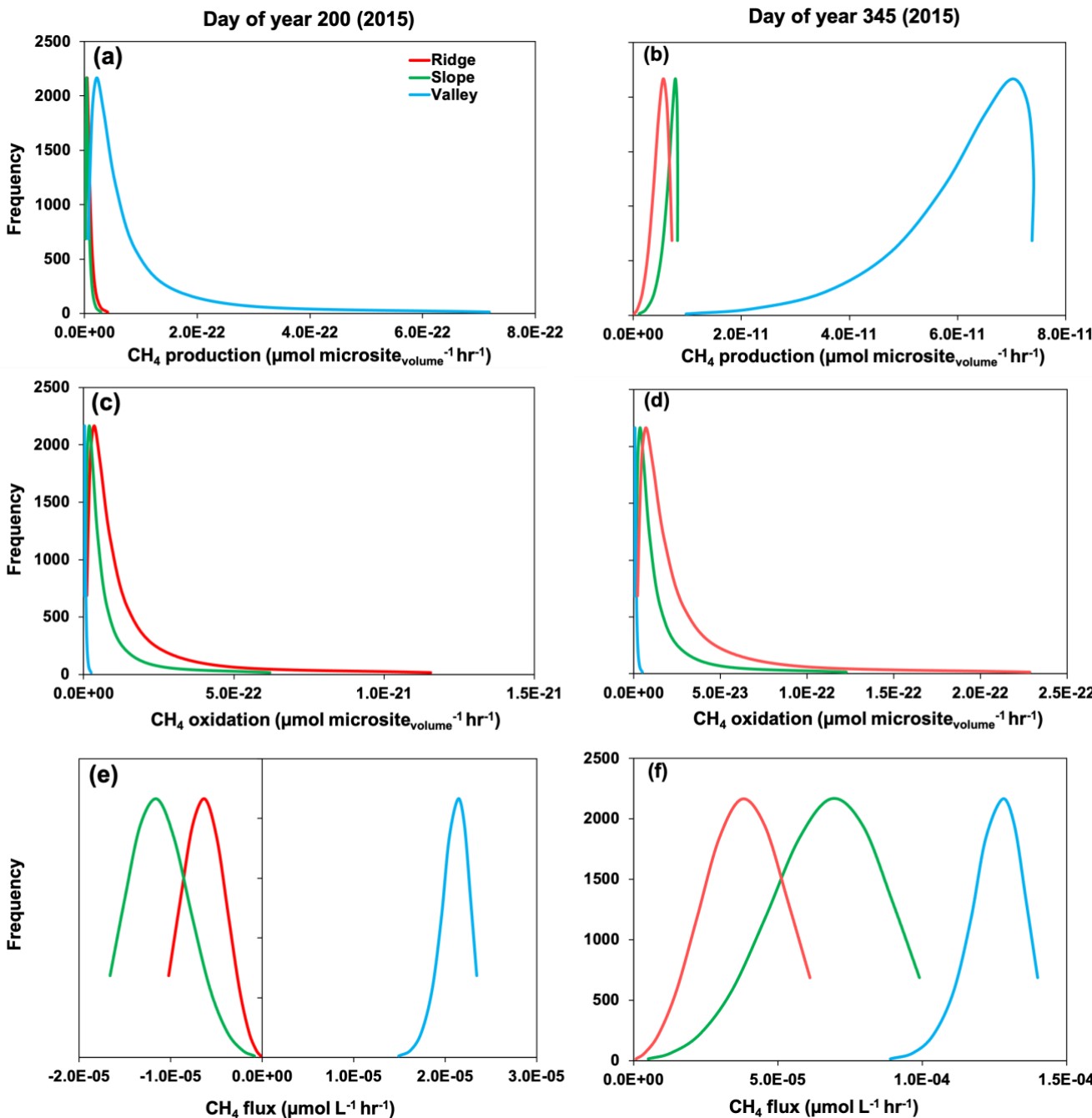

Figure 7: Rates of gross methane ($CH_4$) production (a, b), oxidation (c, d), and net flux (e, f) across simulated soil microsites. Day of year 200 and 345 represent drought and post-drought recovery, respectively (see medium gray and white shading in Fig. 3).

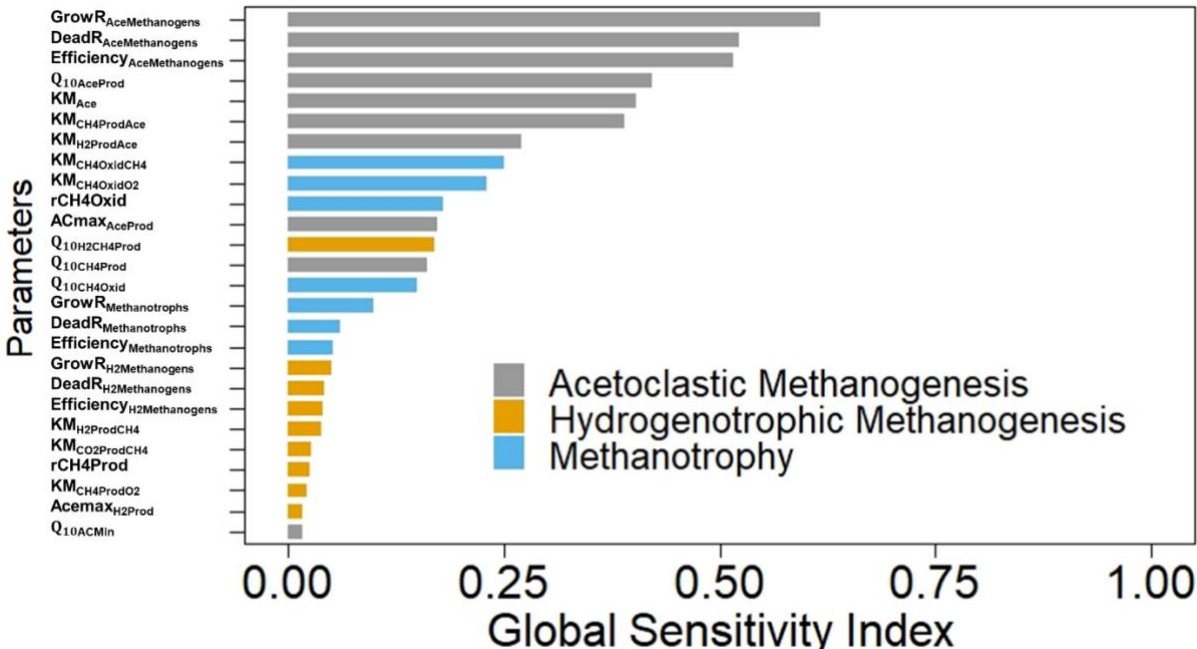

Figure 8: Global sensitivity indices of M3D-DAMM model parameters (defined in Table 1). Gray, yellow, and blue colors represent parameters for aceticlastic methanogenesis, hydrogenotrophic methanogenesis, and methanotrophy, respectively.