# Peer review of "Representing methane emissions from wet tropical forest soils"

_Biogeosciences, 2020_

## Referee Comment (RC1) · Anonymous Referee #1 · 22 Aug 2020

The study "Representing methane emissions from wet tropical forest soils using microbial functional groups constrained by soil diffusivity" by Sihi et al. tries to explain soil methane emission dynamics in tropical forest soils of Puerto Rico during normal and drought conditions. They combine field measurements with modelling efforts (Microbial Model for Methane Dynamics-Dual Arrhenius and Michaelis Menten (M3D-DAMM). Overall, I think it is a really nice study that tries to combine microbial with biogeochemical data to investigate ecosystem methane dynamics. However, I have some general and some minor comments.

The authors should describe the concept of "microsites" more in detail. The au-

thors focused on the top 10 or 10-30 cm of their Ultisol although methane production/consumption dynamics in the deeper clay-accumulating horizons may be more important for the overall net methane emissions from their Ultisols. The authors do not discuss that and do not compare with other soils. How does the abundance of microsites change with soil type, soil depth and other ancillary variables?

Minor comments: L22-25: Is it important to give this information? I would only include the most significant ones that support your guiding questions! What is the difference between "<" and "«". The abstract should self-explanatory.

L33: write aceticlastic methanogenesis instead of "acetotrophic" and "acetoclastic" (check the whole manuscript)

L43: what are wet tropical forest soils? What are wet tropical forests and what is the difference between wet tropical forest soils and upland soils? How do you define that?

L53-65: What are the soil types in the different studies? Since methanogenesis and methanotrophy are substrate-limited, the soil type with its specific biogeochemistry is very important. The authors mention that there are several studies that report effects of drought on net methane emissions across different wet tropical forest soils. Consequently, they should mention the different soil types.

L66-74: Oxygen may not be the only factor for methane emissions upon rewetting. The observed rapid flush of methane in response to a wetting event may be driven by rapid depletion of other electron acceptors, as well. The major focus of this paragraph is on oxygen but what is with acetate, $H_2$ and $CO_2$?

L85-86: and why not $H_2$ and $CO_2$? How do you account for acetate formation during fermentation and homoacetogenesis? How do you account for syntrophic acetate oxidation? How can you explain the "contrasting patterns of observed $CH_4$ emissions", when sources and sinks of acetate etc. is not measured or simply not known?

L97-105: The dominant soil type in the current study is Ultisol. I guess the whole

methane cycling will be very different in other tropical soils (What is with Oxisols?) What are the soil types in the other studies mentioned in the introduction? It would be great to see a more detailed description of the soil type. How did bulk density change with depth? At what depth does clay accumulation (subsurface zone) start? How big is the eluvial horizon?

L107: Why was soil only sampled from the top 10 cm of the soil? I guess methane consumption dominates at the surface due to oxic conditions while in the deeper soil horizons methane production dominates due to more anoxic conditions.

L111-114: What was the detection limit for acetate?

L117-120: The chemical data (what chemical data? Only pH?) for bulk soil is from the 0-10 cm soil depth and acetate and DOC from the pore water from 10-30 cm soil depth? I do not understand how you relate this information, taken from different soil depths, to each other. Finally, where do you find or where do you assume these microsites? (only in the top 10 cm or below? Or may be even below 30 cm?

L144-148: What is with methylotrophic methanogenesis. You should discuss about the potential contribution of methylotrophic methanogenesis (see Norrow et al. 2019).

L173-174: Why not?

L206: Why 15 cm?

L214: How do calculate "total microsites"? I would assume that there are way more microsites in clayey horizons below 15 or even 30 cm soil depth? Overall, I think you have to explain the concept of microsites, more in detail? You are scratching only the soil surface at the moment but in my opinion the biggest methane production potential occurs in deeper parts of the Ultisol.

L270-272: Why are acetate and hydrogen production decreasing when aceticlastic and hydrogenotrophic methanogenesis also decrease? Aceticlastic and hydrogenotrophic methanogens consume acetate and hydrogen, respectively. So, if there is a decrease

in aceticlastic methanogens, I would first assume an increase in acetate concentration and thereafter a sharp decrease if oxygen levels further increase.

L280: How do explain that?

L286-287: Why does the increasing production of acetate lowers the pH?

L301-306: If the diffusion of $H_2$ increases during drought, one may think that hydrogenotrophic methanogenesis should increase as well. However, it does not increase because of increasing oxygen levels. That should be made clear!

L304-309: The diffusion of acetate increases upon rewetting but that of $H_2$ decreases. Why do you observe an increase in overall gross methane production. First, I would assume that under relatively acid conditions, hydrogenotrophic methanogenesis dominates. I think the overall increase of methane emissions upon rewetting is because of oxygen depletion and therefore the stimulation of methanogenesis in general ... and not because of increasing acetate concentrations or a shift in the methanogenic pathway of methane formation. If it is really aceticlastic methanogenesis that is stimulated, you should provide some isotopic data. There is competition for acetate between several microorganisms. In the end it could be simply stimulation or inhibition of fermentation or homoacetogenesis that drives changes in the amount of observed acetate concentrations.

L318: Again, how do you define the microsites?

L346-348: Again, what makes you so sure that it is acetate that drives net methane emissions and not $H_2/CO_2$ and a decrease in oxygen?

L369-372: and homoacetogenesis?

---

## Referee Comment (RC2) · Anonymous Referee #3 · 26 Aug 2020

The manuscript of Debjani Sihi and colleagues brings up a very interesting topic on disentangling gross methane emission and uptake from wet tropical forest soil using a combination of microbial functional group $CH_4$ model and a diffusivity module. This work clearly shows how landscape topography and climate affect net $CH_4$ emissions due to shift of substrate production, soil redox conditions, and diffusivity of $O_2$, $H_2$, and acetate under drought and recovery phases. The experimental work is well performed, convincing and well discussed in the context of previous literature. The manuscript is well organized and clearly written and I enjoyed reading it. I only have a few comments that should be addressed: Line 54 Should it be "increased consumption of atmospheric $CH_4$"? Line258 The correlation seems stronger and more negative in 2015 (-0.36)

than 2016 (-0.61). Line 321-322 You defined pre-drought period from DOY 57-115 instead of DOY 200. The details in results should be checked. Fig. 1 I appreciate the conceptual figures herein, but it looks a bit confusion and I do not well understand what means in panel a. How to relate microsite frequency with soil properties? Do substrate concentration, soil moisture, diffusivity of solute and gas present the similar pattern for one kind of microsite? Why this figure links to Eq.13? Also in panel b, it would be more clear for readers if you could adjust it to a better shape or based on the clue of the present study. Can you try to improve the conceptual figure and clarify this in the legend? Fig.3 and 4 The label of y-axis for soil moisture and oxygen should between 0-1 rather than 0-100, as the unit is V V-1. Otherwise, the unit should change to %. Fig.4 and 6 The unit of CH4 emission should be uniform. Some of them are nmol m-2 S-1, while others are nmole m-2 S-1. Also the unite of acetate (Fig. 2).

---

## Author Comment (AC1) · 2 Oct 2020

C: The study "Representing methane emissions from wet tropical forest soils using micro- bial functional groups constrained by soil diffusivity" by Sihi et al. tries to explain soil methane emission dynamics in tropical forest soils of Puerto Rico during normal and drought conditions. They combine field measurements with modelling efforts (Micro- bial Model for Methane Dynamics-Dual Arrhenius and Michaelis Menten (M3D-DAMM). Overall, I think it is a really nice study that tries to combine microbial with biogeochem- ical data to investigate ecosystem methane dynamics. However, I

have some general and some minor comments. R: Thank you kindly for the positive comments and for the constructive suggestions.

C: The authors should describe the concept of "microsites" more in detail. The authors focused on the top 10 or 10-30 cm of their Ultisol although methane production/consumption dynamics in the deeper clay-accumulating horizons may be more important for the overall net methane emissions from their Ultisols. The authors do not discuss that and do not compare with other soils. How does the abundance of microsites change with soil type, soil depth and other ancillary variables? R: Our sampling strategy, both of soils and of soil water, are geared to accompany the greenhouse gas flux measurements which are taken on the soil surface. Past studies (Silver et al. 1999) have taken methane concentrations at depth in similar soils, and found that concentrations are higher at 10 versus 35 cm. Examination of the soils in the Luquillo mountains have found that SOC maximums are around 35 cm depth (Johnson et al. 2014). For this site, clay is abundant at all depths, e.g., 20-30% clay at 0 to 10 cm depth (L101). Therefore, we believe that our sampling strategy is appropriate for the mechanisms we are trying to address. We will add this information and the Johnson citation to a revision to ensure readers also understand this concurrence of sampling strategy and past observations.

We will address the issue of comparison to other soils in response to a comment below.

There are no specific measurements of microsites at any depth at this site; microsites are inferred because of decades of observations of co-occurrences of oxygen concentrations in the soil and methane fluxes; and because of the rich clay, iron oxides, and visible redox mottling, particularly evident in the valley and slope soils (papers cited in L66-74). Techniques for measuring microsite activities remain very limited to date, here or elsewhere. We will add this note to this effect in a revision.

C: Minor comments: L22-25: Is it important to give this information? I would only include the most significant ones that support your guiding questions! What is the

difference between "<" and "Âń". The abstract should self-explanatory. R: We will simplify the abstract and the manuscript throughout by using only <, =, and >; it is perhaps a subjective difference between "<" and "«". The geochemical parameters in L22-25 were measured and are key model inputs, so we will rephrase to say they were measured at the field site, but we will remove the parenthetical material regarding differences in their values with respect to topographic positions which is not germane to the abstract.

C: L33: write aceticlastic methanogenesis instead of "acetotrophic" and "acetoclastic" (check the whole manuscript) R: Thank you for catching these typos; we will search the entire manuscript to ensure they are always correct. I am confused as to whether it is "aceticlastic" or "acetoclastic" methanogenesis, as I see both in the literature. We will clarify which is appropriate, perhaps with Biogeosciences Editorial Staff, and use it consistently in the revision.

C: L43: what are wet tropical forest soils? What are wet tropical forests and what is the difference between wet tropical forest soils and upland soils? How do you define that? R: The soils are classified as wet tropical forest soils according to Holdridge life zones, which considers rainfall, elevation, latitude, humidity, and evapotranspiration, as described specifically regarding the Luquillo Experimental Forest (Harris et al. 2012). Upland refers only to topography. The soils in our current study (including the valley soils) could be referred to as "upland" in that they are all located in a lower montane region (∼350 m elevation). We don't believe any changes to MS are warranted in response to this comment.

C: L53-65: What are the soil types in the different studies? Since methanogenesis and methanotrophy are substrate-limited, the soil type with its specific biogeochemistry is very important. The authors mention that there are several studies that report effects of drought on net methane emissions across different wet tropical forest soils. Consequently, they should mention the different soil types. R: We will add the soil type (below) when the studies are mentioned in L53-65 and elsewhere as relevant. Aronson et al.

2019 (Costa Rica) Oxisols Davidson et al. 2004, 2008 (Brazil) Oxisols Wood and Silver 2012 (Puerto Rico) Ultisols with a similar climate and parent material O'Connell et al. 2018 (Puerto Rico) Ultisols, same soils as current study

C: L66-74: Oxygen may not be the only factor for methane emissions upon rewetting. The observed rapid flush of methane in response to a wetting event may be driven by rapid depletion of other electron acceptors, as well. The major focus of this paragraph is on oxygen but what is with acetate, H2 and CO2? R: We concur with the reviewer's suggestions; in fact, that is part of why we initiated this study. We wanted additional data on acetate concentrations, and to use a predictive model to better understand the conflicting/collaborating roles of oxygen, substrate, and microbial biomass in controlling methane emissions. In a revision, we will clarify the role of these different substrates in controlling methane emissions in this paragraph so it does not appear we are only focused on oxygen. Although they are important, acetate and particularly H2 measurements are seldom available.

C: L85-86: and why not H2 and CO2? How do you account for acetate formation during fermentation and homoacetogenesis? How do you account for syntrophic acetate oxidation? How can you explain the "contrasting patterns of observed CH4 emissions", when sources and sinks of acetate etc. is not measured or simply not known? R: The idea of this paper is to consider a relatively simple set of mechanisms and see how well these mechanisms can explain the complex observations. This study used a model where the production of methane is modeled by acetoclastic and hydrogenotrophic mechanisms only, and consumption by methanotrophy. Acetate and CO2 are inputs based on measurements of soil water and pH. Acetate is formed by fermentation and by homoacetogenesis as defined in Xu et al. (2015) in Eq A15 and 16 in the appendix, see also revised Fig. 1b. The model is not completely comprehensive and syntrophic acetate oxidation is neglected. In modeling, it is important to balance parsimony and mechanisms; we have made choices here to avoid overfitting. We describe the impact of some neglected processes in Section 4.3. We will add that homoacetogenesis is

included and that syntrophic acetate oxidation is neglected in the methods Section 2.4.1 in a revision.

C: L97-105: The dominant soil type in the current study is Ultisol. I guess the whole methane cycling will be very different in other tropical soils (What is with Oxisols?) What are the soil types in the other studies mentioned in the introduction? It would be great to see a more detailed description of the soil type. How did bulk density change with depth? At what depth does clay accumulation (subsurface zone) start? How big is the eluvial horizon? R: The response to this comment also includes response to the first detailed review comment (comparison to other soils). The soil types of the other tropical studies of methane releases from the introduction are mentioned in a reply above. The studies in Costa Rica and Brazil were specific in showing that methane consumption was the major effect of the seasonal El Niño cycle (Costa Rica) or imposed drought manipulation (Brazil). Only the O'Connell et al. (2018) study showed the enhanced release of methane during post-drought recovery. We will add this distinction clearly in a revision. We cannot say if the O'Connell observations are related to soil type or some other mechanism. Oxisols, because of their high oxide and clay contents, may also have microsites.

We have data on bulk density changes up to 1 m that are in review with Ecology and Evolution. In that manuscript we find that on the surface, bulk density ranges from 0.5 to 0.7 g/cm3. By 25 cm depth, bulk density is 0.8 to 1.1 g/cm2. These results are similar to those published by Johnson et al. (2014).

A publication that provides a detailed soil survey in the immediate vicinity of the field site (Soil Survey Staff, 1995) lists the following soils: Zarzal, Cristol, and Prieto soils. For all soils, the litter layer is minimal. For the Zarzal soil, the surface (A) horizon is usually 4 cm thick, and the B horizon is around 150 cm thick. The Cristol soil surface horizon is listed as 6 cm thick, and B horizon to around 150 cm thick. The Prieto soil surface horizon is listed as 6 cm thick, and B horizon to around 130 cm thick.

C: L107: Why was soil only sampled from the top 10 cm of the soil? I guess methane consumption dominates at the surface due to oxic conditions while in the deeper soil horizons methane production dominates due to more anoxic conditions. R: The soil flux chambers are on the soil surface and the sampling strategy was designed to focus on near-surface measurements, as described above and in another comment response below. These soils are very rich in clays, beginning at the interface with the atmosphere, and are often very wet, and decades of lab- and field-scale studies by the Silver group (see various citations in the manuscript) confirm that methane production can occur in soils at shallow depths, especially in the valley soils but to a lesser extent in the slope and ridgetop soils.

C: L111-114: What was the detection limit for acetate? R: We assumed it was equivalent to the lowest standard by HPLC analysis, i.e., 0.5 ïÅ■M.

C: L117-120: The chemical data (what chemical data? Only pH?) for bulk soil is from the 0-10 cm soil depth and acetate and DOC from the pore water from 10-30 cm soil depth? I do not understand how you relate this information, taken from different soil depths, to each other. Finally, where do you find or where do you assume these microsites? (only in the top 10 cm or below? Or may be even below 30 cm? R: We assume that soil flux chambers placed on the top of the soil surfaces are dominated by fluxes from relatively shallow depths. This summary of fluxchamber methods by two experts in this field states that it is usually assumed surface chambers measure fluxes from about 25 cm in depth (Rochette and Hutchinson 2005). Given this perspective, we collected soil and soil water measurements from the 0 to 30 cm depth to best relate to surface flux chamber measurements. The chemical data used in this study consisted of acetate from the lysimeters located at a maximum depth of 10 cm and a maximum depth of 30 cm. The soil samples on which pH was measured and from which DOC was extracted were collected from 0-10 cm depth.

Microsites are inferred by observations such as originally presented in O'Connell et al. (2018), i.e., sudden releases of methane during post-drought recovery; and from

seminal publications such as Silver et al. (1999). In the latter, co-occurrences of soil oxygen and methane in bulk soils presume the abundance of anaerobic microsites in these upland soils. Because the soils have abundant clay and iron oxides at all depths, it is likely that microsites are pervasive throughout.

C: L144-148: What is with methylotrophic methanogenesis. You should discuss about the potential contribution of methylotrophic methanogenesis (see Norrow et al. 2019). R: Our model does not include methylotrophic methanogenesis. We will add this to the mention of several other processes not considered in our model in Section 2.4.1 and cite the relevant paper.

C: L173-174: Why not? R: We believe the reviewer is asking why iron reduction and oxidation are not included in this study. As explained in Section 4.3, we chose to take a more simplified approach to start, just focusing on substrates and microbial functional groups for methanogenesis (both acetoclastic and hydrogenotrophic methanogenesis) and methanotrophy. We feel that our model does a decent job at reproducing the data, considering "normal" and "drought" conditions, and involving two different time frames of data collection, and we consider that as confirmation of the validity of our approach. We acknowledged in Section 4.3 that iron reduction can alter the pH of the soils and soil water and enhance methane emissions; and that iron reduction can also support anaerobic methane oxidation, as well as using acetate as a substrate and thereby reducing net methane emissions. We acknowledge that our model fits are not perfect, as you can see from Fig 3, the model misses the highest methane fluxes seen during the post drought. This could be a result of not considering the pH effects of iron reduction that enhance methanogenesis. However, other processes in the iron cycle reduce methanogenesis, so the benefit of including unconstrained iron cycling processes is unclear without additional information to constrain the model. Therefore, we felt it was appropriate to focus only on the mechanisms covered in this study.

C: L206: Why 15 cm? R: Please refer to the response for comments regarding L117-120 (above).

C: L214: How do calculate "total microsites"? I would assume that there are way more microsites in clayey horizons below 15 or even 30 cm soil depth? Overall, I think you have to explain the concept of microsites, more in detail? You are scratching only the soil surface at the moment but in my opinion the biggest methane production potential occurs in deeper parts of the Ultisol. R: A seminal study in 1999 by Silver et al (cited in our manuscript) in a nearby Ultisol soil took measurements of methane and oxygen at 10 and 35 cm depth, as well as surface chamber flux measurements. The authors found that methane concentrations were higher at the shallower depths. Most of the subsequent papers from Luquillo Experimental Forest used surface chamber measurements and focused on shallow soil depths. We assumed that size of the microsites should be at least an order magnitude lower than the bulk soil measurements we had for soil methane fluxes. Using this logic, we decided that "diameter" of microsites should be in "mm" scale as the diameter of soil chambers we used are in "cm" scale (15.24 cm). Thus, we did the math to come up with the number of "total microsites" (i.e. 10000) such that the diameter of microsites meets our criteria. We will add this to methods Section 2.4.2.

C: L270-272: Why are acetate and hydrogen production decreasing when aceticlastic and hydrogenotrophic methanogenesis also decrease? Aceticlastic and hydrogenotrophic methanogens consume acetate and hydrogen, respectively. So, if there is a decrease in aceticlastic methanogens, I would first assume an increase in acetate concentration and thereafter a sharp decrease if oxygen levels further increase. R: Decreasing acetate and hydrogen production in the model are consistent with decreasing acetoclastic and hydrogenotrophic methanogenesis. Eq 1, biomass amounts and substrate concentrations together contribute to the reaction producing methanogenesis; it is not sequential.

C: L280: How do explain that? R: We believe the reviewer is referring to predicted changes in the biomass of different microbial functional groups during drought, drought recovery, and post-drought. These are model predictions, that are based upon the

mechanisms within the model, and the input data that constrains model behavior (pH, acetate, DOC, and CH4 fluxes). Although we lack measurements of the microbial biomass of specific microbial taxonomies or functional groups during the events in this paper (as we have acknowledged), microbes can respond rapidly to changes in their environment. It is important to distinguish that the model is predicting changes in biomass of a particular microbial functional group and is not predicting large changes of the bulk microbial biomass in the soil. Bulk microbial biomass in the soil is likely to double or perhaps quadruple in response to changes in conditions, but individuals can grow exponentially (Goberna et al. 2010; Pavlov and Ehrenberg 2013; Roussel et al. 2015; Buan 2018).

C: L286-287: Why does the increasing production of acetate lowers the pH? R: Acetate production is a source of proton (L375 and citations therein, particularly Xu et al. 2015) as seen in Eq 7 and Fig. 1.

C: L301-306: If the diffusion of H2 increases during drought, one may think that hy-drogenotrophic methanogenesis should increase as well. However, it does not in-crease because of increasing oxygen levels. That should be made clear! R: This is true, particularly for the ridge and slope soils (Fig. S6), but is less true for the valley soils because gas diffusion remains limited throughout the event in the valley soil (Fig. S8). This explanation is also consistent with the methanotrophic biomass (Fig. S5). We will revise accordingly.

C: L304-309: The diffusion of acetate increases upon rewetting but that of H2 de-creases. Why do you observe an increase in overall gross methane production. First, I would assume that under relatively acid conditions, hydrogenotrophic methanogenesis dom- inates. I think the overall increase of methane emissions upon rewetting is be-cause of oxygen depletion and therefore the stimulation of methanogenesis in general . . .. and not because of increasing acetate concentrations or a shift in the methanogenic pathway of methane formation. If it is really aceticlastic methanogenesis that is stimu-lated, you should provide some isotopic data. There is competition for acetate between several microorganisms. In the end it could be simply stimulation or inhibition of fer- mentation or homoacetogenesis that drives changes in the amount of observed acetate concentrations. R: In the model and likely in reality, all of these processes are occurring simultaneously, although some may be more important than others. The purpose of this paper is to better understand the unique processes observed in the 2015 drought, that do not have parallels in either Costa Rica or Brazil. A model was used that was originally validated by simulating methane emissions in Arctic soils (Xu et al. 2015), but the experiments were lab-scale incubations. Therefore, the model was enhanced here to consider diffusional processes that may be important at the field scale. From these beginnings, the model seems to explain the observations from the 2015 Puerto Rico drought, and 2016 "normal" scenario. It is to some extent a thought exercise, i.e., "If we were to include the processes X, Y, and Z, could we reproduce the observed data?". If the answer is yes, then that indicates the model provides valid explanations for the observations. However, we acknowledge that the model is not fully comprehensive – in the interest of parsimony, some processes are excluded. And that the presentations in the paper are model simulations, and that in some cases, true validation data, e.g., microbial biomass of the different functional groups or $H_2$ concentrations or isotopic data, is lacking. We believe this exercise is valuable, despite the shortcomings therein, and we have tried to be open about where either measurements or model processes are lacking. The mechanism of hydrogenotrophic methanogenesis does not well explain the observations. Hydrogen should have been freely available during the drought, yet little methanogenesis was observed. As the soils wetted, hydrogen would become less available as its diffusion rate will decrease strongly (Fig. S8). So hydrogenotrophic methanogenesis cannot readily explain the post-drought spike in methane concentrations. Acetate diffusion, however, more readily explains the observations (Fig. S8). As wetting commenced and progressed, acetate may become more available to microorganisms and can enhance methanogenesis. At the same time, wetting decreases oxygen availability, decreasing the role of methanotrophy (Fig. S8) and allowing more methane to escape the subsurface, despite limitations in gas diffusion.

C: L318: Again, how do you define the microsites? R: We are sorry that microsites are inadequately defined in this version. We have revised the figure and caption as follows: "Top panel (a) shows the model representation of soil microsite distribution (modified from Sihi et al., 2020a, also see Eq. 14). The cylinder refers to the volume beneath the soil chambers. The intensity of different cylinder colors figure refers to rate of a process or the intensity of a concentration inside microsites in each theoretical cylinder, e.g., a dark color means a higher rate/intensity, and a light color means a lower rate/intensity for a given process. The 2D graph on the right refers to the probability density function of the rate of the process or intensity of the concentration in the bulk soil. A wide distribution skewed to the right (dark-colored line) implies higher bulk rates of the process or higher concentrations, and a narrow distribution skewed to the left (light-colored line) implies lower bulk rates of the process or lower concentrations, of any of the following: soil moisture, solute concentration, gas concentration, gas diffusion, solute diffusion, methane production, or methane oxidation."

FYI, we have revised the figure below from the original as follows: Added an arrow on the x axis pointing towards the right, denoting increasing concentrations or rates. Moved the light-colored function to the left of the dark-colored function and made it much more narrow and signify less impact on bulk rates/concentrations compared to the dark-colored function (more impact on bulk rates/concentrations).

Please see revised Figure 1 as a pdf attached separately.

C: L346-348: Again, what makes you so sure that it is acetate that drives net methane emissions and not H2/CO2 and a decrease in oxygen? R: All of these processes are happening simultaneously. Our model simulation suggests that acetate is driving most of the methane increases, and that decreases in methanotrophy due to decreases in oxygen, are both more important than hydrogenotrophic methanogenesis, pls see also sensitivity analysis in Fig 8. Fig S5 shows that both kinds of methanogens increase during drought recovery and post-drought, but that acetotrophic methanogens are two orders of magnitude more abundant than hydrogenotrophic methanogens. Additionally,

the acetate hypothesis makes more sense because under drought conditions, acetate may accumulate in microsites. During wetting, the acetate may become more available to the methanogens as solute diffusion becomes enhanced (Fig. S8), resulting in strong methane releases. Hydrogen, on the other hand, would be readily available during drought conditions because its diffusion would not be limited. So, hydrogen substrate availability does not explain the observations of strong methane releases under wetting conditions. In fact, the model simulations suggest that hydrogen diffusion is lessened under wetting conditions (Fig. S8).

C: L369-372: and homoacetogenesis? R: Yes. We will specify in Methods Section 2.4.1 and in Fig. 1b caption that this mechanism is included in the model.

REFERENCES Buan, N.R. Methanogens: pushing the boundaries of biology: Emerging Topics in Life Science, 2, 629–646, https://doi.org/10.1042/ETLS20180031, 2018. Goberna, M., Gadermaier, M., García, C., Wett, B., Insam, H.: Adaptation of methanogenic communities to the cofermentation of cattle excreta and olive mill wastes at 37°C and 55°C, Applied and Environmental Microbiology, 76, 19, 6564–6571, doi:10.1128/AEM.00961-10, 2010. Johnson, A. H., Xing, H. X., and Scatena, F. N.: Controls on Soil Carbon Stocks in El Yunque National Forest, Puerto Rico: Soil Sci. Soc. Am. J., 79, 294-304, doi:10.2136/sssaj2014.05.0199, 2014. Harris, N. L., Lugo, A. E., Brown, S., and Heartsill Scalley, T. (Eds.): Luquillo Experimental Forest: Research history and opportunities, EFR-1, Washington, DC: U.S. Department of Agriculture, 152 p., 2012. O'Connell, C. S., Ruan, L., and Silver, W. L.: Drought drives rapid shifts in tropical rainforest soil biogeochemistry and greenhouse gas emissions. Nat. Commun., 9, 1-9, 10.1038/s41467-018-03352-3, 2018. Pavlov, M. Y., and Ehrenberg, M.: Optimal control of gene expression for fast proteome adaptation to environmental change, doi/10.1073/pnas.1309356110 Rochette, P., and Hutchinson, G. L.: Measurement of soil respiration in situ: Chamber techniques, Publications from USDA-ARS / UNL Faculty, 1379, https://digitalcommons.unl.edu/usdaarsfacpub/1379, 2005. Roussel E. G., Cragg, B. A., Webster, G., Sass, H., Tang, X., Williams, A.
S., Gorra, R., Weightman, A. J., and Parkes, R. J.: Complex coupled metabolic and prokaryotic community responses to increasing temperatures in anaerobic marine sediments: critical temperatures and substrate changes, FEMS Microbiology Ecology, 91, 2015, fiv084, doi: 10.1093/femsec/fiv084, 2015. Sihi, D., Davidson, E. A., Savage, K. E., and Liang, D.: Simultaneous numerical representation of soil microsite production and consumption of carbon dioxide, methane, and nitrous oxide using probability distribution functions, Glob. Change Biol., 26, 200-218, https://doi.org/10.1111/gcb.14855, 2020. Silver, W. L., Lugo, A., and Keller, M.: Soil oxygen availability and biogeochemistry along rainfall and topographic gradients in upland wet tropical forest soils, Biogeochemistry, 44, 301-328, 1999. Soil Survey Staff: Order 1 Soil Survey of the Luquillo Long-Term Ecological Research Grid, Puerto Rico, USDA, NRCS, 1995. Xu, X., Elias, D. A., Graham, D. E., Phelps, T. J., Carroll, S. L., Wullschleger, S. D., and Thornton, P. E.: A microbial functional group-based module for simulating methane production and consumption: Application to an incubated permafrost soil, J. Geophys. Res.-Biogeo., 120, 1315-1333, 10.1002/2015jg002935, 2015.

Please also note the supplement to this comment:
https://bg.copernicus.org/preprints/bg-2020-222/bg-2020-222-AC1-supplement.pdf

―――――――――――――――――

[Figure]

**(a)**

Microsite frequency

$[S_i]$, $SoilM_i$, $Diff_i$, $Prod_i$, $Ox_i$

Increasing concentrations or rates

**(b)**

SOM/Litter

$CO_2$

DOC

Acetate
$H^+$

$H^+$
Acetate

$H_2$

$CO_2$

$CO_2$

$CH_4$

$CH_4$

**(1)**

**(2)**

**(3)**

(1) Acetoclastic methanogenesis
(2) Hydrogenotrophic methanogenesis
(3) Methanotrophy

**Fig. 1.** Revised Fig 1 See text file for caption

---

## Author Comment (AC2) · 2 Oct 2020

C: The manuscript of Debjani Sihi and colleagues brings up a very interesting topic on disentangling gross methane emission and uptake from wet tropical forest soil using a combination of microbial functional group CH4 model and a diffusivity module. This work clearly shows how landscape topography and climate affect net CH4 emissions due to shift of substrate production, soil redox conditions, and diffusivity of O2, H2, and acetate under drought and recovery phases. The experimental work is well performed, convincing and well discussed in the context of previous literature. The manuscript is

well organized and clearly written and I enjoyed reading it. I only have a few comments that should be addressed: R: Thank you kindly for the positive comments and for the constructive suggestions, all of which we have adopted.

C: Line 54 Should it be "increased consumption of atmospheric CH4"? R: Good catch. Thank you, the manuscript will be corrected as suggested.

C: Line258 The correlation seems stronger and more negative in 2015 (-0.36) than 2016 (-0.61). R: The reviewer is correct. The sentence L258 should be changed to "The correlation between CH4 emissions and O2 concentrations was stronger and more negative in 2015 than 2016."

C: Line 321-322 You defined pre-drought period from DOY 57-115 instead of DOY 200. The details in results should be checked. R: I believe we mean to say "during the drought period (DOY 200)". We will double-check all other similar references and ensure there are no additional errors. Thank you.

C: Fig. 1 I appreciate the conceptual figures herein, but it looks a bit confusion and I do not well understand what means in panel a. How to relate microsite frequency with soil properties? R: We agree this figure could use some revisions. Our caption says currently: Top panel (a) shows the model representation of soil microsite distribution (modified from Sihi et al., 2020, also see Eq. 13). Different shades indicate substrate concentration [Si], soil moisture (SoilMi), diffusion (Diffi) of solutes and gases, production (Prodi) and oxidation (Oxi) processes at each microsite.

We propose to revise the caption as follows: "Top panel (a) shows the model representation of soil microsite distribution (modified from Sihi et al., 2020a, also see Eq. 14). The cylinder refers to the volume beneath the soil chambers. The intensity of different cylinder colors refers to rate of a process or the intensity of a concentration inside microsites in each theoretical cylinder, e.g., a dark color means a higher rate/intensity, and a light color means a lower rate/intensity for a given process. The 2D graph on the right refers to the probability density function of the rate of the process or intensity of

the concentration in the bulk soil. A wide distribution skewed to the right (dark colored line) implies higher bulk rates of the process or higher concentrations, and a narrow distribution skewed to the left (light colored line) implies lower bulk rates of the process or lower concentrations, of any of the following: soil moisture, solute concentration, gas concentration, gas diffusion, solute diffusion, methane production, or methane oxidation."

FYI, we have revised the figure from the original as follows: Added an arrow on the x axis pointing towards the right, denoting increasing concentrations or rates. Moved the light-colored function to the left of the dark-colored function and made it much more narrow and signify less impact on bulk rates/concentrations compared to the dark-colored function (more impact on bulk rates/concentrations).

Please see revised Figure as pdf, uploaded separately.

C: Do substrate concentration, soil moisture, diffusivity of solute and gas present the similar pattern for one kind of microsite? R: The frequency distribution for the microsites is the same for all of these, according to Eq 14. But, the diffusivity of liquids is according to Eq 11 and 12; diffusivity of gasses according to Eq 8, 9, and 10. Here is a little more information on the microsites that will be included in the revision methods Section 2.4.2: We assumed that size of the microsites should be at least an order magnitude lower than the bulk soil measurements we had for soil methane fluxes. Using this logic, we decided that "diameter" of microsites should be in "mm" scale as the diameter of soil chambers we used are in "cm" scale (15.24 cm). Thus, we did the math to come up with the number of "total microsites" (i.e. 10000) such that the diameter of microsites meets our criteria. This is why the frequency of microsites is the same for both low rate/intensities (light yellow line Fig 1a) and high rate/intensities (dark line Fig 1a) (see also Fig. 7 in manuscript and Fig. S10 in SI).

C: Why this figure links to Eq.13? R: This figure should link to Eq 14; apologies for the confusion. It will be corrected in the revision.

C: Also in panel b, it would be more clear for readers if you could adjust it to a better shape or based on the clue of the present study. Can you try to improve the conceptual figure and clarify this in the legend? R: We propose to remove the Air/Soil diagram at the top of this figure, and to remove the word "solute diffusion" from the figure. Panel b should only represent the geochemical pathways that the model is representing, and we should rely on panel (a) to address diffusion. We hope that makes the content of both panels more understandable.

We propose to revise the caption from the current version: Bottom panel (b) is the schematic of the microbial functional group-based model coupled with a diffusivity module (Microbial Model for Methane Dynamics-Dual Arrhenius and Michaelis Menten, M3D-DAMM) for simulating soil methane (CH4) dynamics in field soils (Modified from Xu et al., 2015), where SOM = soil organic matter, CO2 = carbon dioxide, DOC = dissolved organic carbon, H+ is the hydronium ion, and H2 = dihydrogen molecule. Proposed revised figure caption: Bottom panel (b) is the schematic of the microbial functional group-based model for simulating soil methane (CH4) dynamics in field soils (modified from Xu et al., 2015). The schematic represents the decomposition of soil organic matter (SOM) and plant litter into carbon dioxide (CO2) and dissolved organic matter (DOC); the production of acetate and hydronium ion (H+) from decomposition and fermentation of DOC which also decreases pH, the production of acetate and hydronium ion (H+) from homoacetogenesis which decreases pH; and the production of dihydrogen ion (H2) and CO2 from decomposition of DOC. The intermediary products then have three possible non-mutually exclusive pathways (1) acetoclastic methanogenesis, which is the production of methane from aqueous acetate found in soil solutions, (2) hydrogenotrophic methanogenesis, which is the production of methane from hydrogen, and (3) methanotrophy, which is the oxidation of methane into carbon dioxide.

Please see revised Figure as pdf, uploaded separately.

C: Fig.3 and 4 The label of y-axis for soil moisture and oxygen should between 0-1

rather than 0-100, as the unit is V V-1. Otherwise, the unit should change to %. R: Thank you for pointing this out. We will adjust the unit of the axis in Figs 3 and 4, S8(g)(h)(i), and S10(i)(j).

C: Fig.4 and 6 The unit of CH4 emission should be uniform. Some of them are nmol m-2 S-1, while others are nmole m-2 S-1. Also the unite of acetate (Fig. 2). R: Thank you, we should use nmol and ïA■mol (and not "mole"). Corrections will be made to figures 2,3,4,7, and S5,S6,S7,S8,S9,S10.

Please also note the supplement to this comment:
https://bg.copernicus.org/preprints/bg-2020-222/bg-2020-222-AC2-supplement.pdf

[Figure]

**Fig. 1.** Revised Fig 1 See text file for caption

---

## Author Response (AR2)

The study "Representing methane emissions from wet tropical forest soils using microbial functional groups constrained by soil diffusivity" by Sihi et al. tries to explain soil methane emission dynamics in tropical forest soils of Puerto Rico during normal and drought conditions. They combine field measurements with modelling efforts (Microbial Model for Methane Dynamics-Dual Arrhenius and Michaelis Menten (M3D-DAMM). Overall, I think it is a really nice study that tries to combine microbial with biogeochemical data to investigate ecosystem methane dynamics. However, I have some general and some minor comments.
*Thank you kindly for the positive comments and for the constructive suggestions that really strengthened our paper.*

The authors should describe the concept of "microsites" more in detail. The authors focused on the top 10 or 10-30 cm of their Ultisol although methane production/consumption dynamics in the deeper clay-accumulating horizons may be more important for the overall net methane emissions from their Ultisols. The authors do not discuss that and do not compare with other soils. How does the abundance of microsites change with soil type, soil depth and other ancillary variables?
*Our sampling strategy, both of soils and of soil water, are geared to accompany the greenhouse gas flux measurements which are taken on the soil surface. Past studies (Silver et al. 1999) have taken methane concentrations at depth in similar soils, and found that concentrations are higher at 10 versus 35 cm. Examination of the soils in the Luquillo mountains have found that SOC maximums are around 35 cm depth (Johnson et al. 2014). For this site, clay is abundant at shallow depths, e.g., 20-30% clay at 0 to 10 cm depth (L177). Therefore, we designed our sampling strategy for shallow depths. We added text and the Johnson citation to a revision to ensure readers also understand this concurrence of sampling strategy and past observations (L189-194). Thanks for pointing this out; adding these details helps the manuscript.*

*There are no specific measurements of microsites at any depth at this site; microsites are inferred because of many observations of co-occurrences of oxygen concentrations in the soil along with $CH_4$ fluxes; and because of the rich clay, iron oxides, and visible redox mottling, particularly evident in the valley and slope soils (papers cited in L111-134). Techniques for measuring microsite activities remain very limited to date, here or elsewhere (L134). We added this information to in a new methods section 2.4.3 devoted to microsite modeling (L457-462).*

*We will address the issue of comparison to other soils in response to a more detailed question below.*

Minor comments: L22-25: Is it important to give this information? I would only include the most significant ones that support your guiding questions! What is the difference between "<" and "«". The abstract should self-explanatory.
*Good point thank you. We simplified the abstract and the manuscript throughout by using only < or >; it was a subjective difference between "<" and "<<". The geochemical parameters in L23-25 were measured and are key model inputs, so we rephrased to say they were measured at the field site, but we removed the parenthetical material regarding differences in their values with respect to topographic*

*positions which is not germane to the abstract.*

L33: write aceticlastic methanogenesis instead of "acetotrophic" and "acetoclastic" (check the whole manuscript)
*Thank you for catching these typos; corrected as suggested.*

L43: what are wet tropical forest soils? What are wet tropical forests and what is the difference between wet tropical forest soils and upland soils? How do you define that?
*The soils are classified as wet tropical forest soils according to the Holdridge life zone system, which considers rainfall, elevation, latitude, humidity, and evapotranspiration, as described specifically regarding the Luquillo Experimental Forest (Harris et al. 2012). Upland refers only to topography. The soils in our current study (including the valley soils) could be referred to as "upland" in that they are all located in a lower montane region (~350 m elevation). We added the Harris citation and a description of what is meant by "wet" tropical forest soils to the methods (L156). See also L88 that mentions that upland tropical soils can produce methane.*

L53-65: What are the soil types in the different studies? Since methanogenesis and methanotrophy are substrate-limited, the soil type with its specific biogeochemistry is very important. The authors mention that there are several studies that report effects of drought on net methane emissions across different wet tropical forest soils. Consequently, they should mention the different soil types.
*We added the soil type (below) when the studies were mentioned in L97-105 and elsewhere as relevant.*
*Aronson et al. 2019 (Costa Rica) Oxisols*
*Davidson et al. 2004, 2008 (Brazil) Oxisols*
*Wood and Silver 2012 (Puerto Rico) Ultisols with a similar climate and parent material*
*O'Connell et al. 2018 (Puerto Rico) Ultisols, same soils as current study*

L66-74: Oxygen may not be the only factor for methane emissions upon rewetting. The observed rapid flush of methane in response to a wetting event may be driven by rapid depletion of other electron acceptors, as well. The major focus of this paragraph is on oxygen but what is with acetate, H2 and CO2?
*We concur with the reviewer's suggestions; in fact, that is part of the reason we initiated this study. We wanted additional data on acetate concentrations, and to use a predictive model to better understand the conflicting/collaborating roles of oxygen, substrate, and microbial biomass in controlling methane emissions. In the revision, we clarified the role of these different substrates in controlling methane emissions in this paragraph so it does not appear we are only focused on oxygen. We also discussed their availability in terms of soil moisture and diffusion limitations (L113-125).*

L85-86: and why not H2 and CO2? How do you account for acetate formation during fermentation and homoacetogenesis? How do you account for syntrophic acetate oxidation? How can you explain the "contrasting patterns of observed CH4 emissions", when sources and sinks of acetate etc. is not measured or simply not known?
*Here, we use the model to provide quantitative answers in accordance with specific hypotheses stemming from the O'Connell et al. 2018 paper. The idea of this paper is to consider a relatively simple set of mechanisms and see how well a model that quantitatively represents these mechanisms can explain the complex observations. This study used a model where the production of methane is modeled by acetoclastic and hydrogenotrophic mechanisms only, and consumption by methanotrophy. Acetate is based on measurements of soil water. There are not measurements to constrain H2 concentrations, but $CO_2$ concentrations in water are estimated according to the pH. Acetate is formed by fermentation and by homoacetogenesis as defined in Xu et al. (2015) in Eq A15 and 16 in their appendix, see also our revised Fig. 1b and caption. There is a pH feedback to acetate production that affects methanogenesis (Eqn 7) that is also described in Fig 1b caption. The model is not completely comprehensive and syntrophic acetate oxidation is neglected. In modeling, we balance parsimony and mechanisms and data limitations; we have made choices here mostly following the original model design (Xu et al. 2015). We describe the impact of some neglected processes in Section 4.3. We added that homoacetogenesis is included and that syntrophic acetate oxidation is neglected in the methods Section 2.4.1 in the revision (L280-281).*

L97-105: The dominant soil type in the current study is Ultisol.  I guess the whole

methane cycling will be very different in other tropical soils (What is with Oxisols?) What are the soil types in the other studies mentioned in the introduction? It would be great to see a more detailed description of the soil type. How did bulk density change with depth? At what depth does clay accumulation (subsurface zone) start? How big is the eluvial horizon?

*The response to this comment also includes response to the first detailed review comment (request for comparison to other soils). The soil types of the other tropical studies of methane releases from the introduction are mentioned in a reply above. The studies in Costa Rica and Brazil on Oxisols were specific in showing that methane consumption was the major effect of the seasonal El Niño cycle (Costa Rica) or imposed drought manipulation (Brazil), which was similar to here. Only the O'Connell et al. (2018) study showed the enhanced released of $CH_4$ during post-drought recovery. We cannot say for certain if the O'Connell observations are related to soil type. Oxisols, because of their high oxide and clay contents, may also have microsites. In particular, hematite clay minerals (found in both soil types) enhances formation of soil aggregates because of their high surface area and charge properties. Soil organic matter can also enhance aggregation and at the same time consume $O_2$. So we added a statement in the introduction (with the soil type mentions) to note that these mechanisms for aggregate formation might also lead to anaerobic microsites in Oxisols and Ultisols (L131-134).*

*We have data on bulk density up to 1 m that are in press with Ecology and Evolution (K. Cabugao et al., in press). In that manuscript we find that on the surface, bulk density ranges from 0.5 to 0.7 g/cm3. By 25 cm depth, bulk density is 0.8 to 1.1 g/cm3. These results are similar to those published by Johnson et al. (2014) and Silver et al. (1999) at nearby sites in the same forest. See L185-185.*

*A publication that provides a detailed soil survey in the immediate vicinity of the field site (Soil Survey Staff, 1995) lists the following soils: Zarzal, Cristol, and Prieto soils. For all soils, the litter layer is minimal due to very rapid litter decomposition rates (Parton et al. 2007, Cusack et al. 2009). For the Zarzal soil, the surface (A) horizon is usually 4 cm thick, and the B horizon is around 150 cm thick. The Cristol soil surface horizon is listed as 6 cm thick, and B horizon to around 150 cm thick. The Prieto soil surface horizon is listed as 6 cm thick, and B horizon to around 130 cm thick. We added this information and the citation to the site description (L181-183).*

L107: Why was soil only sampled from the top 10 cm of the soil? I guess methane consumption dominates at the surface due to oxic conditions while in the deeper soil horizons methane production dominates due to more anoxic conditions.

*These soils are very rich in clays with high NPP and high surface C inputs (Weaver and Murphy, 1990), beginning at the interface with the atmosphere, and are often very wet, and many lab- and field-scale studies by the Silver group (see various citations in the manuscript) confirm that methane production can occur in soils at shallow depths, especially in the valley soils and to a lesser extent in the slope and ridgetop soils. The soil flux chambers are on the soil surface and the sampling strategy was designed to focus on near-surface measurements that are relevant to surface monitoring, as described above and in another comment response below.*

L111-114: What was the detection limit for acetate?
*We assumed it was equivalent to the lowest standard by HPLC analysis, i.e., 0.5 $\mu$M.*

L117-120: The chemical data (what chemical data? Only pH?) for bulk soil is from the 0-10 cm soil depth and acetate and DOC from the pore water from 10-30 cm soil depth? I do not understand how you relate this information, taken from different soil depths, to each other. Finally, where do you find or where do you assume these microsites? (only in the top 10 cm or below? Or may be even below 30 cm?

*We assume that soil flux chambers placed on the top of the soil surfaces are dominated by fluxes from relatively shallow depths. A summary of fluxchamber methods by two experts in this field states that it is usually assumed surface chambers measure fluxes from about 25 cm in depth (Rochette and Hutchinson 2005). Given this perspective, we collected soil and soil water measurements from the 0 to 30 cm depth to best relate to surface flux chamber measurements. The chemical data used in this study consisted of acetate and DOC from the lysimeters located at a minimum depth of 5-10 cm and a maximum depth of 25-30 cm (the lysimeters are 5 cm in length). The soil samples on which pH were measured were collected from 0-10 cm depth. See L189-194 and L259.*

*Microsites are inferred by observations such as originally presented in O'Connell et al. (2018), i.e., sudden releases of methane during post-drought recovery; and from seminal publications such as Silver et al. (1999). In the latter, co-occurrences of soil oxygen and methane in bulk soils presume the abundance of anaerobic microsites in the soils. Because the soils have abundant clay and iron oxides at all depths, it is likely that microsites are pervasive throughout. This has also been observed repeatedly in soil incubation studies using surface soils from this. See section 2.4.3.*

L144-148: What is with methylotrophic methanogenesis. You should discuss about the potential contribution of methylotrophic methanogenesis (see Norrow et al. 2019).
*Our model does not include methylotrophic methanogenesis. We will add this to the mention of several other processes not considered in our model in Section 2.4.1 and cite the relevant paper (L282).*

L173-174: Why not?

*We believe the reviewer is asking why iron reduction and oxidation are not included in this study. As explained in L300 in the methods and in Section 4.3, we chose to take a more simplified approach to start, just focusing on substrates and microbial functional groups for methanogenesis (both aceticlastic and hydrogenotrophic) and methanotrophy. We feel that our model does a reasonable job at reproducing the data, considering "normal" and "drought" conditions, and involving two different time frames of data collection, and we consider that as confirmation of the validity of our approach. We acknowledged in Section 4.3 that iron reduction can alter the pH of the soils and soil water and enhance methane emissions; and that iron reduction can also support anaerobic methane oxidation, as well as using acetate as a substrate and thereby reducing net methane emissions. We acknowledge that our model fits are not perfect, as you can see from Fig 3, the model misses the highest methane fluxes seen during the post drought. This could be a result of not considering the pH effects of iron reduction that enhance methanogenesis. However, other processes in the iron cycle reduce methanogenesis, so the benefit of including unconstrained iron cycling processes is unclear without additional information to constrain the model. Therefore, we felt it was appropriate to focus only on the mechanisms covered in this study.*

L206: Why 15 cm?

*This is the average depth sampled (ranges from 0 to 30 cm).*

L214: How do calculate "total microsites"? I would assume that there are way more microsites in clayey horizons below 15 or even 30 cm soil depth? Overall, I think you have to explain the concept of microsites, more in detail? You are scratching only the soil surface at the moment but in my opinion the biggest methane production potential occurs in deeper parts of the Ultisol.
*A seminal study in 1999 by Silver et al (cited in our manuscript) in a nearby Ultisol soil took measurements of methane and oxygen at 10 and 35 cm depth, as well as surface chamber flux measurements. The authors found that $CH_4$ concentrations were higher at the shallower depths. Most of the subsequent papers from Luquillo Experimental Forest used surface chamber measurements and focused on shallow soil depths, pls see L189-193 We surmise that perhaps diffusion would be quite slow at greater depth in these rich clays, that substrate (SOC) availability which is maximum at 0-30 cm would also be lessened, and that microbial biomass would be lower (Hall et al. 2016). Therefore, we focused on shallow depths.*
*We assumed that size of the microsites should be at least an order magnitude lower than the bulk soil measurements we had for soil $CH_4$ fluxes. Using this logic, we decided that "diameter" of microsites should be in "mm" scale as the diameter of soil chambers we used are in "cm" scale (15.24 cm). Thus, we did the math to come up with the number of "total microsites" (i.e. 10000) such that the diameter of microsites below each chamber meets our criteria, following Sihi et al. 2020a. We added this to the methods Section 2.4.3.*

L270-272: Why are acetate and hydrogen production decreasing when aceticlastic and hydrogenotrophic methanogenesis also decrease? Aceticlastic and hydrogenotrophic methanogens consume acetate and hydrogen, respectively. So, if there is a decrease in aceticlastic methanogens, I would first assume an increase in acetate concentration and thereafter a sharp decrease if oxygen levels further increase.
*Good catch, thank you, the cause and effect in that sentence was inverted (L611). It*

*now reads: "Simulated decreased production of acetate and hydrogen during the 2015 drought in the ridge and slope positions resulted in decreased biomass of aceticlastic methanogens and hydrogenotrophic methanogens (Figs. S5, S6)."*

L280: How do explain that?
*We believe the reviewer is referring to predicted changes in the biomass of different microbial functional groups during drought, drought recovery, and post-drought. These are model predictions, that are based upon the mechanisms within the model, and the input data that constrains model behavior (pH, acetate, DOC, and $CH_4$ fluxes). Although we lack measurements of the microbial biomass of specific microbial taxonomies or functional groups during the events in this paper (as we acknowledged), microbes can respond rapidly to changes in their environment. It is important to distinguish that the model is predicting changes in biomass of a particular microbial functional group and is not predicting large changes of the bulk microbial biomass in the soil. Bulk microbial biomass in the soil is likely to double or perhaps quadruple in response to changes in conditions, but individuals can grow exponentially (Goberna et al. 2010; Pavlov and Ehrenberg 2013; Roussel et al. 2015; Buan 2018). Please see L866.*

L286-287: Why does the increasing production of acetate lowers the pH?
*Acetate production is a source of proton (Eq 7 and Fig. 1).*

L301-306: If the diffusion of H2 increases during drought, one may think that hy-drogenotrophic methanogenesis should increase as well. However, it does not in-crease because of increasing oxygen levels. That should be made clear!
*This is true, particularly for the ridge and slope soils, please see clarification in L657 in Section 3.3. It is somewhat less true for valley soils, L669, so no changes were made there.*

L304-309: The diffusion of acetate increases upon rewetting but that of H2 decreases. Why do you observe an increase in overall gross methane production. First, I would assume that under relatively acid conditions, hydrogenotrophic methanogenesis dom-inates. I think the overall increase of methane emissions upon rewetting is because of oxygen depletion and therefore the stimulation of methanogenesis in general . . .. and not because of increasing acetate concentrations or a shift in the methanogenic pathway of methane formation. If it is really aceticlastic methanogenesis that is stimu-lated, you should provide some isotopic data. There is competition for acetate between several microorganisms. In the end it could be simply stimulation or inhibition of fer-mentation or homoacetogenesis that drives changes in the amount of observed acetate concentrations.
*The main issue is that CH4 emissions are not normally found in the ridge and slope soils (eg, see 2016 data), so a response of a simple increase in methanogenesis cannot explain the observations. The mechanism of hydrogenotrophic methanogenesis does not well explain the observations because as the soils wetted, hydrogen gas would become less available as its diffusion rate will decrease strongly (Fig. S8). Further, the turnover rate of $H_2$ in shallow soils is very high, so it is less likely (than a solute) to accumulate in microsites (Xu et al. 2015). Acetate diffusion in solution, however, more readily explains the observations (Fig. S8). As wetting commenced and progressed, acetate in solution may become more available to microorganisms and can enhance methanogenesis. At the same time, wetting decreases $O_2$ availability, decreasing the role of methanotrophy (Fig. S8) and allowing more methane to escape the subsurface, despite limitations in gas diffusion. Methanogenesis is definitely enhanced in response to decreasing $O_2$ as the reviewer points out. Text in the discussion supporting this interpretation (L732): "The return to dominantly reducing conditions also was predicted to stimulate fermentation and the production of acetate through homoacetogenesis (Fig. S6). Enhanced production and diffusion of acetate during recovery (Fig. S8) triggered growth in the predicted biomass of aceticlastic methanogens (Fig. S5), which in turn, increased rates of aceticlastic methanogenesis (Fig. S9)." See also L737: "Although secondary to aceticlastic methanogenesis, simulated rates of hydrogenotrophic methanogenesis also increased in anaerobic microsites (Figs. S9, S10), mediated by increased production of $H_2$ and subsequent stimulation of the biomass of hydrogenotrophic methanogens during the drought recovery in 2015 (Fig. S5)." The sensitivity analysis (Fig. 8) shows a stronger control by aceticlastic methanogenesis that by hydrogenotrophic methanogenesis. Both are important, however.*

*Unfortunately, there were not isotopic data available for the field study; these type of data are rarely available at the field scale. Assessing patterns in isotopic fractionation or even*

*isotope tracing (gross rates or label chasing) would make an interesting experiment for future research.*

L318: Again, how do you define the microsites?
*In addition to revising the methods section 2.4.3 as suggested above, we have revised Fig. 1 caption as follows:*
*"Top panel (a) shows the model representation of soil microsite distribution (modified from Sihi et al., 2020a, also see Eq. 14). The cylinder refers to the volume beneath the soil chambers. The intensity of different cylinder colors figure refers to rate of a process or the intensity of a concentration inside microsites in each theoretical cylinder, e.g., a dark color means a higher rate/intensity, and a light color means a lower rate/intensity for a given process. The 2D graph on the right refers to the probability density function of the rate of the process or intensity of the concentration in the bulk soil. A wide distribution skewed to the right (dark line) implies higher bulk rates of the process or higher concentrations, and a narrow distribution skewed to the left (light line) implies lower bulk rates of the process or lower concentrations, of any of the following: solute concentration $[S_i]$, gas concentration $[G_i]$, soil moisture $(SoilM_i)$, gas and solute diffusion $(Diff_i)$, methane production $(Prod_i)$, and methane oxidation $(Ox_i)$."*

*We have revised the figure from the original as follows: Add an arrow on the x axis pointing towards the right, denoting increasing concentrations or rates. Moved the light-colored probability-density function to the left of the dark-colored probability-density function and make it much more narrow and signify less impact on bulk rates/concentrations compared to the dark-colored function (more impact on bulk rates/concentrations).*

L346-348: Again, what makes you so sure that it is acetate that drives net methane emissions and not H2/CO2 and a decrease in oxygen?
*All of these processes are happening simultaneously. Our model simulation suggests that acetate is driving most of the $CH_4$ increases, and that decreases in methanotrophy due to decreases in oxygen, are both more important than hydrogenotrophic methanogenesis. Please see also the sensitivity analysis in Fig 8. Fig S5 shows that both kinds of methanogens increase during drought recovery and post-drought, but that acetotrophic methanogens are two orders of magnitude more abundant than hydrogenotrophic methanogens. Additionally, the acetate hypothesis is supported under drought condition as acetate may accumulate in microsites. During wetting, the acetate may become more available to the methanogens as solute diffusion becomes enhanced (Fig. S8), resulting in higher $CH_4$ production. The model simulations suggest that hydrogen diffusion is lessened under wetting conditions which is consistent with what we might expect for a gas diffusing through a liquid versus diffusing through air (Fig. S8). So, hydrogen substrate availability does not completely explain the observations of higher $CH_4$ production under wetting conditions, but it is a contributor. The low contribution of $H_2/CO_2$ is also caused by low concentration and high turnover rate of $H_2$ in soil; particularly in the top soil, which is where this experiment was carried out. Please see L737-752 for improved explanation.*

L369-372: and homoacetogenesis?
*Yes, please see L733.*

*Yes, we mean to say "during the drought period (DOY 200) (L688)". We will double-check all other similar references and ensure there are no additional errors. Thank you.*

Fig. 1 I appreciate the conceptual figures herein, but it looks a bit confusion and I do not well understand what means in panel a. How to relate microsite frequency with soil properties?

*We agree this figure and its description could use some revisions. Our caption says currently:* **Top panel (a) shows the model representation of soil microsite distribution (modified from Sihi et al., 2020, also see Eq. 13). Different shades indicate substrate concentration [Si], soil moisture (SoilMi), diffusion (Diffi) of solutes and gases, production (Prodi) and oxidation (Oxi) processes at each microsite.**

*We revised the caption, extensively, as follows:*
*"Top panel (a) shows the model representation of soil microsite distribution (modified from Sihi et al., 2020a, also see Eq. 14). The cylinder refers to the volume beneath the soil chambers. The intensity of different cylinder colors figure refers to rate of a process or the intensity of a concentration inside microsites in each theoretical cylinder, e.g., a dark color means a higher rate/intensity, and a light color means a lower rate/intensity for a given process. The 2D graph on the right refers to the probability density function of the rate of the process or intensity of the concentration*

*in the bulk soil. A wide distribution skewed to the right (dark line) implies higher bulk rates of the process or higher concentrations, and a narrow distribution skewed to the left (light line) implies lower bulk rates of the process or lower concentrations, of any of the following: solute concentration [$S_i$], gas concentration [$G_i$], soil moisture (SoilM$_i$), gas and solute diffusion (Diff$_i$), methane production (Prod$_i$), and methane oxidation (Ox$_i$)."*

*We also revised the figure from the original as follows:  Add an arrow on the x axis pointing towards the right, denoting increasing concentrations or rates.  Moved the light-colored probability-density function to the left of the dark-colored probability-density function and make it much narrower and signify less impact on bulk rates/concentrations compared to the dark-colored function (more impact on bulk rates/concentrations).*

Do substrate concentration, soil moisture, diffusivity of solute and gas present the similar pattern for one kind of microsite?

*The frequency distribution for the microsites is the same for all of these, according to Eq 14. But, the diffusivity of liquids is according to Eq 11 and 12; diffusivity of gasses according to Eq 8, 9, and 10. Here is a little more information on the microsites that is included in a new methods Section 2.4.3 devoted to microsites: We assumed that size of the microsites should be at least an order magnitude lower than the bulk soil measurements we had for soil methane fluxes. Using this logic, we decided that "diameter" of microsites should be in "mm" scale as the diameter of soil chambers we used are in "cm" scale (15.24 cm). Thus, we did the math to come up with the number of "total microsites" (i.e. 10000) under each fluxchamber such that the diameter of microsites meets our criteria, following Sihi et al. 2020a. This is why the frequency of microsites is the same for both low rate/intensities (light yellow line Fig 1a) and high rate/intensities (dark line Fig 1a) (see also Fig. 7 in manuscript and Fig. S10 in SI)."*

Why this figure links to Eq.13?

*This figure 1 should link to Eq 14; apologies for the confusion and thank you for the catch. It is corrected in the revision.*

Also in panel b, it would be more clear for readers if you could adjust it to a better shape or based on the clue of the present study. Can you try to improve the conceptual figure and clarify this in the legend?

*We removed the Air/Soil diagram at the top of this figure, and the word "solute diffusion" from the figure. Panel b should only represent the geochemical pathways that the model is representing, and we should rely on panel (a) to address diffusion. We hope that makes the content of both panels more understandable.*

*We revised the caption from the current version:  **Bottom panel (b) is the schematic of the microbial functional group-based model coupled with a diffusivity module (Microbial Model for Methane Dynamics-Dual Arrhenius and Michaelis Menten, M3D-DAMM) for simulating soil methane (CH$_4$) dynamics in field soils (Modified from Xu et al., 2015), where SOM = soil organic matter, CO$_2$ = carbon dioxide, DOC = dissolved organic carbon, H$^+$ is the hydronium ion, and H$_2$ = dihydrogen molecule.***
*Revised figure caption: Bottom panel (b) is the schematic of the microbial functional group-based model for simulating soil methane (CH$_4$) dynamics in field soils (modified from Xu et al., 2015). The schematic represents the decomposition of soil organic matter (SOM) and plant litter into carbon dioxide (CO$_2$) and dissolved organic matter (DOC); the production of acetate and hydronium ion (H$^+$) from decomposition and fermentation of DOC which also decreases pH, the production of acetate and hydronium ion (H$^+$) from homoacetogenesis which decreases pH; and the production of  dihydrogen ion (H$_2$) and CO$_2$ from decomposition of DOC. The intermediary products then have three possible non-mutually exclusive pathways (1) acetoclastic methanogenesis, which is the production of methane from aqueous acetate found in soil solutions, (2) hydrogenotrophic methanogenesis, which is the production of methane from hydrogen, and (3) methanotrophy, which is the oxidation of methane into carbon dioxide.*

Fig.3 and 4 The label of y-axis for soil moisture and oxygen should between 0-1 rather than 0-100, as the unit is V V-1. Otherwise, the unit should change to %.

*Thank you for pointing this out.  We adjusted the unit of the axes in Figs 3 and 4,*

*S8(g)(h)(i), and S10(i)(j) in the revision.*

Fig.4 and 6 The unit of CH4 emission should be uniform. Some of them are nmol m-2 S-1, while others are nmole m-2 S-1. Also the unite of acetate (Fig. 2).
*Thank you, we should use nmol and μmol (and not "mole"). Corrections are made to figures 2,3,4,7, and S5,S6,S7,S8,S9,S10 in the revision.*

[revised manuscript text omitted]

236 where $\text{Reaction}_{rate_i}$ (in $nM$ cm$^{-3}$ hr$^{-1}$) is rate of CH$_4$ production and/or consumption under variable substrate

237 concentrations. $\text{Biomass}_{func_i}$ ($nM$ cm$^{-3}$) represents microbial functional groups: aceticlastic methanogens,

238 hydrogenotrophic methanogens, and aerobic methanotrophs, respectively. Growth rates and substrate use

239 efficiencies of microbial functional groups are represented as $\text{GrowR}_{func_i}$ (hr$^{-1}$) and $\text{Efficiency}_{func_i}$ (unitless),

240 respectively (Table 1). The substrate limitation on CH$_4$ production is imposed by assuming a Michaelis-Menten

241 relationship with the half-saturation constants for CH$_4$ production and oxidation being $\text{KM}_{func_{1...n}}$ ($nM$ cm$^{-3}$).

242 Although minor contributions of iron dependent anaerobic CH$_4$ oxidation to net CH$_4$ emissions can be expected in

243 our study site (Ettwig et al., 2016), we did not represent this process here as anaerobic oxidation of CH$_4$ is still not

244 fully understood and it is generally low in most ecosystems.

245 The extent of change in $\text{Biomass}_{func_i}$ ($d\text{Biomass}_{func_i}$) is controlled by the balance between $\text{Growth}_{func_i}$ and

246 $\text{Death}_{func_i}$ following:

247 $$\frac{d\text{Biomass}_{func_i}}{dt_{func_i}} = \text{Growth}_{func_i} - \text{Death}_{func_i} \qquad (2)$$

248 $$\text{Growth}_{func_i} = \text{Efficiency}_{func_i} \times \text{Reaction}_{rate_i} \qquad (3)$$

249 where $\text{Growth}_{func_i}$ is calculated as a multiplicative function of $\text{Efficiency}_{func_i}$ and the $\text{Reaction}_{rate_i}$

250 $$\text{Death}_{func_i} = \text{DeadR}_{func_i} \times \text{Biomass}_{func_i} \qquad (4)$$

[revised manuscript text omitted]

Font: (Default) +Body (Times New Roman), 10 pt, Not Italic

| Page 3: [2] Formatted | Mayes, Melanie A. | 11/17/20 8:18:00 PM |
|---|---|---|

Font: (Default) +Body (Times New Roman), 10 pt

| Page 4: [3] Formatted | Mayes, Melanie A. | 11/17/20 8:18:00 PM |
|---|---|---|

Font: (Default) +Body (Times New Roman), 10 pt, Not Italic

| Page 4: [4] Formatted | Mayes, Melanie A. | 11/17/20 8:18:00 PM |
|---|---|---|

Font: (Default) +Body (Times New Roman), 10 pt

| Page 4: [5] Formatted | Mayes, Melanie A. | 11/17/20 8:18:00 PM |
|---|---|---|

Font: (Default) +Body (Times New Roman), 10 pt, Not Italic

| Page 4: [6] Formatted | Mayes, Melanie A. | 11/17/20 8:18:00 PM |
|---|---|---|

Font: (Default) +Body (Times New Roman), 10 pt

| Page 4: [7] Formatted | Mayes, Melanie A. | 11/17/20 8:18:00 PM |
|---|---|---|

Font: (Default) +Body (Times New Roman), 10 pt, Not Italic

| Page 4: [8] Formatted | Mayes, Melanie A. | 11/17/20 8:18:00 PM |
|---|---|---|

Font: (Default) +Body (Times New Roman), 10 pt

| Page 4: [9] Formatted | Mayes, Melanie A. | 11/17/20 8:18:00 PM |
|---|---|---|

Font: (Default) +Body (Times New Roman), 10 pt, Not Italic

| Page 4: [10] Formatted | Mayes, Melanie A. | 11/17/20 8:18:00 PM |
|---|---|---|

Font: (Default) +Body (Times New Roman), 10 pt

| Page 4: [11] Formatted | Mayes, Melanie A. | 11/17/20 8:18:00 PM |
|---|---|---|

Font: (Default) +Body (Times New Roman), 10 pt, Not Highlight

| Page 4: [12] Formatted | Mayes, Melanie A. | 11/17/20 8:18:00 PM |
|---|---|---|

Font: (Default) +Body (Times New Roman), 10 pt, Not Italic

| Page 4: [13] Formatted | Mayes, Melanie A. | 11/17/20 8:18:00 PM |
|---|---|---|

Font: (Default) +Body (Times New Roman), 10 pt

| Page 4: [14] Formatted | Mayes, Melanie A. | 11/17/20 8:18:00 PM |
|---|---|---|

Font: (Default) +Body (Times New Roman), 10 pt, Not Italic

| Page 4: [15] Formatted | Mayes, Melanie A. | 11/17/20 8:18:00 PM |
|---|---|---|

Font: (Default) +Body (Times New Roman), 10 pt

| Page 4: [16] Formatted | Mayes, Melanie A. | 11/17/20 8:18:00 PM |
|---|---|---|

Font: (Default) +Body (Times New Roman), 10 pt, Not Italic

| Page 4: [17] Formatted | Mayes, Melanie A. | 11/17/20 8:18:00 PM |
|---|---|---|

Font: (Default) +Body (Times New Roman), 10 pt

| Page 4: [18] Formatted | Mayes, Melanie A. | 11/17/20 8:18:00 PM |
|---|---|---|

Font: (Default) +Body (Times New Roman), 10 pt, Not Italic

| Page 4: [19] Formatted | Mayes, Melanie A. | 11/17/20 8:18:00 PM |
|---|---|---|

Font: (Default) +Body (Times New Roman), 10 pt, Not Italic, Superscript

| Page 4: [20] Formatted | Mayes, Melanie A. | 11/17/20 8:18:00 PM |
|---|---|---|

Font: (Default) +Body (Times New Roman), 10 pt, Not Italic

| Page 4: [21] Formatted | Mayes, Melanie A. | 11/17/20 8:18:00 PM |
|---|---|---|

Font: (Default) +Body (Times New Roman), 10 pt

| Page 4: [22] Formatted | Mayes, Melanie A. | 11/17/20 8:18:00 PM |
|---|---|---|

Font: (Default) +Body (Times New Roman), 10 pt, Not Italic

| Page 4: [23] Formatted | Mayes, Melanie A. | 11/17/20 8:18:00 PM |
|---|---|---|

Font: (Default) +Body (Times New Roman), 10 pt, Not Highlight

| Page 4: [24] Formatted | Mayes, Melanie A. | 11/17/20 8:18:00 PM |
|---|---|---|

Font: (Default) +Body (Times New Roman), 10 pt, Not Italic

| Page 4: [25] Formatted | Mayes, Melanie A. | 11/17/20 8:18:00 PM |
|---|---|---|

Font: (Default) +Body (Times New Roman), 10 pt

| Page 4: [26] Formatted | Mayes, Melanie A. | 11/17/20 8:18:00 PM |
|---|---|---|

Font: (Default) +Body (Times New Roman), 10 pt, Not Italic

| Page 4: [27] Formatted | Mayes, Melanie A. | 11/17/20 8:18:00 PM |
|---|---|---|

Font: (Default) +Body (Times New Roman), 10 pt

| Page 4: [28] Formatted | Mayes, Melanie A. | 11/17/20 8:18:00 PM |
|---|---|---|

Font: (Default) +Body (Times New Roman), 10 pt, Subscript

| Page 4: [29] Formatted | Mayes, Melanie A. | 11/17/20 8:18:00 PM |
|---|---|---|

Font: (Default) +Body (Times New Roman), 10 pt

| Page 4: [30] Formatted | Mayes, Melanie A. | 11/17/20 8:18:00 PM |
|---|---|---|

Font: (Default) +Body (Times New Roman), 10 pt

| Page 4: [31] Formatted | Mayes, Melanie A. | 11/17/20 8:18:00 PM |
|---|---|---|

Font: (Default) +Body (Times New Roman), 10 pt

| Page 4: [32] Formatted | Mayes, Melanie A. | 11/17/20 8:18:00 PM |
|---|---|---|

Font: (Default) +Body (Times New Roman), 10 pt

| Page 5: [33] Formatted | Mayes, Melanie A. | 11/17/20 8:18:00 PM |
|---|---|---|

Font: 10 pt

| Page 5: [33] Formatted | Mayes, Melanie A. | 11/17/20 8:18:00 PM |
|---|---|---|

Font: 10 pt

| Page 5: [33] Formatted | Mayes, Melanie A. | 11/17/20 8:18:00 PM |
|---|---|---|

Font: 10 pt

| Page 5: [33] Formatted | Mayes, Melanie A. | 11/17/20 8:18:00 PM |
|---|---|---|

Font: 10 pt

| Page 5: [33] Formatted | Mayes, Melanie A. | 11/17/20 8:18:00 PM |
|---|---|---|

Font: 10 pt

| Page 5: [34] Formatted | Mayes, Melanie A. | 11/17/20 8:18:00 PM |
|---|---|---|

Font: (Default) +Body (Times New Roman), 10 pt, Not Italic

| Page 5: [34] Formatted | Mayes, Melanie A. | 11/17/20 8:18:00 PM |
|---|---|---|

Font: (Default) +Body (Times New Roman), 10 pt, Not Italic

| Page 5: [34] Formatted | Mayes, Melanie A. | 11/17/20 8:18:00 PM |
|---|---|---|

Font: (Default) +Body (Times New Roman), 10 pt, Not Italic

| Page 5: [34] Formatted | Mayes, Melanie A. | 11/17/20 8:18:00 PM |
|---|---|---|

Font: (Default) +Body (Times New Roman), 10 pt, Not Italic

| Page 5: [34] Formatted | Mayes, Melanie A. | 11/17/20 8:18:00 PM |
|---|---|---|

Font: (Default) +Body (Times New Roman), 10 pt, Not Italic

| Page 5: [34] Formatted | Mayes, Melanie A. | 11/17/20 8:18:00 PM |
|---|---|---|

Font: (Default) +Body (Times New Roman), 10 pt, Not Italic

| Page 5: [34] Formatted | Mayes, Melanie A. | 11/17/20 8:18:00 PM |
|---|---|---|

Font: (Default) +Body (Times New Roman), 10 pt, Not Italic

| Page 5: [34] Formatted | Mayes, Melanie A. | 11/17/20 8:18:00 PM |
|---|---|---|

Font: (Default) +Body (Times New Roman), 10 pt, Not Italic

| Page 5: [34] Formatted | Mayes, Melanie A. | 11/17/20 8:18:00 PM |
|---|---|---|

Font: (Default) +Body (Times New Roman), 10 pt, Not Italic

| Page 5: [34] Formatted | Mayes, Melanie A. | 11/17/20 8:18:00 PM |
|---|---|---|

Font: (Default) +Body (Times New Roman), 10 pt, Not Italic

| Page 5: [34] Formatted | Mayes, Melanie A. | 11/17/20 8:18:00 PM |
|---|---|---|

Font: (Default) +Body (Times New Roman), 10 pt, Not Italic

| Page 5: [34] Formatted | Mayes, Melanie A. | 11/17/20 8:18:00 PM |
|---|---|---|

Font: (Default) +Body (Times New Roman), 10 pt, Not Italic

| Page 5: [34] Formatted | Mayes, Melanie A. | 11/17/20 8:18:00 PM |
|---|---|---|

Font: (Default) +Body (Times New Roman), 10 pt, Not Italic

| Page 5: [34] Formatted | Mayes, Melanie A. | 11/17/20 8:18:00 PM |
|---|---|---|

Font: (Default) +Body (Times New Roman), 10 pt, Not Italic

| Page 5: [34] Formatted | Mayes, Melanie A. | 11/17/20 8:18:00 PM |
|---|---|---|

Font: (Default) +Body (Times New Roman), 10 pt, Not Italic

| Page 5: [35] Formatted | Mayes, Melanie A. | 11/17/20 8:18:00 PM |
|---|---|---|

Font: 10 pt

| Page 5: [35] Formatted | Mayes, Melanie A. | 11/17/20 8:18:00 PM |
|---|---|---|

Font: 10 pt

| Page 5: [35] Formatted | Mayes, Melanie A. | 11/17/20 8:18:00 PM |
|---|---|---|

Font: 10 pt

| Page 5: [35] Formatted | Mayes, Melanie A. | 11/17/20 8:18:00 PM |
|---|---|---|

Font: 10 pt

| Page 5: [36] Formatted | Mayes, Melanie A. | 11/17/20 8:18:00 PM |
|---|---|---|

Font: 10 pt

| Page 5: [36] Formatted | Mayes, Melanie A. | 11/17/20 8:18:00 PM |
|---|---|---|

Font: 10 pt

| Page 5: [36] Formatted | Mayes, Melanie A. | 11/17/20 8:18:00 PM |
|---|---|---|

Font: 10 pt

| Page 5: [36] Formatted | Mayes, Melanie A. | 11/17/20 8:18:00 PM |
|---|---|---|

Font: 10 pt

| Page 5: [36] Formatted | Mayes, Melanie A. | 11/17/20 8:18:00 PM |
|---|---|---|

Font: 10 pt

| Page 6: [37] Formatted | Mayes, Melanie A. | 11/17/20 8:18:00 PM |
|---|---|---|

Font: 10 pt

| Page 6: [37] Formatted | Mayes, Melanie A. | 11/17/20 8:18:00 PM |
|---|---|---|

Font: 10 pt

| Page 6: [37] Formatted | Mayes, Melanie A. | 11/17/20 8:18:00 PM |
|---|---|---|

Font: 10 pt

| Page 6: [37] Formatted | Mayes, Melanie A. | 11/17/20 8:18:00 PM |
|---|---|---|

Font: 10 pt

| Page 6: [38] Formatted | Mayes, Melanie A. | 11/17/20 8:18:00 PM |
|---|---|---|

Font: 10 pt

| Page 6: [38] Formatted | Mayes, Melanie A. | 11/17/20 8:18:00 PM |
|---|---|---|

Font: 10 pt

| Page 6: [38] Formatted | Mayes, Melanie A. | 11/17/20 8:18:00 PM |
|---|---|---|

Font: 10 pt

| Page 6: [38] Formatted | Mayes, Melanie A. | 11/17/20 8:18:00 PM |
|---|---|---|

Font: 10 pt

| Page 6: [39] Deleted | Mayes, Melanie A. | 11/17/20 8:29:00 PM |
|---|---|---|

| Page 6: [39] Deleted | Mayes, Melanie A. | 11/17/20 8:29:00 PM |
|---|---|---|

| Page 6: [40] Formatted | Mayes, Melanie A. | 11/17/20 8:18:00 PM |
|---|---|---|

Font: 10 pt

| Page 6: [40] Formatted | Mayes, Melanie A. | 11/17/20 8:18:00 PM |
|---|---|---|

Font: 10 pt

| Page 6: [40] Formatted | Mayes, Melanie A. | 11/17/20 8:18:00 PM |
|---|---|---|

Font: 10 pt

| Page 6: [40] Formatted | Mayes, Melanie A. | 11/17/20 8:18:00 PM |
|---|---|---|

Font: 10 pt

| Page 6: [41] Formatted | Mayes, Melanie A. | 11/17/20 8:18:00 PM |
|---|---|---|

Font: 10 pt

| Page 6: [41] Formatted | Mayes, Melanie A. | 11/17/20 8:18:00 PM |
|---|---|---|

Font: 10 pt

| Page 6: [41] Formatted | Mayes, Melanie A. | 11/17/20 8:18:00 PM |
|---|---|---|

Font: 10 pt

| Page 6: [41] Formatted | Mayes, Melanie A. | 11/17/20 8:18:00 PM |
|---|---|---|

Font: 10 pt

| Page 6: [42] Formatted | Mayes, Melanie A. | 11/17/20 8:18:00 PM |
|---|---|---|

Font: (Default) +Body (Times New Roman), 10 pt

| Page 6: [43] Formatted | Mayes, Melanie A. | 11/17/20 8:18:00 PM |
|---|---|---|

Font: 10 pt

| Page 6: [43] Formatted | Mayes, Melanie A. | 11/17/20 8:18:00 PM |
|---|---|---|

Font: 10 pt

| Page 6: [43] Formatted | Mayes, Melanie A. | 11/17/20 8:18:00 PM |
|---|---|---|

Font: 10 pt

| Page 6: [43] Formatted | Mayes, Melanie A. | 11/17/20 8:18:00 PM |
|---|---|---|

Font: 10 pt

| Page 6: [44] Deleted | Mayes, Melanie A. | 11/17/20 8:30:00 PM |
|---|---|---|

| Page 6: [44] Deleted | Mayes, Melanie A. | 11/17/20 8:30:00 PM |
|---|---|---|

| Page 6: [45] Formatted | Mayes, Melanie A. | 11/17/20 8:18:00 PM |
|---|---|---|

Font: 10 pt

| Page 6: [45] Formatted | Mayes, Melanie A. | 11/17/20 8:18:00 PM |
|---|---|---|

Font: 10 pt

| Page 6: [45] Formatted | Mayes, Melanie A. | 11/17/20 8:18:00 PM |
|---|---|---|

Font: 10 pt

| Page 6: [45] Formatted | Mayes, Melanie A. | 11/17/20 8:18:00 PM |
|---|---|---|

Font: 10 pt

| Page 6: [46] Formatted | Mayes, Melanie A. | 11/17/20 8:18:00 PM |
|---|---|---|

Font: 10 pt

| Page 6: [46] Formatted | Mayes, Melanie A. | 11/17/20 8:18:00 PM |
|---|---|---|

Font: 10 pt

| Page 6: [46] Formatted | Mayes, Melanie A. | 11/17/20 8:18:00 PM |
|---|---|---|

Font: 10 pt

| Page 6: [46] Formatted | Mayes, Melanie A. | 11/17/20 8:18:00 PM |
|---|---|---|

Font: 10 pt

| Page 6: [47] Formatted | Mayes, Melanie A. | 11/17/20 8:18:00 PM |
|---|---|---|

Font: 10 pt

| Page 6: [47] Formatted | Mayes, Melanie A. | 11/17/20 8:18:00 PM |
|---|---|---|

Font: 10 pt

| Page 6: [47] Formatted | Mayes, Melanie A. | 11/17/20 8:18:00 PM |
|---|---|---|

Font: 10 pt

| Page 6: [47] Formatted | Mayes, Melanie A. | 11/17/20 8:18:00 PM |
|---|---|---|

Font: 10 pt

| Page 6: [48] Formatted | Mayes, Melanie A. | 11/17/20 8:18:00 PM |
|---|---|---|

Font: 10 pt

| Page 6: [48] Formatted | Mayes, Melanie A. | 11/17/20 8:18:00 PM |
|---|---|---|

Font: 10 pt

| Page 6: [48] Formatted | Mayes, Melanie A. | 11/17/20 8:18:00 PM |
|---|---|---|

Font: 10 pt

| Page 6: [48] Formatted | Mayes, Melanie A. | 11/17/20 8:18:00 PM |
|---|---|---|

Font: 10 pt

| Page 6: [49] Formatted | Mayes, Melanie A. | 11/17/20 8:18:00 PM |
|---|---|---|

Font: 10 pt

| Page 6: [49] Formatted | Mayes, Melanie A. | 11/17/20 8:18:00 PM |
|---|---|---|

Font: 10 pt

| Page 6: [49] Formatted | Mayes, Melanie A. | 11/17/20 8:18:00 PM |
|---|---|---|

Font: 10 pt

| Page 6: [49] Formatted | Mayes, Melanie A. | 11/17/20 8:18:00 PM |
|---|---|---|

Font: 10 pt

| Page 6: [49] Formatted | Mayes, Melanie A. | 11/17/20 8:18:00 PM |
|---|---|---|

Font: 10 pt

| Page 6: [49] Formatted | Mayes, Melanie A. | 11/17/20 8:18:00 PM |
|---|---|---|

Font: 10 pt

| Page 6: [49] Formatted | Mayes, Melanie A. | 11/17/20 8:18:00 PM |
|---|---|---|

Font: 10 pt

| Page 6: [50] Formatted | Mayes, Melanie A. | 11/17/20 8:18:00 PM |
|---|---|---|

Font: 10 pt

| Page 6: [50] Formatted | Mayes, Melanie A. | 11/17/20 8:18:00 PM |
|---|---|---|

Font: 10 pt

| Page 6: [50] Formatted | Mayes, Melanie A. | 11/17/20 8:18:00 PM |
|---|---|---|

Font: 10 pt

| Page 6: [50] Formatted | Mayes, Melanie A. | 11/17/20 8:18:00 PM |
|---|---|---|

Font: 10 pt

| Page 6: [50] Formatted | Mayes, Melanie A. | 11/17/20 8:18:00 PM |
|---|---|---|

Font: 10 pt

| Page 6: [50] Formatted | Mayes, Melanie A. | 11/17/20 8:18:00 PM |
|---|---|---|

Font: 10 pt

| Page 6: [50] Formatted | Mayes, Melanie A. | 11/17/20 8:18:00 PM |

Font: 10 pt

| Page 6: [51] Formatted | Mayes, Melanie A. | 11/17/20 8:18:00 PM |

Font: 10 pt

| Page 6: [51] Formatted | Mayes, Melanie A. | 11/17/20 8:18:00 PM |

Font: 10 pt

| Page 6: [51] Formatted | Mayes, Melanie A. | 11/17/20 8:18:00 PM |

Font: 10 pt

| Page 6: [52] Formatted | Mayes, Melanie A. | 11/17/20 8:18:00 PM |

Font: 10 pt

| Page 6: [52] Formatted | Mayes, Melanie A. | 11/17/20 8:18:00 PM |

Font: 10 pt

| Page 6: [52] Formatted | Mayes, Melanie A. | 11/17/20 8:18:00 PM |

Font: 10 pt

| Page 6: [52] Formatted | Mayes, Melanie A. | 11/17/20 8:18:00 PM |

Font: 10 pt

| Page 6: [52] Formatted | Mayes, Melanie A. | 11/17/20 8:18:00 PM |

Font: 10 pt

| Page 6: [52] Formatted | Mayes, Melanie A. | 11/17/20 8:18:00 PM |

Font: 10 pt

| Page 6: [52] Formatted | Mayes, Melanie A. | 11/17/20 8:18:00 PM |

Font: 10 pt

| Page 6: [52] Formatted | Mayes, Melanie A. | 11/17/20 8:18:00 PM |

Font: 10 pt

| Page 6: [53] Formatted | Mayes, Melanie A. | 11/17/20 8:18:00 PM |

Font: 10 pt

| Page 6: [53] Formatted | Mayes, Melanie A. | 11/17/20 8:18:00 PM |

Font: 10 pt

| Page 6: [53] Formatted | Mayes, Melanie A. | 11/17/20 8:18:00 PM |

Font: 10 pt

| Page 6: [53] Formatted | Mayes, Melanie A. | 11/17/20 8:18:00 PM |

Font: 10 pt

| Page 6: [54] Formatted | Mayes, Melanie A. | 11/17/20 8:18:00 PM |

Font: 10 pt

| Page 6: [54] Formatted | Mayes, Melanie A. | 11/17/20 8:18:00 PM |

Font: 10 pt

| Page 6: [54] Formatted | Mayes, Melanie A. | 11/17/20 8:18:00 PM |

Font: 10 pt

| Page 6: [54] Formatted | Mayes, Melanie A. | 11/17/20 8:18:00 PM |

Font: 10 pt

| Page 6: [55] Formatted | Mayes, Melanie A. | 11/17/20 8:18:00 PM |

Font: 10 pt

| Page 6: [55] Formatted | Mayes, Melanie A. | 11/17/20 8:18:00 PM |

Font: 10 pt

| Page 6: [55] Formatted | Mayes, Melanie A. | 11/17/20 8:18:00 PM |

Font: 10 pt

| Page 6: [55] Formatted | Mayes, Melanie A. | 11/17/20 8:18:00 PM |

Font: 10 pt

| Page 6: [56] Formatted | Mayes, Melanie A. | 11/17/20 8:18:00 PM |

Font: 10 pt

| Page 6: [56] Formatted | Mayes, Melanie A. | 11/17/20 8:18:00 PM |

Font: 10 pt

| Page 6: [56] Formatted | Mayes, Melanie A. | 11/17/20 8:18:00 PM |

Font: 10 pt

| Page 6: [56] Formatted | Mayes, Melanie A. | 11/17/20 8:18:00 PM |

Font: 10 pt

| Page 6: [57] Formatted | Mayes, Melanie A. | 11/17/20 8:18:00 PM |

Font: 10 pt

| Page 6: [57] Formatted | Mayes, Melanie A. | 11/17/20 8:18:00 PM |

Font: 10 pt

| Page 6: [57] Formatted | Mayes, Melanie A. | 11/17/20 8:18:00 PM |

Font: 10 pt

| Page 6: [57] Formatted | Mayes, Melanie A. | 11/17/20 8:18:00 PM |

Font: 10 pt

| Page 6: [57] Formatted | Mayes, Melanie A. | 11/17/20 8:18:00 PM |

Font: 10 pt

| Page 6: [57] Formatted | Mayes, Melanie A. | 11/17/20 8:18:00 PM |

Font: 10 pt

| Page 6: [57] Formatted | Mayes, Melanie A. | 11/17/20 8:18:00 PM |

Font: 10 pt

| Page 6: [57] Formatted | Mayes, Melanie A. | 11/17/20 8:18:00 PM |

Font: 10 pt

| Page 6: [58] Formatted | Mayes, Melanie A. | 11/17/20 8:18:00 PM |

Font: 10 pt

| Page 6: [58] Formatted | Mayes, Melanie A. | 11/17/20 8:18:00 PM |

Font: 10 pt

| Page 6: [58] Formatted | Mayes, Melanie A. | 11/17/20 8:18:00 PM |

Font: 10 pt

| Page 6: [58] Formatted | Mayes, Melanie A. | 11/17/20 8:18:00 PM |

Font: 10 pt

| Page 6: [58] Formatted | Mayes, Melanie A. | 11/17/20 8:18:00 PM |

Font: 10 pt

| Page 6: [59] Formatted | Mayes, Melanie A. | 11/17/20 8:18:00 PM |

Font: 10 pt

| Page 6: [59] Formatted | Mayes, Melanie A. | 11/17/20 8:18:00 PM |

Font: 10 pt

| Page 6: [59] Formatted | Mayes, Melanie A. | 11/17/20 8:18:00 PM |

Font: 10 pt

| Page 6: [59] Formatted | Mayes, Melanie A. | 11/17/20 8:18:00 PM |

Font: 10 pt

| Page 6: [60] Formatted | Mayes, Melanie A. | 11/17/20 8:18:00 PM |

Font: 10 pt

| Page 6: [60] Formatted | Mayes, Melanie A. | 11/17/20 8:18:00 PM |

Font: 10 pt

| Page 6: [60] Formatted | Mayes, Melanie A. | 11/17/20 8:18:00 PM |

Font: 10 pt

| Page 6: [60] Formatted | Mayes, Melanie A. | 11/17/20 8:18:00 PM |

Font: 10 pt

| Page 6: [61] Formatted | Mayes, Melanie A. | 11/17/20 8:18:00 PM |

Font: 10 pt

| Page 6: [62] Formatted | Mayes, Melanie A. | 11/17/20 8:18:00 PM |

Font: (Default) +Body (Times New Roman), 10 pt

| Page 6: [62] Formatted | Mayes, Melanie A. | 11/17/20 8:18:00 PM |

Font: (Default) +Body (Times New Roman), 10 pt

| Page 6: [63] Formatted | Mayes, Melanie A. | 11/17/20 8:18:00 PM |

Font: (Default) +Body (Times New Roman), 10 pt

| Page 6: [64] Formatted | Mayes, Melanie A. | 11/17/20 8:18:00 PM |

Font: 10 pt

| Page 6: [65] Formatted | Mayes, Melanie A. | 11/17/20 8:18:00 PM |

Font: 10 pt

| Page 6: [65] Formatted | Mayes, Melanie A. | 11/17/20 8:18:00 PM |

Font: 10 pt

| Page 6: [65] Formatted | Mayes, Melanie A. | 11/17/20 8:18:00 PM |

Font: 10 pt

| Page 6: [65] Formatted | Mayes, Melanie A. | 11/17/20 8:18:00 PM |

Font: 10 pt

| Page 6: [65] Formatted | Mayes, Melanie A. | 11/17/20 8:18:00 PM |

Font: 10 pt

| Page 6: [65] Formatted | Mayes, Melanie A. | 11/17/20 8:18:00 PM |

Font: 10 pt

| Page 6: [65] Formatted | Mayes, Melanie A. | 11/17/20 8:18:00 PM |

Font: 10 pt

| Page 6: [65] Formatted | Mayes, Melanie A. | 11/17/20 8:18:00 PM |

Font: 10 pt

| Page 6: [65] Formatted | Mayes, Melanie A. | 11/17/20 8:18:00 PM |

Font: 10 pt

| Page 6: [66] Formatted | Mayes, Melanie A. | 11/17/20 8:18:00 PM |

Font: 10 pt

| Page 6: [67] Formatted | Mayes, Melanie A. | 11/17/20 8:18:00 PM |

Font: 10 pt

| Page 6: [67] Formatted | Mayes, Melanie A. | 11/17/20 8:18:00 PM |

Font: 10 pt

| Page 6: [67] Formatted | Mayes, Melanie A. | 11/17/20 8:18:00 PM |

Font: 10 pt

| Page 6: [67] Formatted | Mayes, Melanie A. | 11/17/20 8:18:00 PM |

Font: 10 pt

| Page 6: [67] Formatted | Mayes, Melanie A. | 11/17/20 8:18:00 PM |

Font: 10 pt

| Page 6: [67] Formatted | Mayes, Melanie A. | 11/17/20 8:18:00 PM |

Font: 10 pt

| Page 6: [67] Formatted | Mayes, Melanie A. | 11/17/20 8:18:00 PM |

Font: 10 pt

| Page 6: [67] Formatted | Mayes, Melanie A. | 11/17/20 8:18:00 PM |

Font: 10 pt

| Page 6: [67] Formatted | Mayes, Melanie A. | 11/17/20 8:18:00 PM |

Font: 10 pt

| Page 6: [67] Formatted | Mayes, Melanie A. | 11/17/20 8:18:00 PM |

Font: 10 pt

| Page 6: [67] Formatted | Mayes, Melanie A. | 11/17/20 8:18:00 PM |

Font: 10 pt

| Page 6: [67] Formatted | Mayes, Melanie A. | 11/17/20 8:18:00 PM |

Font: 10 pt

| Page 6: [67] Formatted | Mayes, Melanie A. | 11/17/20 8:18:00 PM |

Font: 10 pt

| Page 6: [67] Formatted | Mayes, Melanie A. | 11/17/20 8:18:00 PM |

Font: 10 pt

| Page 6: [67] Formatted | Mayes, Melanie A. | 11/17/20 8:18:00 PM |
|---|---|---|

Font: 10 pt

| Page 6: [67] Formatted | Mayes, Melanie A. | 11/17/20 8:18:00 PM |
|---|---|---|

Font: 10 pt

| Page 6: [67] Formatted | Mayes, Melanie A. | 11/17/20 8:18:00 PM |
|---|---|---|

Font: 10 pt

| Page 6: [67] Formatted | Mayes, Melanie A. | 11/17/20 8:18:00 PM |
|---|---|---|

Font: 10 pt

| Page 6: [67] Formatted | Mayes, Melanie A. | 11/17/20 8:18:00 PM |
|---|---|---|

Font: 10 pt

| Page 6: [68] Formatted | Mayes, Melanie A. | 11/17/20 8:18:00 PM |
|---|---|---|

Font: 10 pt

| Page 6: [69] Formatted | Mayes, Melanie A. | 11/17/20 8:18:00 PM |
|---|---|---|

Font: 10 pt

| Page 6: [69] Formatted | Mayes, Melanie A. | 11/17/20 8:18:00 PM |
|---|---|---|

Font: 10 pt

| Page 6: [69] Formatted | Mayes, Melanie A. | 11/17/20 8:18:00 PM |
|---|---|---|

Font: 10 pt

| Page 6: [69] Formatted | Mayes, Melanie A. | 11/17/20 8:18:00 PM |
|---|---|---|

Font: 10 pt

| Page 6: [69] Formatted | Mayes, Melanie A. | 11/17/20 8:18:00 PM |
|---|---|---|

Font: 10 pt

| Page 6: [70] Deleted | Mayes, Melanie A. | 11/15/20 11:25:00 AM |
|---|---|---|

| Page 6: [70] Deleted | Mayes, Melanie A. | 11/15/20 11:25:00 AM |
|---|---|---|

| Page 6: [71] Formatted | Mayes, Melanie A. | 11/17/20 8:18:00 PM |
|---|---|---|

Font: (Default) +Body (Times New Roman), 10 pt

| Page 6: [72] Formatted | Mayes, Melanie A. | 11/17/20 8:18:00 PM |
|---|---|---|

Font: 10 pt

| Page 6: [72] Formatted | Mayes, Melanie A. | 11/17/20 8:18:00 PM |
|---|---|---|

Font: 10 pt

| Page 6: [72] Formatted | Mayes, Melanie A. | 11/17/20 8:18:00 PM |
|---|---|---|

Font: 10 pt

| Page 6: [72] Formatted | Mayes, Melanie A. | 11/17/20 8:18:00 PM |
|---|---|---|

Font: 10 pt

| Page 6: [72] Formatted | Mayes, Melanie A. | 11/17/20 8:18:00 PM |

Font: 10 pt

| Page 6: [73] Formatted | Mayes, Melanie A. | 11/17/20 8:18:00 PM |

Font: 10 pt

| Page 6: [73] Formatted | Mayes, Melanie A. | 11/17/20 8:18:00 PM |

Font: 10 pt

| Page 6: [73] Formatted | Mayes, Melanie A. | 11/17/20 8:18:00 PM |

Font: 10 pt

| Page 6: [73] Formatted | Mayes, Melanie A. | 11/17/20 8:18:00 PM |

Font: 10 pt

| Page 6: [74] Formatted | Mayes, Melanie A. | 11/17/20 8:18:00 PM |

Font: 10 pt

| Page 6: [74] Formatted | Mayes, Melanie A. | 11/17/20 8:18:00 PM |

Font: 10 pt

| Page 6: [74] Formatted | Mayes, Melanie A. | 11/17/20 8:18:00 PM |

Font: 10 pt

| Page 6: [74] Formatted | Mayes, Melanie A. | 11/17/20 8:18:00 PM |

Font: 10 pt

| Page 6: [75] Formatted | Mayes, Melanie A. | 11/17/20 8:18:00 PM |

Font: 10 pt

| Page 6: [75] Formatted | Mayes, Melanie A. | 11/17/20 8:18:00 PM |

Font: 10 pt

| Page 6: [75] Formatted | Mayes, Melanie A. | 11/17/20 8:18:00 PM |

Font: 10 pt

| Page 6: [75] Formatted | Mayes, Melanie A. | 11/17/20 8:18:00 PM |

Font: 10 pt

| Page 6: [76] Formatted | Mayes, Melanie A. | 11/17/20 8:18:00 PM |

Font: 10 pt

| Page 6: [76] Formatted | Mayes, Melanie A. | 11/17/20 8:18:00 PM |

Font: 10 pt

| Page 6: [76] Formatted | Mayes, Melanie A. | 11/17/20 8:18:00 PM |

Font: 10 pt

| Page 6: [76] Formatted | Mayes, Melanie A. | 11/17/20 8:18:00 PM |

Font: 10 pt

| Page 6: [76] Formatted | Mayes, Melanie A. | 11/17/20 8:18:00 PM |

Font: 10 pt

| Page 6: [77] Formatted | Mayes, Melanie A. | 11/17/20 8:18:00 PM |

Font: 10 pt

| Page 6: [77] Formatted | Mayes, Melanie A. | 11/17/20 8:18:00 PM |

Font: 10 pt

| Page 6: [77] Formatted | Mayes, Melanie A. | 11/17/20 8:18:00 PM |
|---|---|---|

Font: 10 pt

| Page 7: [78] Formatted | Mayes, Melanie A. | 11/17/20 8:18:00 PM |
|---|---|---|

Font: 10 pt

| Page 7: [78] Formatted | Mayes, Melanie A. | 11/17/20 8:18:00 PM |
|---|---|---|

Font: 10 pt

| Page 7: [78] Formatted | Mayes, Melanie A. | 11/17/20 8:18:00 PM |
|---|---|---|

Font: 10 pt

| Page 7: [78] Formatted | Mayes, Melanie A. | 11/17/20 8:18:00 PM |
|---|---|---|

Font: 10 pt

| Page 7: [79] Formatted | Mayes, Melanie A. | 11/17/20 8:18:00 PM |
|---|---|---|

Font: (Default) +Body (Times New Roman), 10 pt

| Page 7: [80] Formatted | Mayes, Melanie A. | 11/17/20 8:18:00 PM |
|---|---|---|

Font: 10 pt

| Page 7: [80] Formatted | Mayes, Melanie A. | 11/17/20 8:18:00 PM |
|---|---|---|

Font: 10 pt

| Page 7: [80] Formatted | Mayes, Melanie A. | 11/17/20 8:18:00 PM |
|---|---|---|

Font: 10 pt

| Page 7: [81] Formatted | Mayes, Melanie A. | 11/17/20 8:18:00 PM |
|---|---|---|

Font: 10 pt

| Page 7: [81] Formatted | Mayes, Melanie A. | 11/17/20 8:18:00 PM |
|---|---|---|

Font: 10 pt

| Page 7: [81] Formatted | Mayes, Melanie A. | 11/17/20 8:18:00 PM |
|---|---|---|

Font: 10 pt

| Page 7: [82] Formatted | Mayes, Melanie A. | 11/17/20 8:18:00 PM |
|---|---|---|

Font: 10 pt

| Page 7: [82] Formatted | Mayes, Melanie A. | 11/17/20 8:18:00 PM |
|---|---|---|

Font: 10 pt

| Page 7: [82] Formatted | Mayes, Melanie A. | 11/17/20 8:18:00 PM |
|---|---|---|

Font: 10 pt

| Page 7: [83] Formatted | Mayes, Melanie A. | 11/17/20 8:18:00 PM |
|---|---|---|

Font: 10 pt

| Page 7: [83] Formatted | Mayes, Melanie A. | 11/17/20 8:18:00 PM |
|---|---|---|

Font: 10 pt

| Page 7: [83] Formatted | Mayes, Melanie A. | 11/17/20 8:18:00 PM |
|---|---|---|

Font: 10 pt

| Page 7: [84] Formatted | Mayes, Melanie A. | 11/17/20 8:18:00 PM |
|---|---|---|

Font: 10 pt

| Page 7: [85] Formatted | Mayes, Melanie A. | 11/17/20 8:18:00 PM |
|---|---|---|

Font: 10 pt

**Page 7: [85] Formatted**      **Mayes, Melanie A.**      **11/17/20 8:18:00 PM**

Font: 10 pt

**Page 7: [85] Formatted**      **Mayes, Melanie A.**      **11/17/20 8:18:00 PM**

Font: 10 pt

**Page 7: [86] Formatted**      **Mayes, Melanie A.**      **11/17/20 8:18:00 PM**

Font: 10 pt

**Page 7: [86] Formatted**      **Mayes, Melanie A.**      **11/17/20 8:18:00 PM**

Font: 10 pt

**Page 7: [86] Formatted**      **Mayes, Melanie A.**      **11/17/20 8:18:00 PM**

Font: 10 pt

**Page 7: [86] Formatted**      **Mayes, Melanie A.**      **11/17/20 8:18:00 PM**

Font: 10 pt

**Page 7: [86] Formatted**      **Mayes, Melanie A.**      **11/17/20 8:18:00 PM**

Font: 10 pt

**Page 7: [86] Formatted**      **Mayes, Melanie A.**      **11/17/20 8:18:00 PM**

Font: 10 pt

**Page 7: [86] Formatted**      **Mayes, Melanie A.**      **11/17/20 8:18:00 PM**

Font: 10 pt

**Page 7: [86] Formatted**      **Mayes, Melanie A.**      **11/17/20 8:18:00 PM**

Font: 10 pt

**Page 7: [86] Formatted**      **Mayes, Melanie A.**      **11/17/20 8:18:00 PM**

Font: 10 pt

**Page 7: [86] Formatted**      **Mayes, Melanie A.**      **11/17/20 8:18:00 PM**

Font: 10 pt

**Page 7: [86] Formatted**      **Mayes, Melanie A.**      **11/17/20 8:18:00 PM**

Font: 10 pt

**Page 7: [86] Formatted**      **Mayes, Melanie A.**      **11/17/20 8:18:00 PM**

Font: 10 pt

**Page 7: [87] Formatted**      **Mayes, Melanie A.**      **11/17/20 8:18:00 PM**

Font: 10 pt

**Page 7: [87] Formatted**      **Mayes, Melanie A.**      **11/17/20 8:18:00 PM**

Font: 10 pt

**Page 7: [87] Formatted**      **Mayes, Melanie A.**      **11/17/20 8:18:00 PM**

Font: 10 pt

**Page 7: [88] Formatted**      **Mayes, Melanie A.**      **11/17/20 8:18:00 PM**

Font: 10 pt

**Page 7: [88] Formatted**      **Mayes, Melanie A.**      **11/17/20 8:18:00 PM**

Font: 10 pt

| Page 7: [88] Formatted | Mayes, Melanie A. | 11/17/20 8:18:00 PM |

Font: 10 pt

| Page 7: [89] Formatted | Mayes, Melanie A. | 11/17/20 8:18:00 PM |

Font: 10 pt

| Page 7: [89] Formatted | Mayes, Melanie A. | 11/17/20 8:18:00 PM |

Font: 10 pt

| Page 7: [89] Formatted | Mayes, Melanie A. | 11/17/20 8:18:00 PM |

Font: 10 pt

| Page 7: [89] Formatted | Mayes, Melanie A. | 11/17/20 8:18:00 PM |

Font: 10 pt

| Page 7: [89] Formatted | Mayes, Melanie A. | 11/17/20 8:18:00 PM |

Font: 10 pt

| Page 7: [89] Formatted | Mayes, Melanie A. | 11/17/20 8:18:00 PM |

Font: 10 pt

| Page 7: [89] Formatted | Mayes, Melanie A. | 11/17/20 8:18:00 PM |

Font: 10 pt

| Page 7: [90] Formatted | Mayes, Melanie A. | 11/17/20 8:18:00 PM |

Font: 10 pt

| Page 7: [91] Formatted | Mayes, Melanie A. | 11/17/20 8:18:00 PM |

Font: (Default) +Body (Times New Roman), 10 pt

| Page 7: [92] Formatted | Mayes, Melanie A. | 11/17/20 8:18:00 PM |

Font: 10 pt

| Page 7: [92] Formatted | Mayes, Melanie A. | 11/17/20 8:18:00 PM |

Font: 10 pt

| Page 7: [92] Formatted | Mayes, Melanie A. | 11/17/20 8:18:00 PM |

Font: 10 pt

| Page 7: [92] Formatted | Mayes, Melanie A. | 11/17/20 8:18:00 PM |

Font: 10 pt

| Page 7: [92] Formatted | Mayes, Melanie A. | 11/17/20 8:18:00 PM |

Font: 10 pt

| Page 7: [92] Formatted | Mayes, Melanie A. | 11/17/20 8:18:00 PM |

Font: 10 pt

| Page 7: [92] Formatted | Mayes, Melanie A. | 11/17/20 8:18:00 PM |

Font: 10 pt

| Page 7: [92] Formatted | Mayes, Melanie A. | 11/17/20 8:18:00 PM |

Font: 10 pt

| Page 7: [92] Formatted | Mayes, Melanie A. | 11/17/20 8:18:00 PM |

Font: 10 pt

| Page 7: [92] Formatted | Mayes, Melanie A. | 11/17/20 8:18:00 PM |
|---|---|---|

Font: 10 pt

| Page 7: [93] Formatted | Mayes, Melanie A. | 11/17/20 8:18:00 PM |
|---|---|---|

Font: 10 pt

| Page 7: [93] Formatted | Mayes, Melanie A. | 11/17/20 8:18:00 PM |
|---|---|---|

Font: 10 pt

| Page 7: [93] Formatted | Mayes, Melanie A. | 11/17/20 8:18:00 PM |
|---|---|---|

Font: 10 pt

| Page 7: [93] Formatted | Mayes, Melanie A. | 11/17/20 8:18:00 PM |
|---|---|---|

Font: 10 pt

| Page 7: [93] Formatted | Mayes, Melanie A. | 11/17/20 8:18:00 PM |
|---|---|---|

Font: 10 pt

| Page 7: [93] Formatted | Mayes, Melanie A. | 11/17/20 8:18:00 PM |
|---|---|---|

Font: 10 pt

| Page 7: [94] Formatted | Mayes, Melanie A. | 11/17/20 8:18:00 PM |
|---|---|---|

Font: 10 pt

| Page 7: [94] Formatted | Mayes, Melanie A. | 11/17/20 8:18:00 PM |
|---|---|---|

Font: 10 pt

| Page 7: [94] Formatted | Mayes, Melanie A. | 11/17/20 8:18:00 PM |
|---|---|---|

Font: 10 pt

| Page 7: [95] Formatted | Mayes, Melanie A. | 11/17/20 8:18:00 PM |
|---|---|---|

Font: 10 pt

| Page 7: [95] Formatted | Mayes, Melanie A. | 11/17/20 8:18:00 PM |
|---|---|---|

Font: 10 pt

| Page 7: [95] Formatted | Mayes, Melanie A. | 11/17/20 8:18:00 PM |
|---|---|---|

Font: 10 pt

| Page 7: [95] Formatted | Mayes, Melanie A. | 11/17/20 8:18:00 PM |
|---|---|---|

Font: 10 pt

| Page 7: [96] Formatted | Mayes, Melanie A. | 11/17/20 8:18:00 PM |
|---|---|---|

Font: 10 pt

| Page 7: [96] Formatted | Mayes, Melanie A. | 11/17/20 8:18:00 PM |
|---|---|---|

Font: 10 pt

| Page 7: [96] Formatted | Mayes, Melanie A. | 11/17/20 8:18:00 PM |
|---|---|---|

Font: 10 pt

| Page 7: [96] Formatted | Mayes, Melanie A. | 11/17/20 8:18:00 PM |
|---|---|---|

Font: 10 pt

| Page 7: [97] Formatted | Mayes, Melanie A. | 11/17/20 8:18:00 PM |
|---|---|---|

Font: 10 pt

| Page 7: [97] Formatted | Mayes, Melanie A. | 11/17/20 8:18:00 PM |
|---|---|---|

Font: 10 pt

| Page 7: [97] Formatted | Mayes, Melanie A. | 11/17/20 8:18:00 PM |
|---|---|---|

Font: 10 pt

| Page 7: [98] Formatted | Mayes, Melanie A. | 11/17/20 8:18:00 PM |
|---|---|---|

Font: 10 pt

| Page 7: [98] Formatted | Mayes, Melanie A. | 11/17/20 8:18:00 PM |
|---|---|---|

Font: 10 pt

| Page 7: [98] Formatted | Mayes, Melanie A. | 11/17/20 8:18:00 PM |
|---|---|---|

Font: 10 pt

| Page 7: [98] Formatted | Mayes, Melanie A. | 11/17/20 8:18:00 PM |
|---|---|---|

Font: 10 pt

| Page 7: [98] Formatted | Mayes, Melanie A. | 11/17/20 8:18:00 PM |
|---|---|---|

Font: 10 pt

| Page 7: [98] Formatted | Mayes, Melanie A. | 11/17/20 8:18:00 PM |
|---|---|---|

Font: 10 pt

| Page 7: [98] Formatted | Mayes, Melanie A. | 11/17/20 8:18:00 PM |
|---|---|---|

Font: 10 pt

| Page 7: [98] Formatted | Mayes, Melanie A. | 11/17/20 8:18:00 PM |
|---|---|---|

Font: 10 pt

| Page 7: [98] Formatted | Mayes, Melanie A. | 11/17/20 8:18:00 PM |
|---|---|---|

Font: 10 pt

| Page 7: [98] Formatted | Mayes, Melanie A. | 11/17/20 8:18:00 PM |
|---|---|---|

Font: 10 pt

| Page 7: [98] Formatted | Mayes, Melanie A. | 11/17/20 8:18:00 PM |
|---|---|---|

Font: 10 pt

| Page 7: [98] Formatted | Mayes, Melanie A. | 11/17/20 8:18:00 PM |
|---|---|---|

Font: 10 pt

| Page 7: [98] Formatted | Mayes, Melanie A. | 11/17/20 8:18:00 PM |
|---|---|---|

Font: 10 pt

| Page 7: [98] Formatted | Mayes, Melanie A. | 11/17/20 8:18:00 PM |
|---|---|---|

Font: 10 pt

| Page 7: [99] Formatted | Mayes, Melanie A. | 11/17/20 8:18:00 PM |
|---|---|---|

Font: (Default) +Body (Times New Roman), 10 pt, Not Italic

| Page 7: [100] Formatted | Mayes, Melanie A. | 11/17/20 8:18:00 PM |
|---|---|---|

Font: (Default) +Body (Times New Roman), 10 pt

| Page 7: [100] Formatted | Mayes, Melanie A. | 11/17/20 8:18:00 PM |
|---|---|---|

Font: (Default) +Body (Times New Roman), 10 pt

| Page 7: [101] Formatted | Mayes, Melanie A. | 11/17/20 8:18:00 PM |
|---|---|---|

Font: (Default) +Body (Times New Roman), 10 pt

| Page 7: [101] Formatted | Mayes, Melanie A. | 11/17/20 8:18:00 PM |
|---|---|---|

Font: (Default) +Body (Times New Roman), 10 pt

| Page 7: [102] Formatted | Mayes, Melanie A. | 11/17/20 8:18:00 PM |
|---|---|---|

Font: (Default) +Body (Times New Roman), 10 pt

| Page 7: [103] Formatted | Mayes, Melanie A. | 11/17/20 8:18:00 PM |
|---|---|---|

Font: (Default) +Body (Times New Roman), 10 pt

| Page 7: [104] Formatted | Mayes, Melanie A. | 11/17/20 8:18:00 PM |
|---|---|---|

Font: (Default) +Body (Times New Roman), 10 pt, Not Italic

| Page 7: [104] Formatted | Mayes, Melanie A. | 11/17/20 8:18:00 PM |
|---|---|---|

Font: (Default) +Body (Times New Roman), 10 pt, Not Italic

| Page 7: [104] Formatted | Mayes, Melanie A. | 11/17/20 8:18:00 PM |
|---|---|---|

Font: (Default) +Body (Times New Roman), 10 pt, Not Italic

| Page 7: [104] Formatted | Mayes, Melanie A. | 11/17/20 8:18:00 PM |
|---|---|---|

Font: (Default) +Body (Times New Roman), 10 pt, Not Italic

| Page 7: [105] Formatted | Mayes, Melanie A. | 11/17/20 8:18:00 PM |
|---|---|---|

Font: (Default) +Body (Times New Roman), 10 pt, Not Italic

| Page 7: [105] Formatted | Mayes, Melanie A. | 11/17/20 8:18:00 PM |
|---|---|---|

Font: (Default) +Body (Times New Roman), 10 pt, Not Italic

| Page 7: [106] Formatted | Mayes, Melanie A. | 11/17/20 8:18:00 PM |
|---|---|---|

Font: (Default) +Body (Times New Roman), 10 pt

| Page 7: [107] Deleted | Mayes, Melanie A. | 11/17/20 11:56:00 AM |
|---|---|---|

| Page 7: [107] Deleted | Mayes, Melanie A. | 11/17/20 11:56:00 AM |
|---|---|---|

| Page 7: [107] Deleted | Mayes, Melanie A. | 11/17/20 11:56:00 AM |
|---|---|---|

| Page 7: [107] Deleted | Mayes, Melanie A. | 11/17/20 11:56:00 AM |
|---|---|---|

| Page 7: [108] Formatted | Mayes, Melanie A. | 11/17/20 8:18:00 PM |
|---|---|---|

Font: (Default) +Body (Times New Roman), 10 pt

| Page 7: [108] Formatted | Mayes, Melanie A. | 11/17/20 8:18:00 PM |
|---|---|---|

Font: (Default) +Body (Times New Roman), 10 pt

| Page 7: [109] Formatted | Mayes, Melanie A. | 11/17/20 8:18:00 PM |
|---|---|---|

Font: (Default) +Body (Times New Roman), 10 pt

| Page 7: [109] Formatted | Mayes, Melanie A. | 11/17/20 8:18:00 PM |
|---|---|---|

Font: (Default) +Body (Times New Roman), 10 pt

| Page 7: [110] Formatted | Mayes, Melanie A. | 11/17/20 8:18:00 PM |
|---|---|---|

Font: (Default) +Body (Times New Roman), 10 pt, Not Italic

| Page 7: [110] Formatted | Mayes, Melanie A. | 11/17/20 8:18:00 PM |
|---|---|---|

Font: (Default) +Body (Times New Roman), 10 pt, Not Italic

| Page 7: [111] Formatted | Mayes, Melanie A. | 11/17/20 8:18:00 PM |
|---|---|---|

Font: (Default) +Body (Times New Roman), 10 pt

| Page 7: [111] Formatted | Mayes, Melanie A. | 11/17/20 8:18:00 PM |
|---|---|---|

Font: (Default) +Body (Times New Roman), 10 pt

| Page 7: [111] Formatted | Mayes, Melanie A. | 11/17/20 8:18:00 PM |
|---|---|---|

Font: (Default) +Body (Times New Roman), 10 pt

| Page 8: [112] Formatted | Mayes, Melanie A. | 11/17/20 8:18:00 PM |
|---|---|---|

Font: (Default) +Body (Times New Roman), 10 pt, Subscript

| Page 8: [112] Formatted | Mayes, Melanie A. | 11/17/20 8:18:00 PM |
|---|---|---|

Font: (Default) +Body (Times New Roman), 10 pt, Subscript

| Page 8: [112] Formatted | Mayes, Melanie A. | 11/17/20 8:18:00 PM |
|---|---|---|

Font: (Default) +Body (Times New Roman), 10 pt, Subscript

| Page 8: [112] Formatted | Mayes, Melanie A. | 11/17/20 8:18:00 PM |
|---|---|---|

Font: (Default) +Body (Times New Roman), 10 pt, Subscript

| Page 8: [112] Formatted | Mayes, Melanie A. | 11/17/20 8:18:00 PM |
|---|---|---|

Font: (Default) +Body (Times New Roman), 10 pt, Subscript

| Page 8: [112] Formatted | Mayes, Melanie A. | 11/17/20 8:18:00 PM |
|---|---|---|

Font: (Default) +Body (Times New Roman), 10 pt, Subscript

| Page 8: [112] Formatted | Mayes, Melanie A. | 11/17/20 8:18:00 PM |
|---|---|---|

Font: (Default) +Body (Times New Roman), 10 pt, Subscript

| Page 8: [113] Deleted | Whendee Silver | 11/16/20 9:57:00 PM |
|---|---|---|

| Page 8: [113] Deleted | Whendee Silver | 11/16/20 9:57:00 PM |
|---|---|---|

| Page 8: [114] Formatted | Mayes, Melanie A. | 11/17/20 8:18:00 PM |
|---|---|---|

Font: (Default) +Body (Times New Roman), 10 pt, Not Italic

| Page 8: [114] Formatted | Mayes, Melanie A. | 11/17/20 8:18:00 PM |
|---|---|---|

Font: (Default) +Body (Times New Roman), 10 pt, Not Italic

| Page 8: [114] Formatted | Mayes, Melanie A. | 11/17/20 8:18:00 PM |
|---|---|---|

Font: (Default) +Body (Times New Roman), 10 pt, Not Italic

| Page 8: [114] Formatted | Mayes, Melanie A. | 11/17/20 8:18:00 PM |
|---|---|---|

Font: (Default) +Body (Times New Roman), 10 pt, Not Italic

| Page 8: [114] Formatted | Mayes, Melanie A. | 11/17/20 8:18:00 PM |
|---|---|---|

Font: (Default) +Body (Times New Roman), 10 pt, Not Italic

| Page 8: [114] Formatted | Mayes, Melanie A. | 11/17/20 8:18:00 PM |
|---|---|---|

Font: (Default) +Body (Times New Roman), 10 pt, Not Italic

| Page 8: [115] Deleted | Mayes, Melanie A. | 11/14/20 3:51:00 PM |
|---|---|---|

| Page 8: [115] Deleted | Mayes, Melanie A. | 11/14/20 3:51:00 PM |
|---|---|---|

| Page 8: [116] Formatted | Mayes, Melanie A. | 11/17/20 8:18:00 PM |
|---|---|---|

Font: 10 pt

| Page 8: [116] Formatted | Mayes, Melanie A. | 11/17/20 8:18:00 PM |
|---|---|---|

Font: 10 pt

| Page 8: [116] Formatted | Mayes, Melanie A. | 11/17/20 8:18:00 PM |
|---|---|---|

Font: 10 pt

| Page 8: [116] Formatted | Mayes, Melanie A. | 11/17/20 8:18:00 PM |
|---|---|---|

Font: 10 pt

| Page 8: [116] Formatted | Mayes, Melanie A. | 11/17/20 8:18:00 PM |
|---|---|---|

Font: 10 pt

| Page 8: [116] Formatted | Mayes, Melanie A. | 11/17/20 8:18:00 PM |
|---|---|---|

Font: 10 pt

| Page 8: [116] Formatted | Mayes, Melanie A. | 11/17/20 8:18:00 PM |
|---|---|---|

Font: 10 pt

| Page 8: [116] Formatted | Mayes, Melanie A. | 11/17/20 8:18:00 PM |
|---|---|---|

Font: 10 pt

| Page 8: [116] Formatted | Mayes, Melanie A. | 11/17/20 8:18:00 PM |
|---|---|---|

Font: 10 pt

| Page 8: [116] Formatted | Mayes, Melanie A. | 11/17/20 8:18:00 PM |
|---|---|---|

Font: 10 pt

| Page 8: [116] Formatted | Mayes, Melanie A. | 11/17/20 8:18:00 PM |
|---|---|---|

Font: 10 pt

| Page 8: [116] Formatted | Mayes, Melanie A. | 11/17/20 8:18:00 PM |
|---|---|---|

Font: 10 pt

| Page 8: [117] Formatted | Mayes, Melanie A. | 11/17/20 8:18:00 PM |
|---|---|---|

Font: 10 pt

| Page 8: [117] Formatted | Mayes, Melanie A. | 11/17/20 8:18:00 PM |
|---|---|---|

Font: 10 pt

| Page 8: [117] Formatted | Mayes, Melanie A. | 11/17/20 8:18:00 PM |
|---|---|---|

Font: 10 pt

| Page 8: [118] Formatted | Mayes, Melanie A. | 11/17/20 8:18:00 PM |
|---|---|---|

Font: (Default) +Body (Times New Roman), 10 pt, Not Italic

| Page 8: [118] Formatted | Mayes, Melanie A. | 11/17/20 8:18:00 PM |
|---|---|---|

Font: (Default) +Body (Times New Roman), 10 pt, Not Italic

| Page 8: [118] Formatted | Mayes, Melanie A. | 11/17/20 8:18:00 PM |
|---|---|---|

Font: (Default) +Body (Times New Roman), 10 pt, Not Italic

| Page 8: [119] Formatted | Mayes, Melanie A. | 11/17/20 8:18:00 PM |
|---|---|---|

Font: (Default) +Body (Times New Roman), 10 pt

| Page 8: [119] Formatted | Mayes, Melanie A. | 11/17/20 8:18:00 PM |
|---|---|---|

Font: (Default) +Body (Times New Roman), 10 pt

| Page 8: [119] Formatted | Mayes, Melanie A. | 11/17/20 8:18:00 PM |
|---|---|---|

Font: (Default) +Body (Times New Roman), 10 pt

| Page 8: [120] Deleted | Whendee Silver | 11/16/20 10:05:00 PM |
|---|---|---|

| Page 8: [120] Deleted | Whendee Silver | 11/16/20 10:05:00 PM |
|---|---|---|

| Page 8: [121] Formatted | Mayes, Melanie A. | 11/17/20 8:18:00 PM |
|---|---|---|

Font: (Default) +Body (Times New Roman), 10 pt

| Page 8: [121] Formatted | Mayes, Melanie A. | 11/17/20 8:18:00 PM |
|---|---|---|

Font: (Default) +Body (Times New Roman), 10 pt

| Page 8: [121] Formatted | Mayes, Melanie A. | 11/17/20 8:18:00 PM |
|---|---|---|

Font: (Default) +Body (Times New Roman), 10 pt

| Page 8: [121] Formatted | Mayes, Melanie A. | 11/17/20 8:18:00 PM |
|---|---|---|

Font: (Default) +Body (Times New Roman), 10 pt

| Page 8: [122] Formatted | Mayes, Melanie A. | 11/17/20 8:18:00 PM |
|---|---|---|

Font: (Default) +Body (Times New Roman), 10 pt

| Page 8: [122] Formatted | Mayes, Melanie A. | 11/17/20 8:18:00 PM |
|---|---|---|

Font: (Default) +Body (Times New Roman), 10 pt

| Page 8: [122] Formatted | Mayes, Melanie A. | 11/17/20 8:18:00 PM |
|---|---|---|

Font: (Default) +Body (Times New Roman), 10 pt

| Page 8: [122] Formatted | Mayes, Melanie A. | 11/17/20 8:18:00 PM |
|---|---|---|

Font: (Default) +Body (Times New Roman), 10 pt

| Page 8: [122] Formatted | Mayes, Melanie A. | 11/17/20 8:18:00 PM |
|---|---|---|

Font: (Default) +Body (Times New Roman), 10 pt

| Page 8: [123] Formatted | Mayes, Melanie A. | 11/17/20 8:18:00 PM |
|---|---|---|

Font: (Default) +Body (Times New Roman), 10 pt

| Page 8: [123] Formatted | Mayes, Melanie A. | 11/17/20 8:18:00 PM |
|---|---|---|

Font: (Default) +Body (Times New Roman), 10 pt

**Page 8: [123] Formatted**      **Mayes, Melanie A.**      **11/17/20 8:18:00 PM**

Font: (Default) +Body (Times New Roman), 10 pt

**Page 8: [124] Formatted**      **Mayes, Melanie A.**      **11/17/20 8:18:00 PM**

Font: (Default) +Body (Times New Roman), 10 pt

**Page 8: [124] Formatted**      **Mayes, Melanie A.**      **11/17/20 8:18:00 PM**

Font: (Default) +Body (Times New Roman), 10 pt

**Page 8: [124] Formatted**      **Mayes, Melanie A.**      **11/17/20 8:18:00 PM**

Font: (Default) +Body (Times New Roman), 10 pt

**Page 8: [124] Formatted**      **Mayes, Melanie A.**      **11/17/20 8:18:00 PM**

Font: (Default) +Body (Times New Roman), 10 pt

**Page 8: [125] Formatted**      **Mayes, Melanie A.**      **11/17/20 8:18:00 PM**

Font: (Default) +Body (Times New Roman), 10 pt

**Page 8: [125] Formatted**      **Mayes, Melanie A.**      **11/17/20 8:18:00 PM**

Font: (Default) +Body (Times New Roman), 10 pt

**Page 8: [126] Formatted**      **Mayes, Melanie A.**      **11/17/20 8:18:00 PM**

Font: (Default) +Body (Times New Roman), 10 pt, Not Italic

**Page 8: [126] Formatted**      **Mayes, Melanie A.**      **11/17/20 8:18:00 PM**

Font: (Default) +Body (Times New Roman), 10 pt, Not Italic

**Page 8: [126] Formatted**      **Mayes, Melanie A.**      **11/17/20 8:18:00 PM**

Font: (Default) +Body (Times New Roman), 10 pt, Not Italic

**Page 8: [126] Formatted**      **Mayes, Melanie A.**      **11/17/20 8:18:00 PM**

Font: (Default) +Body (Times New Roman), 10 pt, Not Italic

**Page 8: [126] Formatted**      **Mayes, Melanie A.**      **11/17/20 8:18:00 PM**

Font: (Default) +Body (Times New Roman), 10 pt, Not Italic

**Page 8: [126] Formatted**      **Mayes, Melanie A.**      **11/17/20 8:18:00 PM**

Font: (Default) +Body (Times New Roman), 10 pt, Not Italic

**Page 8: [126] Formatted**      **Mayes, Melanie A.**      **11/17/20 8:18:00 PM**

Font: (Default) +Body (Times New Roman), 10 pt, Not Italic

**Page 8: [126] Formatted**      **Mayes, Melanie A.**      **11/17/20 8:18:00 PM**

Font: (Default) +Body (Times New Roman), 10 pt, Not Italic

**Page 8: [126] Formatted**      **Mayes, Melanie A.**      **11/17/20 8:18:00 PM**

Font: (Default) +Body (Times New Roman), 10 pt, Not Italic

**Page 8: [126] Formatted**      **Mayes, Melanie A.**      **11/17/20 8:18:00 PM**

Font: (Default) +Body (Times New Roman), 10 pt, Not Italic

**Page 8: [127] Formatted**      **Mayes, Melanie A.**      **11/17/20 8:18:00 PM**

Font: (Default) +Body (Times New Roman), 10 pt

**Page 8: [127] Formatted**      **Mayes, Melanie A.**      **11/17/20 8:18:00 PM**

Font: (Default) +Body (Times New Roman), 10 pt

| Page 13: [128] Formatted | Mayes, Melanie A. | 11/17/20 8:18:00 PM |
|---|---|---|

Font: (Default) +Body (Times New Roman), 10 pt, Not Highlight

| Page 13: [128] Formatted | Mayes, Melanie A. | 11/17/20 8:18:00 PM |
|---|---|---|

Font: (Default) +Body (Times New Roman), 10 pt, Not Highlight

| Page 13: [129] Deleted | Mayes, Melanie A. | 11/15/20 11:12:00 AM |
|---|---|---|

| Page 13: [129] Deleted | Mayes, Melanie A. | 11/15/20 11:12:00 AM |
|---|---|---|

| Page 13: [130] Formatted | Mayes, Melanie A. | 11/17/20 8:18:00 PM |
|---|---|---|

Font: (Default) +Body (Times New Roman), 10 pt, Not Italic

| Page 13: [130] Formatted | Mayes, Melanie A. | 11/17/20 8:18:00 PM |
|---|---|---|

Font: (Default) +Body (Times New Roman), 10 pt, Not Italic

| Page 13: [130] Formatted | Mayes, Melanie A. | 11/17/20 8:18:00 PM |
|---|---|---|

Font: (Default) +Body (Times New Roman), 10 pt, Not Italic

| Page 13: [130] Formatted | Mayes, Melanie A. | 11/17/20 8:18:00 PM |
|---|---|---|

Font: (Default) +Body (Times New Roman), 10 pt, Not Italic

| Page 13: [130] Formatted | Mayes, Melanie A. | 11/17/20 8:18:00 PM |
|---|---|---|

Font: (Default) +Body (Times New Roman), 10 pt, Not Italic

| Page 13: [130] Formatted | Mayes, Melanie A. | 11/17/20 8:18:00 PM |
|---|---|---|

Font: (Default) +Body (Times New Roman), 10 pt, Not Italic

| Page 13: [130] Formatted | Mayes, Melanie A. | 11/17/20 8:18:00 PM |
|---|---|---|

Font: (Default) +Body (Times New Roman), 10 pt, Not Italic

| Page 13: [130] Formatted | Mayes, Melanie A. | 11/17/20 8:18:00 PM |
|---|---|---|

Font: (Default) +Body (Times New Roman), 10 pt, Not Italic

| Page 13: [130] Formatted | Mayes, Melanie A. | 11/17/20 8:18:00 PM |
|---|---|---|

Font: (Default) +Body (Times New Roman), 10 pt, Not Italic

| Page 13: [130] Formatted | Mayes, Melanie A. | 11/17/20 8:18:00 PM |
|---|---|---|

Font: (Default) +Body (Times New Roman), 10 pt, Not Italic

| Page 13: [130] Formatted | Mayes, Melanie A. | 11/17/20 8:18:00 PM |
|---|---|---|

Font: (Default) +Body (Times New Roman), 10 pt, Not Italic

| Page 13: [130] Formatted | Mayes, Melanie A. | 11/17/20 8:18:00 PM |
|---|---|---|

Font: (Default) +Body (Times New Roman), 10 pt, Not Italic

| Page 13: [130] Formatted | Mayes, Melanie A. | 11/17/20 8:18:00 PM |
|---|---|---|

Font: (Default) +Body (Times New Roman), 10 pt, Not Italic

| Page 13: [130] Formatted | Mayes, Melanie A. | 11/17/20 8:18:00 PM |
|---|---|---|

Font: (Default) +Body (Times New Roman), 10 pt, Not Italic

**Page 13: [130] Formatted**        Mayes, Melanie A.        11/17/20 8:18:00 PM

Font: (Default) +Body (Times New Roman), 10 pt, Not Italic

**Page 13: [130] Formatted**        Mayes, Melanie A.        11/17/20 8:18:00 PM

Font: (Default) +Body (Times New Roman), 10 pt, Not Italic

**Page 13: [130] Formatted**        Mayes, Melanie A.        11/17/20 8:18:00 PM

Font: (Default) +Body (Times New Roman), 10 pt, Not Italic

**Page 13: [130] Formatted**        Mayes, Melanie A.        11/17/20 8:18:00 PM

Font: (Default) +Body (Times New Roman), 10 pt, Not Italic

**Page 13: [130] Formatted**        Mayes, Melanie A.        11/17/20 8:18:00 PM

Font: (Default) +Body (Times New Roman), 10 pt, Not Italic

**Page 13: [130] Formatted**        Mayes, Melanie A.        11/17/20 8:18:00 PM

Font: (Default) +Body (Times New Roman), 10 pt, Not Italic

**Page 13: [130] Formatted**        Mayes, Melanie A.        11/17/20 8:18:00 PM

Font: (Default) +Body (Times New Roman), 10 pt, Not Italic

**Page 13: [130] Formatted**        Mayes, Melanie A.        11/17/20 8:18:00 PM

Font: (Default) +Body (Times New Roman), 10 pt, Not Italic

**Page 13: [130] Formatted**        Mayes, Melanie A.        11/17/20 8:18:00 PM

Font: (Default) +Body (Times New Roman), 10 pt, Not Italic

**Page 13: [130] Formatted**        Mayes, Melanie A.        11/17/20 8:18:00 PM

Font: (Default) +Body (Times New Roman), 10 pt, Not Italic

**Page 13: [130] Formatted**        Mayes, Melanie A.        11/17/20 8:18:00 PM

Font: (Default) +Body (Times New Roman), 10 pt, Not Italic

**Page 13: [130] Formatted**        Mayes, Melanie A.        11/17/20 8:18:00 PM

Font: (Default) +Body (Times New Roman), 10 pt, Not Italic

**Page 13: [130] Formatted**        Mayes, Melanie A.        11/17/20 8:18:00 PM

Font: (Default) +Body (Times New Roman), 10 pt, Not Italic

**Page 13: [130] Formatted**        Mayes, Melanie A.        11/17/20 8:18:00 PM

Font: (Default) +Body (Times New Roman), 10 pt, Not Italic

**Page 13: [130] Formatted**        Mayes, Melanie A.        11/17/20 8:18:00 PM

Font: (Default) +Body (Times New Roman), 10 pt, Not Italic

**Page 13: [130] Formatted**        Mayes, Melanie A.        11/17/20 8:18:00 PM

Font: (Default) +Body (Times New Roman), 10 pt, Not Italic

**Page 13: [130] Formatted**        Mayes, Melanie A.        11/17/20 8:18:00 PM

Font: (Default) +Body (Times New Roman), 10 pt, Not Italic

**Page 13: [130] Formatted**        Mayes, Melanie A.        11/17/20 8:18:00 PM

Font: (Default) +Body (Times New Roman), 10 pt, Not Italic

| Page 13: [130] Formatted | Mayes, Melanie A. | 11/17/20 8:18:00 PM |
|---|---|---|

Font: (Default) +Body (Times New Roman), 10 pt, Not Italic

| Page 13: [130] Formatted | Mayes, Melanie A. | 11/17/20 8:18:00 PM |
|---|---|---|

Font: (Default) +Body (Times New Roman), 10 pt, Not Italic

| Page 13: [130] Formatted | Mayes, Melanie A. | 11/17/20 8:18:00 PM |
|---|---|---|

Font: (Default) +Body (Times New Roman), 10 pt, Not Italic

| Page 13: [130] Formatted | Mayes, Melanie A. | 11/17/20 8:18:00 PM |
|---|---|---|

Font: (Default) +Body (Times New Roman), 10 pt, Not Italic

| Page 13: [130] Formatted | Mayes, Melanie A. | 11/17/20 8:18:00 PM |
|---|---|---|

Font: (Default) +Body (Times New Roman), 10 pt, Not Italic

| Page 13: [130] Formatted | Mayes, Melanie A. | 11/17/20 8:18:00 PM |
|---|---|---|

Font: (Default) +Body (Times New Roman), 10 pt, Not Italic

| Page 13: [130] Formatted | Mayes, Melanie A. | 11/17/20 8:18:00 PM |
|---|---|---|

Font: (Default) +Body (Times New Roman), 10 pt, Not Italic

| Page 13: [130] Formatted | Mayes, Melanie A. | 11/17/20 8:18:00 PM |
|---|---|---|

Font: (Default) +Body (Times New Roman), 10 pt, Not Italic

| Page 13: [130] Formatted | Mayes, Melanie A. | 11/17/20 8:18:00 PM |
|---|---|---|

Font: (Default) +Body (Times New Roman), 10 pt, Not Italic

| Page 13: [130] Formatted | Mayes, Melanie A. | 11/17/20 8:18:00 PM |
|---|---|---|

Font: (Default) +Body (Times New Roman), 10 pt, Not Italic

| Page 13: [130] Formatted | Mayes, Melanie A. | 11/17/20 8:18:00 PM |
|---|---|---|

Font: (Default) +Body (Times New Roman), 10 pt, Not Italic

| Page 13: [131] Deleted | Mayes, Melanie A. | 11/15/20 3:53:00 PM |
|---|---|---|

| Page 13: [131] Deleted | Mayes, Melanie A. | 11/15/20 3:53:00 PM |
|---|---|---|

| Page 13: [131] Deleted | Mayes, Melanie A. | 11/15/20 3:53:00 PM |
|---|---|---|

| Page 13: [131] Deleted | Mayes, Melanie A. | 11/15/20 3:53:00 PM |
|---|---|---|

| Page 23: [132] Deleted | Mayes, Melanie A. | 11/10/20 2:17:00 PM |
|---|---|---|